# VisualAgentBench: Towards Large Multimodal Models as Visual Foundation Agents

Xiao Liu[1,*], Tianjie Zhang[3,†,*], Yu Gu[2,*], Iat Long Iong[1], Yifan Xu[1], Xixuan Song[1], Shudan Zhang[1], Hanyu Lai[1], Xinyi Liu[1,†], Hanlin Zhao[1,†], Jiadai Sun[1,†], Xinyue Yang[1,†], Yu Yang[1,†], Zehan Qi[1], Shuntian Yao[1,†], Xueqiao Sun[1], Siyi Cheng[4,†], Qinkai Zheng[1], Hao Yu[1], Hanchen Zhang[1], Wenyi Hong[1], Ming Ding[1], Lihang Pan[1], Xiaotao Gu[1], Aohan Zeng[1], Zhengxiao Du[1], Chan Hee Song[2], Yu Su[2], Yuxiao Dong[1], Jie Tang[1]

[1]Tsinghua University, [2]The Ohio State University, [3]Zhejiang University, [4]Peking University

## ABSTRACT

Large Multimodal Models (LMMs) have ushered in a new era in artificial intelligence, merging capabilities in both language and vision to form highly capable **Visual Foundation Agents** that are postulated to excel across a myriad of tasks. However, existing benchmarks fail to sufficiently challenge or showcase the full potential of LMMs as visual foundation agents in complex, real-world environments. To address this gap, we introduce VisualAgentBench (VAB), a comprehensive and unified benchmark specifically designed to train and evaluate LMMs as visual foundation agents across diverse scenarios in one standard setting, including Embodied, Graphical User Interface, and Visual Design, with tasks formulated to probe the depth of LMMs' understanding and interaction capabilities. Through rigorous testing across 9 proprietary LMM APIs and 9 open models (18 in total), we demonstrate the considerable yet still developing visual agent capabilities of these models. Additionally, VAB explores the synthesizing of visual agent trajectory data through hybrid methods including Program-based Solvers, LMM Agent Bootstrapping, and Human Demonstrations, offering insights into obstacles, solutions, and trade-offs one may meet in developing open LMM agents. Our work not only aims to benchmark existing models but also provides an instrumental playground for future development into visual foundation agents. Code, train, and test data are available at https://github.com/THUDM/VisualAgentBench.

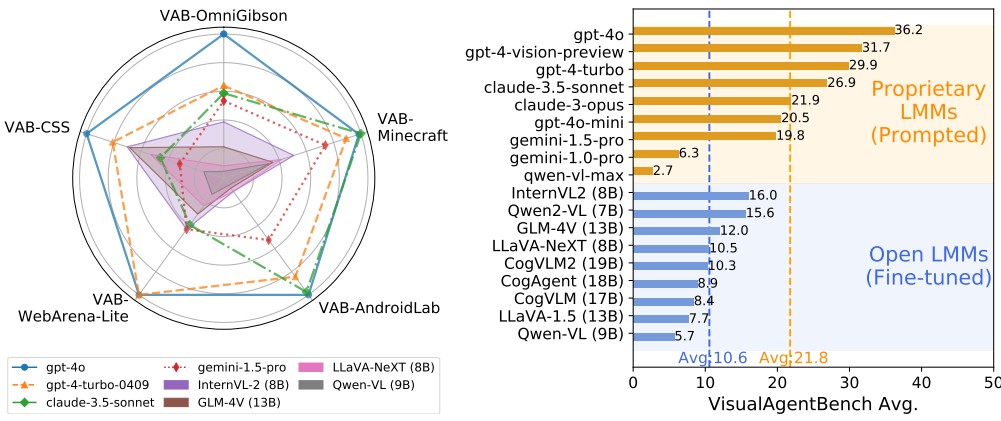

(a) Typical LMMs' VAB performance (relative) against the best in each environment.

(b) Average VAB Success Rates of tested LMMs across 5 environments. Dashed lines for two LMM types' average.

Figure 1: Overview of Proprietary and Open LMMs on VisualAgentBench. After Behavior Cloning (BC) on trajectories, Open LMMs demonstrate potential to serve as visual foundation agents.

---

*Equal contribution. Email: {shawliu9,mistyreed63849}@gmail.com, gu.826@osu.edu
†Work done when these authors visited Tsinghua University. Corresponding author: JT.

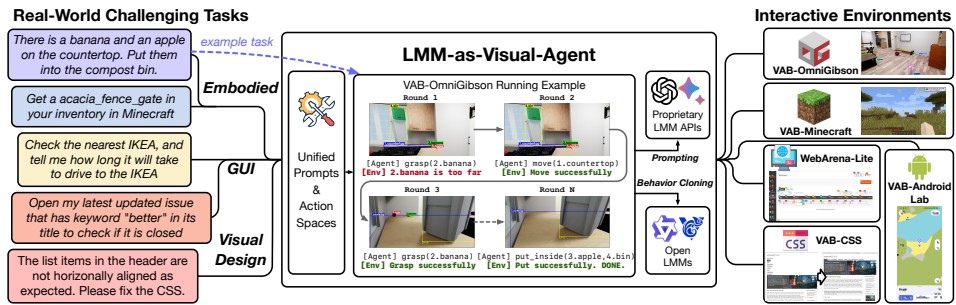

Figure 2: VISUALAGENTBENCH is the first systematic benchmark for training and evaluating LMM-as-Visual-Foundation-Agent with both proprietary and open LMMs across a diverse set of practical challenges. Based on created tasks, we unify the benchmarking of both proprietary LMM APIs via prompting and open LMMs via behavior cloning training in interactive environments.

# 1 INTRODUCTION

Recent advancements in Foundation Models, particularly Large Language Models (LLMs) (Brown et al., 2020; Chowdhery et al., 2022; Touvron et al., 2023; Zeng et al., 2022) and Large Multimodal Models (LMMs) (Liu et al., 2024c; OpenAI, 2023; 2024a; Anthropic, 2024), have showcased their profound capabilities in understanding and processing vast amounts of world knowledge, factual information, and common sense reasoning. Notably, these models have demonstrated potential as intelligent agents (Searle, 1970; Maes, 1994; Wooldridge & Jennings, 1995), addressing a broad spectrum of real-world challenges (Liu et al., 2023a). LMMs, in particular, enhance the capabilities of these agents by integrating visual inputs, thereby expanding the scope of intelligent agent applications.

This progress has given rise to the concept of **Foundation Agents**—generalist agents adept at mastering a plethora of skills across various virtual and embodied environments, mirroring human versatility. These agents, especially those powered by LMMs, are envisioned to excel in multitask environments without the need for task-specific fine-tuning, a paradigm already set by LLM-based language agents. The burgeoning field of visual foundation agents offers promising pathways toward achieving AGI, with the potential to significantly elevate human productivity and creativity.

However, the setup for LMM-as-Visual-Foundation-Agent remains underdeveloped. Most existing evaluations on LMMs focus on traditional vision tasks (Singh et al., 2019; Lu et al., 2022; Liu et al., 2023b; Kazemzadeh et al., 2014), or on performance in standardized human exams (Yue et al., 2023; Lu et al., 2023). They rarely measure the models' higher-level planning abilities or specific strengths as visual agents. In contrast, the role of LLMs as agents in text environments has been extensively explored and validated as a reliable measure of their capabilities (Yao et al., 2023; Liu et al., 2023a).

Recent benchmarks for multimodal agents, while valuable, inadequately address the comprehensive multitask evaluation required for LMM-as-Visual-Foundation-Agent. Their focuses are often limited to single environments such as Household (Shridhar et al., 2020a;b), Gaming (Fan et al., 2022; Wu et al., 2023), Web (Deng et al., 2024; Zhou et al., 2023; Koh et al., 2024), or Desktop (Xie et al., 2024; Kapoor et al., 2024). The narrow scope and varied settings prevents a holistic assessment of LMMs' multitask agent capabilities. Furthermore, the prevalent prompting-only evaluation in existing benchmarks does not suffice for open LMMs (Liu et al., 2024c; Bai et al., 2023), which typically show limited instruction-following capabilities so far, thus hindering a comprehensive evaluation.

To bridge this gap, we introduce VISUALAGENTBENCH (VAB)—the first systematic benchmark to multitask train and evaluate visual foundation agents across a diverse array of realistic vision-centric tasks. We present three representative scenarios and develop five distinct datasets for this study: *Embodied* (VAB-OmniGibson, VAB-Minecraft), *Graphical User Interface (GUI)* (VAB-AndroidLab (Xu et al., 2024), VAB-WebArena-lite (Zhou et al., 2023)), and *Visual Design* (VAB-CSS), enabling comprehensive testing and development of agents that can navigate complex spaces, interact with digital interfaces (Liu et al., 2024d), and understand aesthetic and functional aspects of visual design. This diversity not only challenges the agents' capabilities across different settings but also enhances their adaptability and utility in practical applications, paving the way for more robust and versatile visual foundation agents.

**Contributions.** Our main contributions in VAB are as follow: Standardized Environments, LMM-oriented Test Data Creation, Train Data Synthesis, Extensive Benchmarking, and Analytical Insights.

- **Standardized Environments.** VAB standardizes the interfaces, prompting, and data formats to facilitate a consistent evaluation of LMM agents across environments. The efforts include adapting previously LMM-irrelevant environment and creating the new. Each task is assessed through interactive evaluation, where LMMs engage directly with the environment, and their performance is measured by specific judge functions. The feature substantially distinguishes VAB from many other benchmarks (Deng et al., 2024; Rawles et al., 2024; Kapoor et al., 2024) based on offline human-annotated trajectories.
- **LMM-oriented Test Data Creation.** Test set queries and judge functions are newly created for all environments except for WebArena. To acquire massive tasks, we develop a "Prototype-Instantiation" method to evolve valid task instructions and judge functions grounded on environments. We refer to task categories and prototypes in some previous study (Li et al., 2023; Zhu et al., 2023) which unsatisfies the practical use for evaluating LMM agents or has no public data.
- **Train Data Synthesis.** VAB strives to offer the first multitask multi-environment trajectory train set for developing LMM agents, containing $4,482$ high-quality training trajectories spanning 5 environments. It explores and investigate how to synthesize multimodal agent task instructions and trajectory data via hybrid strategies of ***Program-based Solvers, LMM Agent Bootstrapping, and Human Demonstrations***. Thereby, it also for the first time enables a holistic agent evaluation that includes many open LMMs with valid scores for meaningful comparison.
- **Extensive Benchmarking & Analysis.** Our extensive testing over 18 LMMs, including 9 proprietary LMM APIs and 9 open LMMs, demonstrates the impressive progress of LMM-as-Visual-Foundation-Agent. Top proprietary LMMs, such as `gpt-4o`, are solving 36.2% of challenging problems with mere prompting. Behavior cloning (BC) on the VAB train set remarkably enhances the capabilities of open LMMs as visual foundation agents, with most surpassing the performance of weaker proprietary LMMs and approaching close towards the strong `gemini-1.5-pro`.
- **Analytical Insights.** VAB provides deep insights into the general status quo and detailed dimensions of grounding and planning for LMM agents. For grounding, we quantitatively analyze the use of object labels, set-of-marks, and visual difference ability. For planning, we study the actual impact of Chain-of-Thought and error recovering ability for LMM agents.

## 2 VISUALAGENTBENCH: TASKS AND ENVIRONMENTS

In this section, we will first introduce the problem definition of LMM-as-Visual-Foundation-Agent, and then the detailed description of each environment and dataset.

**LMM-as-Visual-Foundation-Agent.** An agentic problem could be generally formulated as a Partially Observable Markov Decision Process (POMDP) problem, where $\mathcal{S}$ denotes the state space, $\mathcal{A}$ denotes the action space, $\mathcal{T}$ denotes the transition function, $\mathcal{R}$ refers to the reward function, $\mathcal{I}$ refers to the instruction space, and $\mathcal{O}$ refers to the observation space. Compared to LLM-as-Agent (Liu et al., 2023a), the observation space $\mathcal{O}$ must incorporate visual inputs (e.g., images or videos) in LMM-as-Visual-Foundation-Agent, significantly extending the application scope but also casting a substantial challenge for LMMs to reconcile their multimodal understanding and high-level reasoning.

**Overview of VAB.** In VAB, we carefully select the most representative and promising tasks that could be enabled by LMM-based agents. These tasks generally fall into three categories: embodied agents, including household and game environments; GUI agents, covering mobile and web apps; and visual design agents, focusing on frontend CSS debugging (Figure 2). They span diverse domains and feature unique challenges, providing an ideal testbed for a comprehensive evaluation of LMM-based agents. When constructing VAB, we strictly follow the principles outlined in Appendix A.1. Our efforts focus on addressing gaps in evaluating LMM-based agents while leveraging existing resources to avoid redundancy, ensuring all our work is meaningful and avoids reinventing the wheel. For 4 out of 5 tasks, we collect new data from scratch. For web agents, we adapt and clean WebArena (Zhou et al., 2023) as our test set, as it is already suitable for LMM-based evaluation. For household agents, we use the OmniGibson environment from Behavior-1k (Li et al., 2023) and create new tasks based on high-level actions we defined, which are crucial for evaluating LMM-based agents and absent in

Table 1: Comparison between VAB and related benchmarks. VAB is the first comprehensive multi-domain agent benchmark offering interactive environments, supporting multimodal agent evaluation, and providing a large and diverse set of training trajectories for visual agent tuning. "#Test Ins." refers to the number of test instances; "#Train Traj." refers to the number of training trajectories for SFT, "RL" means no training trajectory is available and only a reinforcement learning setup is provided; "#Avg. Turns" refers to the average number of turns per training trajectory.

| | Category | #Env. | #Test Ins. | #Train Traj. | #Avg. Turns | Multimodal | Interactive Env. |
|---|---|---|---|---|---|---|---|
| ALFWorld (Shridhar et al., 2020b) | Household | 1 | 134 | 6,374 | 7.54 | ✗ | ✓ |
| Alfred (Shridhar et al., 2020a) | Household | 1 | 1,529 | 6,574 | 7.51 | ✓ | ✓ |
| Behavior-1K (Li et al., 2023) | Household | 1 | 1,000 | RL | - | ✓ | ✓ |
| MineDojo (Fan et al., 2022) | Game | 1 | 3,141 | RL | - | ✓ | ✓ |
| SmartPlay (Wu et al., 2023) | Game | 6 | 20 | - | - | ✗ | ✓ |
| Mind2Web (Deng et al., 2024) | Web | 1 | 1,341 | 1,009 | 7.71 | ✓ | ✗ |
| WebArena (Zhou et al., 2023) | Web | 1 | 812 | - | - | ✓ | ✓ |
| VisualWebArena (Koh et al., 2024) | Web | 1 | 910 | - | - | ✓ | ✓ |
| META-GUI (Sun et al., 2022) | Mobile | 1 | 483 | 3,692 | 7.64 | ✓ | ✗ |
| OSWorld (Xie et al., 2024) | Desktop | 1 | 369 | - | - | ✓ | ✓ |
| OmniACT (Kapoor et al., 2024) | Desktop & Web | 2 | 9,802 | - | - | ✓ | ✗ |
| AgentBench (Liu et al., 2023a) | Multi-domain | 8 | 1,091 | - | - | ✗ | ✓ |
| VISUALAGENTBENCH | Multi-domain | 5 | 746 | 4,482 | 11.22 | ✓ | ✓ |

existing datasets. We similarly construct our tasks in Minecraft using the MineRL environment[1] with our self-defined high-level actions. Finally, for our mobile app and CSS debugging tasks, we create new interactive environments due to the lack of suitable resources in the literature and collect data based on these environments. An overview of VAB is shown in Table 1.

## 2.1 EMBODIED AGENT

Embodied agents have been a central topic in AI, naturally involving multimodal sensory data, including language and vision signals. The multimodal capabilities of LMMs could enable new possibilities for embodied agents.

**VAB-OmniGibson.** One of the most actively researched environments in embodied AI is the household environment due to its complexity and range of everyday tasks (Huang et al., 2022; Song et al., 2023; Shridhar et al., 2020a). We build the household environment for embodied agents using OmniGibson, a high-fidelity simulator based on Nvidia Omniverse that features diverse scenes and realistic physical effects.[2] An example activity in VAB-OmniGibson would be "*Put all 8 plates from the countertops into the cabinet in the kitchen*", where agents should accomplish the tasks using provided high-level actions (e.g.,"grasp", "put_inside"). We adopt the task *success rate* as the evaluation metric. (Cf. Appendix B).

**VAB-Minecraft.** Minecraft has become a popular open-world environment for developing generalist embodied agents due to its diverse tasks (e.g., survival, harvesting, crafting, combat, and creative tasks), varied environments, and interactive mobs, necessitating generalized agent abilities (Fan et al., 2022; Lifshitz et al., 2024). In VAB-Minecraft, the agent is expected to accomplish a wide range of tasks, including item collection and killing hostile mobs. An example task in VAB-Minecraft would be "*Get a fishing rod in your inventory*", and the LMM agent need to interact with the game environment using provided scripts (e.g.,"craft", "smelt") or calling a low-level controller Steve-1 (Lifshitz et al., 2024) with prompt. We adopt the task *success rate* as metric. (Cf. Appendix C)

## 2.2 GUI AGENT

GUI is another typical scenario where LMM agents may excel. Compared to embodied environments, GUI environments are more information-intensive and require a good understanding of UI elements and layouts. We provide two interactive and reproducible GUI environments, Mobile (i.e., Android) and WebArena, to evaluate LMM GUI agents in a practical manner.

---

[1]https://minerl.readthedocs.io
[2]https://www.nvidia.com/en-us/omniverse/

**VAB-AndroidLab (Xu et al., 2024).** Automated agents on Android GUI are instrumental. Although pioneer works like (Burns et al., 2022; Rawles et al., 2024) have explored training and evaluating these agents, they typically use Step Success Rate evaluated offline. Recent works (Yang et al., 2023b; Wang et al., 2024a) leverage LMMs as Android GUI agents but lack reproducible executive evaluation frameworks. We address this by creating tasks for LMM agents to perform human-like actions (e.g., `Tap`, `Swipe`) on smartphones using Android Virtual Device (AVD). For example, "*Find a hotpot restaurant nearby and make a reservation for me tonight.*" Agents must understand the Android GUI and make decisions based on screen observations (Cf. Appendix D). The visual SoM-based prototype developed in VAB for benchmark was later enriched with XML-based text setting, both text and multimodal training, and further detailed metrics and analysis on more models in (Xu et al., 2024).

**VAB-WebArena-Lite (Zhou et al., 2023).** Web browsing is an ideal testbed for evaluating LMMs as GUI agents. Previous works (Shi et al., 2017; Liu et al., 2018; Deng et al., 2024; Yao et al., 2022) mainly focus on offline evaluation. We adopt WebArena (Zhou et al., 2023), a benchmark for text-based web GUI agents with 812 tasks across 5 websites. LMMs perform tasks based on user instructions, such as finding and summarizing customer reviews on `OneStopShop`. We use HTML SoM (Koh et al., 2024) to annotate operable HTML elements, enabling LMMs to generate actions via `playwright`. WebArena-Lite is a subset of 165 tasks, refined and adapted for multimodal evaluation, removing cross-website tasks and fixing implausible conditions. (Cf. Appendix E)

### 2.3 VISUAL DESIGN AGENT

Visual design tasks demand a nuanced understanding of visual signals, which text-only LLMs cannot handle with any easy augmentation, unlike embodied or GUI agent tasks that can rely on external object detectors (Song et al., 2023) or textual representations like accessibility trees (Xie et al., 2024).

**VAB-CSS.** We create a new task to evaluate LMMs on web frontend design, focusing on CSS style adjustments. Fixing CSS styles is a labor-intensive task that often requires engineers to iteratively adjust an element through trial and error. Such a task inherently entails fine-grained visual grounding and reasoning across a series of rendering outcomes resulting from iterative CSS edits. In VAB-CSS, the agent iteratively edits the CSS style using provided tools until it thinks the rendering matches a given target design. We adopt *success rate (SR)* as the metric, which evaluates whether the final rendering matches the target design. (Cf. Appendix F)

## 3 METHODOLOGY FOR VAB DATA COLLECTION

For agent tasks, it is known to be very challenging to design practical and verifiable task instances; let alone creating high-quality training trajectories on top of them later. In constructing VAB, we not only aim to deliver a high-quality agent benchmark but also endeavor to develop a systematic methodology for the problem of LMM-as-Visual-Foundation-Agent data curation. For task instance collection, we follow a two-stage paradigm (*prototyping* and *instantiation*) for each new task instance to ensure data quality and executability. Additionally, we harness a suite of hybrid strategies to collect training trajectories that can be used to tune open LMMs into better visual foundation agents. Our rigorous data collection process in VAB is crucial for presenting a high-quality resource for LMM-based agents (Figure 3). The statistics of different tasks in VAB are shown in Table 2.

### 3.1 TASK INSTANCE COLLECTION: PROTOTYPING AND INSTANTIATION

Curating meaningful and testable task instances for LMM agent tasks can be difficult. On one hand, they should be diverse and useful to cover real-world applications. On the other hand, they should be grounded to environments carefully to ensure feasibility and practicality. As a result, we collect all our task instances in a two-stage paradigm:

- **Prototyping:** We gather many task prototypes representing high-level goals based on the functionality provided by the environment. Related items are temporarily set to placeholders.
- **Instantiation:** Task prototypes are grounded to concrete items and conditions collected from the environment. Judging functions are thereby set up by instantiated tasks. Instructions are then rephrased by LLMs to enhance expression diversity.

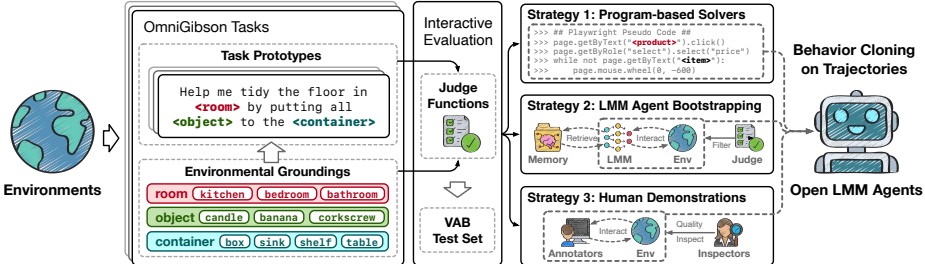

Figure 3: Data collection process in VAB. We follow a principled two-stage paradigm to collect task instances and then adopt various methods to further collect training trajectories for them.

Table 2: Statistics of all datasets in VAB.

|  | VAB-OmniGibson | VAB-Minecraft | VAB-AndroidLab | VAB-WebArena-Lite | VAB-CSS |
| --- | --- | --- | --- | --- | --- |
| #Action Space | 20 | 6 | 7 | 12 | 4 |
| #Test Instance | 181 | 116 | 119 | 165 | 165 |
| #Train Trajectory | 872 | 382 | 1,213 | 1,186 | 829 |
| #Train Step | 20,153 | 5,197 | 10,175 | 9,522 | 5,250 |
| #Max Round Limit | 100 | 100 | 25 | 20 | 10 |

Following the mentioned principles, we develop diverse task sets for all VAB environments. For VAB-OmniGibbon, we source 89 prototypes of general household activities, instantiating them in various scenes to create 992 instances, with 181 reserved for testing. In VAB-Minecraft, we focus on object-collecting tasks, generating 628 instances from our defined prototypes, allocating 116 for testing. For VAB-AndroidLab, we select 8 Android applications and create 119 test instructions, while developing approximately 70 task prototypes from 18 common apps for training. In WebArena-Lite, we refine 165 test samples and generate 1186 training instances from 40 task prototypes across various web applications. For VAB-CSS, we design tasks around CSS rule corruptions, creating 1210 instances with 165 for testing, each annotated with natural language descriptions of visual changes. This comprehensive approach ensures a rich and varied task environment across our selected domains. For more details, please refer to Appendix A.3.

## 3.2 TRAINING TRAJECTORY COLLECTION: 3-LEVELED STRATEGIES

Recently, there has been a rise in benchmarks for interactively evaluating LLM or LMM agents (Liu et al., 2023a; Zhou et al., 2023; Xie et al., 2024). Despite showcasing the substantial potential of LLM and LMM as agents, they usually only provide the test set and thus fail to facilitate the improving of open LLMs and LMMs on agent tasks. In light of the challenge, in VAB we are devoted to offering a preliminary behavior cloning (BC) (Nakano et al., 2021; Zeng et al., 2023) setup for training open LMM agents. In VAB, we summarize our trajectory collecting into 3-leveled strategies:

1. **Program-based Solvers:** Trajectories are collected by prototype-specific programs written by human experts (e.g., `Playwright` scripts for automating web browsing tasks).
2. **LMM Agent Bootstrapping:** Trajectories are collected by prompted LMM agents (e.g., `gpt-4o`), with optional memory augmentations (Wang et al., 2023c). For instance, in Minecraft we allow agent to access memories for solving easier sub-goals (e.g., how to collect a stick) when constructing trajectories for more complex goals (e.g., how to collect a hammer).
3. **Human Demonstrations:** Trajectories are annotated by human experts. It is necessary for scenarios where humans are indispensable (e.g., mobile apps require logged-in human accounts).

These strategies are quite different from each other and present their own unique advantages in certain environments. We summarize their recommendation levels on 4 dimensions (Cf. Table 3):

- **Average Cost:** The most important dimension. Program-based solvers are most cost-effective, followed by human demonstrations. LMM bootstrapping is currently the most expensive due to proprietary API costs, but this may decrease as open LMMs improve.
- **Adaptability:** It indicates how easy we can implement a strategy to an environment. LMM bootstrapping is highly adaptable with good prompts. Program-based solvers require moderate implementation time. Human demonstrations need training and may face accessibility issues.

Table 3: Recommendation levels for 3 strategies used in curating VAB's agent-tuning trajectory data on different dimensions. (Cf. Section 3.2 for detailed explanation on each dimension)

| Strategy | Avg. Cost | Adaptability | Versatility | Flexibility | Adoption |
|---|---|---|---|---|---|
| Program-based Solvers | ★★★ | ★★ | ★ | ★ | VAB-OmniGibson, VAB-WebArena-Lite |
| LMM Agent Bootstrapping | ★ | ★★★ | ★★ | ★★★ | VAB-Minecraft, VAB-AndroidLab, VAB-CSS |
| Human Demonstrations | ★★ | ★ | ★★★ | ★★ | VAB-AndroidLab |

- **Versatility:** It refers to how versatile tasks a strategy could deal with. Human annotators can handle the widest range of tasks, followed by LMM agents. Program-based solvers are limited to predefined prototypes.
- **Flexibility:** It denotes the trial and error process in the solution trajectories. LMM bootstrapping naturally incorporates trial-and-error processes. Program-based solvers struggle with this, while human annotators are often discouraged from it for quality control reasons.

Considering all mentioned dimensions and their trade-offs, we adopt a hybrid set of strategies for each of the 5 environments in VAB as shown in Table 3. We employ diverse strategies tailored to each domain's unique characteristics. For VAB-OmniGibson, we utilize program-based solvers due to high hardware requirements and need for cost-effective, adaptable solutions. In VAB-Minecraft, we opt for LMM agent bootstrapping to handle inherent randomness and exploration requirements. For VAB-AndroidLab, we primarily rely on human demonstrations, supplemented by LMM agent bootstrapping for offline apps, addressing XML legibility and app-specific login challenges. In VAB-WebArena-Lite, we choose program-based solvers, leveraging mature `Playwright` automation tool and addressing difficulties faced by human annotators with unfamiliar interfaces. For VAB-CSS, we implement LMM agent bootstrapping to accommodate iterative CSS debugging, using `gpt-4o` for initial trajectories with hints to improve success rates. For more details, please refer to Appendix A.4.

# 4 BASELINE EXPERIMENT

## 4.1 SETUP

**Baselines.** We evaluate on both proprietary LMM APIs and selected open LMMs. For proprietary LMMs, we include models from OpenAI GPT (OpenAI, 2024a; 2023; 2024b), Anthropic Claude (Anthropic, 2024), Google Gemini (Reid et al., 2024; Team et al., 2023), and Qwen-VL-Max (Bai et al., 2023). For open LMMs, we select nine state-of-the-art models as representative fine-tuning baselines in VAB: InternVL-2 (Chen et al., 2024), Qwen2-VL (Wang et al., 2024b), GLM-4V (GLM et al., 2024), CogVLM2 (Wang et al., 2023b), CogAgent (Hong et al., 2023), CogVLM (Wang et al., 2023b), LLaVA-NeXT (Liu et al., 2024b), LLaVA-1.5 (Liu et al., 2024a), Qwen-VL (Bai et al., 2023).

**Prompting & Open LMM Training.** We format agent interactions as user-assistant conversations with system prompts containing task descriptions and action spaces. For most environments, only the latest visual input is provided to reduce costs, except for CSS agents which require comparing multiple screenshots. During fine-tuning of open LMMs, we remove few-shot demonstrations, concatenate history with role tokens, and vertically combine multiple images for CSS agents when needed. All models undergo full-parameter fine-tuning for 5k steps with batch size 64, with CSS data duplicated to improve adaptation to the screenshot format. See details in Appendix A.5.

## 4.2 MAIN RESULTS

Table 4 shows the main results on VAB, including both prompting proprietary LMMs and fine-tuned open LMMs. We have several important observations on the status quo of LMM-as-Visual-Foundation-Agent.

**VAB is challenging for existing LMMs.** We observe that existing LMMs face significant challenges when evaluated on VAB. The majority of proprietary LMMs, with mere prompting, achieve an overall success rate above 20%, demonstrating their multimodal understanding and reasoning abilities. The most capable LMM, `gpt-4o`, achieves an overall success rate of 36.2%. However, these performances are still far from satisfactory and not yet qualified for direct deployment. Notably, despite its superiority on existing benchmarks, `claude-3.5-sonnet` still falls significantly

Table 4: Main results on VISUALAGENTBENCH. The metric reported is success rate (SR), which measures the proportion of successful tasks in all evaluated tasks. Open LMMs are evaluated using multitask fine-tuning rather than direct prompting, as they were unable to effectively follow system prompts from VAB in our preliminary trials. For # Params of open LMMs, we report the sizes of their language and vision part respectively.

| Type | Model | #Params | AVG | Embodied | | GUI | | Visual Design |
| | | | | OmniGibson | Minecraft | AndroidLab | WebArena-Lite | CSS |
| --- | --- | --- | --- | --- | --- | --- | --- | --- |
| | gpt-4o-2024-05-13 | N/A | **36.2** | **41.4** | 55.2 | **31.9** | 18.2 | **34.5** |
| | gpt-4-vision-preview | N/A | 31.7 | 36.5 | 47.4 | 26.9 | 18.8 | 29.1 |
| | gpt-4-turbo-0409 | N/A | 29.9 | 26.5 | 50.0 | 26.9 | 18.2 | 27.9 |
| Proprietary | claude-3.5-sonnet | N/A | 26.9 | 24.3 | **56.0** | 31.1 | 7.2 | 15.8 |
| LMMs | claude-3-opus | N/A | 21.9 | 14.9 | 51.7 | 15.1 | 7.9 | 20.0 |
| (Prompting) | gpt-4o-mini-2024-07-18 | N/A | 20.5 | 12.2 | 30.2 | 22.7 | **20.6** | 17.0 |
| | gemini-1.5-pro | N/A | 19.8 | 22.1 | 41.4 | 16.8 | 7.9 | 10.9 |
| | gemini-1.0-pro | N/A | 6.3 | 4.4 | 11.2 | 11.8 | 4.2 | 0.0 |
| | qwen-vl-max | N/A | 2.7 | 0.0 | 6.0 | 2.5 | 3.0 | 1.8 |
| | InternVL-2 (Chen et al., 2024) | 7B + 0.3B | 16.0 | 16.0 | 28.4 | 3.4 | 7.9 | 24.2 |
| | Qwen2-VL (Wang et al., 2024b) | 7B + 0.3B | 15.6 | 13.8 | 24.1 | 5.9 | 6.7 | 27.3 |
| | GLM-4V (GLM et al., 2024) | 9B + 4B | 12.0 | 8.8 | 19.8 | 2.5 | 5.5 | 23.6 |
| Open LMMs | LLaVA-NeXT (Liu et al., 2024b) | 8B + 0.3B | 10.5 | 3.3 | 23.3 | 3.4 | 4.2 | 18.2 |
| (Fine-tuning) | CogVLM2 (Hong et al., 2024) | 8B + 12B | 10.3 | 3.3 | 25.9 | 1.7 | 3.0 | 17.6 |
| | CogAgent (Hong et al., 2023) | 7B + 11B | 8.9 | 6.6 | 20.7 | 2.5 | 0.6 | 13.9 |
| | CogVLM (Wang et al., 2023b) | 7B + 10B | 8.4 | 3.3 | 19.8 | 4.2 | 4.2 | 10.3 |
| | LLaVA-1.5 (Liu et al., 2024a) | 13B + 1B | 7.7 | 1.7 | 25.9 | 0.8 | 2.4 | 7.9 |
| | Qwen-VL (Bai et al., 2023) | 7B + 2B | 5.7 | 1.7 | 18.1 | 1.7 | 2.4 | 4.8 |

behind `gpt-4o`. Additionally, we present the first systematic evaluation of `gpt-4o-mini` on agent tasks, which reveals that its performance is considerably inferior to `gpt-4o` but comparable to `claude-3-opus` and `gemini-1.5-pro`.

**Trajectory SFT can improve LMM agents.** For open LMMs, we find they can rarely follow the system prompt's instruction without fine-tuning in preliminary trials, resulting in 0% success rates. After training on VAB, open LMMs present significant improvements. The strongest one, `InternVL-2`, even outperforms `gemini-1.0-pro` on all evaluated environments and `claude-3-opus` on CSS agent task. These results suggest that learning from trajectories would be a promising direction for us to build visual foundation agents.

**Gaps between top proprietary and open LMMs are huge but likely to be narrowed.** Despite training improvements, the gap between proprietary and tested open LMMs is wider than expected. While many claim performance comparable to `gpt-4-vision-preview` on traditional vision benchmarks like image captioning and VQA, their fundamental ability to function as practical visual foundation agents remains far from comparable, even after fine-tuning on VAB datasets. This demonstrates that VAB could serve as an ideal testbed for benchmarking practical LMM performance. With larger backbone LLMs (insufficiently tested here due to computing resource limitations) and more high-quality trajectory data, open LMMs may eventually match or surpass proprietary ones.

## 5 ANALYSIS

Multimodal agent tasks encompass two critical challenges: *visual grounding* and *planning*. We conduct fine-grained analyses to gain deeper insights into performance in these two aspects and offer valuable perspectives for the future development of visual foundation agents based on LMMs.

### 5.1 VISUAL GROUNDING ANALYSIS

Visual grounding refers to the ability to associate language concepts with content in visual perception (Fukui et al., 2016; Zheng et al., 2024), which is crucial for LMM-as-Visual-Foundation-Agent. We look into 3 typical design choices in VAB related to visual grounding to show its current status and challenges.

**The use of object labels in embodied environment.** Despite the strong image caption and object recognizing ability of LMMs, they seem not to play well in the context of an embodied agent task.

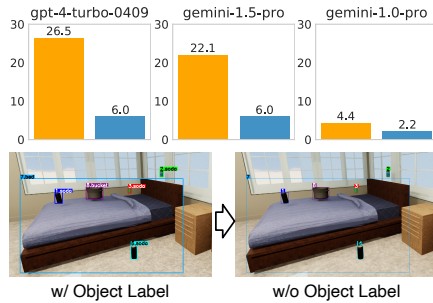

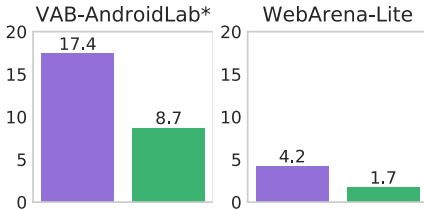

Figure 4: Compare w/ and w/o Object Labels.

Figure 5: Compare SoM and REC in GUI agent tasks, trained on CogVLM2. VAB-AndroidLab* here is an earlier version different from the one in Table 4.

Table 5: LMM performances drop drastically on VAB-CSS when the language description is removed.

|  | gpt-4o-2024-05-13 | gpt-4-turbo-0409 | gpt-4-vision-preview |
|---|---|---|---|
| *w/* NL | 34.5 | 27.9 | 29.1 |
| *w/o* NL | 24.2 ↓10.3% | 1.9 ↓26.1% | 2.4 ↓26.7% |

In VAB-OmniGibson, we compare the LMM-as-Visual-Foundation-Agent performance with and without object labels annotated in the vision input. The result in Figure 4 shows that LMM agents significantly underperform without object labels. It indicates that notwithstanding LMMs' strong performance on downstream benchmarks, they can still struggle in the same task in agent context.

**The use of Set-of-Marks (SoM) in GUI environment.** For GUI tasks, we augment image input with SoM by default as eliciting accurate bounding box coordinates from LMMs is difficult, representing a referring expression comprehension (REC) task (Qiao et al., 2020). Using our training trajectories, we evaluate whether LMMs can perform visual grounding by directly outputting bounding boxes without SoMs. We fine-tuned CogVLM2 with and without SoM, and results in Figure 5 show that it struggles to directly output bounding boxes, highlighting SoM's instrumental role in visual grounding.

**Visual difference grounding.** Our new frontend design task examines visual difference grounding, which unlike traditional single-scene visual grounding, requires the LMM to ground "layout differences" (Cf. Appendix F.5) between two images through comparison. Current VAB-CSS results in Table 4 use a lenient setting where we provide explicit language descriptions of adjustments needed to match the images, rather than requiring direct perception of differences (Cf. Table 5 and Appendix F.2 for examples).

## 5.2 PERFORMANCE ON PLANNING

**The role of thought in ReAct.** ReAct (Yao et al., 2023) is one of the most commonly used frameworks for language agents. The central concept emphasizes the importance of integrating the agent's reasoning and actions by intertwining the output with both thought and action components. However, in our study, we find that the thought step may not be essential. When using gpt-4o and claude-3.5-sonnet as the backbone of the agents, directly outputting an action field can yield comparable or even superior performance compared to using the ReAct framework (see Table 6).

**Recovering from errors during planning.** In real-world applications, agents require the error recovery ability to dynamically adjust their actions and plans based on environmental feedback. To understand it in LMMs, we analyze two representative models: gpt-4o, the most powerful model currently available, and glm-4v, a prominent open LMM. Their performance, illustrated in Figure 6, reveals that gpt-4o exhibits robust error recovery across most tasks, with GUI tasks being an exception due to their often irreversible nature. Importantly, we find that incorporating error recovery scenarios in training data significantly enhances the performance of fine-tuned open LMMs, as observed in VAB-OmniGibson and VAB-CSS (Cf. Appendix B.1 and Appendix F.2).

## 6 RELATED WORK

**LMM-as-Visual-Agent.** In pre-LMM era, most visual agents are built with task specific training (Shridhar et al., 2020a) and reinforcement learning (Kempka et al., 2016). With the rapid development of LMMs (OpenAI, 2024a; Reid et al., 2024; OpenAI, 2023; Bai et al., 2023; Anthropic,

Table 6: ReAct (w/ CoT) has varied (potentially positive or negative) impacts on visual agent tasks.

| Model | Prompting | VAB-Minecraft | VAB-AndroidLab | VAB-CSS |
|---|---|---|---|---|
| `gpt-4o` | *w/* Thought | 55.2 | 30.4 | 34.5 |
| | *w/o* Thought | 48.3 ↓6.9% | 31.9 ↑1.5% | 38.2 ↑3.7% |
| `claude-3.5-sonnet` | *w/* Thought | 56.0 | 29.0 | 15.8 |
| | *w/o* Thought | 55.2 ↓0.8% | 31.1 ↑2.1% | 17.6 ↑1.8% |

Figure 6: Comparison of overall success rates and success rates when incorrect actions are present in trajectories for various tasks.

2024; Team et al., 2023; GLM et al., 2024), the study of LMM-based visual agents begins to thrive. Leveraging the general capabilities of LMMs, these visual agents have the potential to perform complex tasks in various scenarios, including embodied and game tasks (Brohan et al., 2022; Yang et al., 2023a; Driess et al., 2023; Tan et al., 2024), GUI interaction (Zheng et al., 2024; Zhou et al., 2023; Koh et al., 2024; Xie et al., 2024; Kapoor et al., 2024; Yang et al., 2023b), and visual design tasks (Si et al., 2024; Laurençon et al., 2024). However, these complex scenarios pose several challenges for LMM-based visual agents: basic visual understanding and grounding (Zheng et al., 2024; Yue et al., 2023), vision-text information comprehension (Kil et al., 2024), instruction following, and long-term planning ability (Wu et al., 2023; Liu et al., 2023a). Most general-purpose LMMs still lack strong zero-shot capabilities, leading to different application paradigms when deploying LMMs as visual agents. While prompting methods offer great convenience, they may not achieve satisfactory performance in many areas (Zhou et al., 2023; Drouin et al., 2024). Consequently, task-specific training and alignment remain common practices in these applications (Lai et al., 2024). In response, VAB aims to establish a comprehensive benchmark for LMM-based visual agents, covering a wide range of typical applications. In the meantime, VAB seeks to provide an in-depth evaluation of both prompting and training approaches, ultimately fostering the development of LMM visual agents.

**Benchmarking LMM-based visual agents.** With the rapid development of LMM agents and their impressive performance in various scenarios (Xie et al., 2024; Kapoor et al., 2024; Yang et al., 2023b;a; Si et al., 2024; Mu et al., 2024), it has made the evaluation of LMM agent an urgent problem. In the GUI interaction domain, recent works have proposed static datasets (Deng et al., 2024; Rawles et al., 2024; Sun et al., 2022) and interactive environments (Zhou et al., 2023; Koh et al., 2024; Xie et al., 2024) to evaluate LMM agents in different applications, including web (Zhou et al., 2023; Koh et al., 2024; Deng et al., 2024), mobile phone (Rawles et al., 2024; Sun et al., 2022), and desktop (Xie et al., 2024). In the embodied domain, previous works have proposed various game environments (Guss et al., 2019; Fan et al., 2022) and household environments (Li et al., 2023), but few works have explored benchmarking LMM agents on these environments. Most existing benchmarks are designed for relatively narrow domains and lack a comprehensive evaluation across different applications of LMM agents. Additionally, many benchmarks focus solely on the prompting evaluation of LMM agents. VAB aims to provide a training set for open-source foundation LMMs, offering a new perspective on benchmarking these models and advancing their diverse applicabilition.

# 7 CONCLUSION

We present VisualAgentBench (VAB), a comprehensive benchmark for evaluating Large Multimodal Models as visual foundation agents across diverse scenarios. Our testing of 18 LMM models reveals their developing capabilities in this domain. VAB also explores methods for synthesizing visual agent trajectory data, providing insights for future advancements.

ACKNOWLEDGMENT

We would like to thank the anonymous reviewers for their suggestions in refining this work. This work is supported by NSFC 624B2084, 62495063, and 62425601, Daimler Greater China Ltd. and Tsinghua University Joint Institute for Sustainable Mobility, and New Cornerstone Science Foundation through the XPLORER PRIZE.

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

# Part I

# Appendix

## Table of Contents

# A  OVERVIEW

## A.1  DESIGN FEATURES OF VAB

Given that LMMs are still evolving rapidly, we adhere to several principles in our design of VAB to accommodate the current capabilities and limitations of LMMs.

- **Vision-Centric:** VAB agent tasks are designed to primarily rely on visual inputs to solve problems. While additional text inputs could be beneficial, VAB aims to evaluate how LMMs perform when perceiving the environment as humans do in agent tasks. For example, while HTML is shown useful for Web GUI Agent (Zhou et al., 2023; Deng et al., 2024), humans typically browse the internet from screens without reading HTMLs.
- **High-Level Decision Making:** VAB focuses on evaluating LMMs' high-level decision-making abilities. Compared to prior smaller visual-language models that specifically target low-level policies (Lynch & Sermanet, 2020; Brohan et al., 2022; Lifshitz et al., 2024), LMMs excel at high-level planning and interacting (Driess et al., 2023) in text response thanks to their commonsense, knowledge, and flexible instruction following with mere prompting. Therefore, in VAB, we simplify the low-level control by providing convenient action interfaces, and ask tested LMMs to concentrate on delivering high-level decision sequences in text.
- **Interactive Evaluation:** Evaluating LLMs or LMMs on real-world agent tasks is challenging, as task goals can be achieved by various means. As a result, it becomes a mainstream practice to evaluate in an interactive manner (Liu et al., 2023a; Zhou et al., 2023; Jimenez et al., 2023; Xie et al., 2024). VAB also adheres to this principle.
- **Trajectories for Behavior Cloning:** Many previous execution-based agent benchmarks for LLMs and LMMs, despite being realistic and challenging, often fail to provide effective training sets for the community to use for improvement. LLMs and LMMs need behavior cloning training on trajectories for better performance (Nakano et al., 2021; Zeng et al., 2023; Lai et al., 2024). However, creating such datasets consisting of valid instructions, trajectories, and reward functions is costly and requires annotators' good understanding of the environment. In response to the challenge, for each VAB environment we endeavor to deliver instructions created with a hybrid set of strategies (Cf. Section 3.2). Experiments show that our constructed training sets can effectively improve the performance of open LMMs on VAB.

Note that as the field advances, some of the above principles may become obsolete and irrelevant. We will continuously update VAB to accommodate the progress of LMMs.

## A.2  DETAILS ON USE OF VISUAL INFORMATION IN EACH ENVIRONMENT

Vision-centric design is a crucial feature for VAB with regard to planning and grounding. Here we will give a brief look at the specific use of vision from both perspectives.

- **VAB-OmniGibson**: The agent perceives the simulated environment through egocentric vision. It must interpret visual information to identify the affordances of objects within its view and understand their spatial relationships (e.g., whether an object is reachable).
- **VAB-Minecraft**: Similar to VAB-OmniGibson, the agent must interpret the current scene in the game environment to determine its next action. In addition, VAB-Minecraft includes moving elements such as animals and monsters, which places greater demands on the agent's visual understanding to complete tasks.
- **VAB-AndroidLab**: The agent operates on the graphical interface of an Android system to complete tasks without access to system APIs and relies solely on visual inputs. At each step, it analyzes the current screenshot to predict an action—identifying interactive elements such as app icons or buttons based solely on the screenshot. Executing the action generates a new screenshot for the next iteration.
- **WebArena-Lite**: Similar to VAB-AndroidLab, WebArena-Lite is also a GUI-based environment. In the original paper of WebArena (Zhou et al., 2023), they use HTML/Accessibility Tree as the input, whereas in VAB, we focus on visual inputs and mainly focus on screenshots to predict actions at each step.

- **VAB-CSS**: The agent must carefully adjust the CSS style file until the rendering matches the given screenshot. It needs to perceive fine-grained visual differences between two screenshots, which can often be minimal. This makes VAB-CSS a highly vision-intensive task.

## A.3 DETAILS ON TASK INSTANCE COLLECTION

For VAB-OmniGibson, a prototype is a general household activity, such as recycling office papers. We source these prototypes either by sampling from Behavior-1K or by annotating them ourselves. Instantiating a prototype involves grounding it in a specific scene (e.g., specific rooms with office papers and recycling bins) generated in OmniGibson. To increase task diversity, we instantiate each prototype with multiple random scenes and various initializations of object positions in the room. In total, we collect 992 instances using 89 prototypes. We sample 181 out of them as our test set.

For VAB-Minecraft, we target high-level task prototypes related to object collecting and then instantiate them with game configurations using different world seeds or spawn points. We manually check to ensure that each high-level goal is achievable within its configuration. In total, we collect 628 task instances using high-quality prototypes defined by us, with 116 instances designated as the test set. Additionally, we sample 132 task prototypes from JARVIS-1, resulting in 596 task instances that could be leveraged to collect our training trajectories later.

For VAB-AndroidLab, we first select 8 typical Android applications, from system services to third-party applications (e.g., Maps, Music, etc.) that could be evaluated offline. We come up with 119 test instructions for them and prepare valid groundings in the AVD snapshot (e.g., an MP3 file to play in the Music APP). For the training task construction, we filter 18 commonly used APPs and summarize their major functions to around 70 task prototypes.

For WebArena-Lite, we filtered and cleaned 165 test samples from the original WebArena dataset and collected new task instances for web applications to use in training trajectory collection. Specifically, we summarize each website's basic functions and valid items for synthetic queries, created 40 task prototypes, and fill them with valid and invalid items (e.g., `product categories`, `prices`) to generate specific instructions, resulting in 1,186 training task instances.

For VAB-CSS, a task prototype simply corresponds to one possible corruption of a CSS rule such as adding or altering a CSS property. To instantiate a task for a specific website, we randomly select a corruption that results in noticeable visual changes, determined by an SSIM (Wang et al., 2004) score below 0.8.[3] In addition, we manually annotate each instance with a natural language description of the difference between the two images as an additional clue to the agent. In total, we collect 1,210 instances and use 165 to form the test set.

## A.4 DETAILS ON TRAINING TRAJECTORY COLLECTION

For VAB-OmniGibson, we adopt the program-based solvers focusing on the cost and adaptability. OmniGibson has no friendly interface for humans to operate on, and requires high-end laptops with GPUs supporting ray tracing and large main memory ($> 10$ GB) to run. Thus it is unlikely for us to find a large number of qualified annotators to label for OmniGibson. LMM agent bootstrapping is fine but uneconomical, as the task usually takes more steps than others (i.e., up to 100). Program-based solvers, instead, are suitable for collecting massive high-quality trajectories in OmniGibson.

For VAB-Minecraft, we adopt LMM agent bootstrapping considering adaptability. Minecraft requires some flexible explorations (as environments are generated randomly), which is beyond the scope of program-based solvers. Humans need to be well-trained for some time on playing Minecraft before becoming qualified annotators. Since previous work has explored the usage of memory augmentation (Wang et al., 2023c) for improving LMM agents in Minecraft, it becomes practical to leverage the bootstrapping strategy by LMM APIs such as `gpt-4o` for creating training trajectories.

For VAB-AndroidLab, we primarily adopt human demonstrations, accompanied with some LMM Agent Bootstrapping considering the versatility and flexibility. As android XMLs are less legible and operable than HTMLs on web with existing automation tools, program-based solvers are not applicable. Additionally, for many apps require login and internet connection, human demonstration

---

[3]This is an empirical choice based on our own experience.

is the best solution. LMM agent bootstrapping is employed in some offline APPs such as system settings to enhance trajectory flexibility.

For VAB-WebArena-Lite, we adopt program-based solvers due to cost and adaptability. On the one hand, there have been a mature web automation tool `Playwright` that supports Python. On the other hand, although WebArena (Zhou et al., 2023) is adopting some mirror websites for their real-world counterparts, their interfaces could be vastly different (e.g., OpenStreetMap in WebArena vs. Google Maps in real-world). Consequently, human annotators struggle to label demonstrations on these websites efficiently in our preliminary trials. For LMM agents, they tend to perform too poorly under mere prompting on WebArena (with success rate less than 20%) for efficient trajectory construction.

For VAB-CSS, we adopt LMM agent bootstrapping, mostly owing to concerns on flexibility. A critical challenge for the agent in debugging CSS styles is to iteratively adjust the CSS rules through a trial and error process, which can be flexibly achieved using the LMM agent bootstrapping scheme. In particular, we first use `gpt-4o` to collect trajectories that finally resolve the CSS issue. However, `gpt-4o` can only achieve a success rate lower than 40%. To collect additional trajectories, we hint the agent with the target CSS rule to edit, after 5 steps of trials, on tasks where the agent initially fails.

### A.5 DETAILS ON PROMPTING AND TRAINING OPEN LMMS IN EXPERIMENTS

**Prompting.** We format LMM-as-Visual-Foundation-Agent as two roles (i.e., `user` and `assistant`) interacting in multiple rounds. The task description, action spaces, few-shot demonstrations, and important notices for each environment are formatted as the `system` prompt at the beginning of the conversation. Task instruction is given in the first `user` round. Environmental observations and feedback are passed via `user` in later rounds. Considering current LMM APIs' poorer support of multi-image and outrageous cost when interaction rounds soar up, in Embodied and GUI agents we only offer the vision input of the latest `user` round (following (Koh et al., 2024)) while reserving history text contents. An exception is the CSS agent in Visual Design. In this case, comparing differences in visual inputs is essential, and the interaction rounds are typically fewer than 10. Therefore, we retain all image inputs in the conversation history for this task.

**Training for Open LMMs.** We generally follow the prompting format of proprietary LMM APIs in each environment to organize our training trajectories, and make several minor modifications. In the system prompt we remove the few-shot demonstrations as we would fine-tune models. In addition, during fine-tuning, since open LMMs perform poorly on multi-image input (especially for CogVLM and CogAgent, whose expert architecture disallows simple implementation of multi-image input), we only use the vision input of the latest `user` turn, and concatenate histories together using role tokens (i.e., "`<|user|>`") and linebreaks. For CSS agent where multi-image input is necessary, we concatenate history images vertically into one as the input. To benchmark the potential of LMMs to serve as visual foundation agents, we conduct multitask fine-tuning over the dataset aggregation of all environments. To optimize performance, all LMMs undergo full-parameter fine-tuning, with a batch size of 64 and 5k training steps. Other hyperparameters are configured using the default ones provided by the model's original repository or the third-party's integrated training framework. For data composition, we uniformly combine all training samples except for VAB-CSS, which we duplicate an additional 2 times as the preliminary experiments show that the task requires more extensive training for open LMMs to adapt to the screenshot concatenation format.

### A.6 ANALYSIS ON AGENT ERROR MODES

Here we provide some statistics about major types of errors we observe, by sampling around 20 error traces in each environment for `gpt-4o` and `internvl-2` (in Table 7 and Table 8).

- **Visual Grounding Error:** Wrong detection or recognition of objects/elements in the visual observation.
- **Invalid Action:** Outputting wrong formats of actions.
- **Loop:** Agent repeats generating the same actions without quitting.
- **Task Limit Exceed:** The agent does not accomplish the goal within reasonable maximum steps.

- **Hallucinated Task Completion:** Agent makes wrong judgment on whether it has accomplished the task.

Table 7: Statistics on `gpt-4o` agent error modes.

| Tasks | Visual Grounding Error | Invalid Action | Loop | Task Limit Exceed | Hallucinated Task Completion |
|---|---|---|---|---|---|
| VAB-OmniGibson | 0.30 | 0.04 | 0.17 | 0.17 | 0.30 |
| VAB-Minecraft | N/A | 0.00 | 0.24 | 0.76 | 0.00 |
| WebArena-Lite | 0.15 | 0.10 | 0.40 | 0.05 | 0.30 |
| VAB-AndroidLab | 0.10 | 0.00 | 0.65 | 0.15 | 0.10 |
| VAB-CSS | N/A | 0.00 | 0.05 | 0.55 | 0.40 |

Table 8: Statistics on `internvl-2` agent error modes.

| Tasks | Visual Grounding Error | Invalid Action | Loop | Task Limit Exceed | Hallucinated Task Completion |
|---|---|---|---|---|---|
| VAB-OmniGibson | 0.00 | 0.00 | 0.25 | 0.50 | 0.25 |
| VAB-Minecraft | N/A | 0.00 | 0.76 | 0.24 | 0.00 |
| WebArena-Lite | 0.05 | 0.00 | 0.40 | 0.10 | 0.45 |
| VAB-AndroidLab | 0.05 | 0.05 | 0.60 | 0.05 | 0.25 |
| VAB-CSS | N/A | 0.00 | 0.45 | 0.30 | 0.25 |

## A.7 PROXY METRICS FOR EVALUATING PROGRESS OF TASK COMPLETION

For embodied and visual design problems in VAB (including 3 environments: VAB-OmniGibson, VAB-Minecraft, and VAB-CSS), we are setting up proxy metrics for evaluating progress of task completion (in Table 9).

- **VAB-OmniGibson:** To complete a task, the LMM agent must achieve multiple subgoals (e.g., opening a specific door). Upon task termination, we compute the percentage of successfully achieved subgoals to provide an intermediate score.
- **VAB-Minecraft:** To acquire the goal item, the LMM agent must gather a series of items as ingredients. Consequently, we allocate intermediate scores to the agent as they collect these ingredients.
- **VAB-CSS:** To fix the CSS style to match the target screenshot, we can use screenshot similarity as a proxy metric for measuring progress of completion.

Table 9: Proxy Metrics for Evaluating Progress

| Models | VAB-OmniGibson | VAB-Minecraft | VAB-CSS |
|---|---|---|---|
| gpt-4o-2024-05-13 | 62.6 | 61.9 | 46.7 |
| gpt-4-vision-preview | 58.4 | 55.9 | 38.2 |
| gpt-4-turbo-0409 | 50.6 | 59.7 | 37.6 |
| claude-3.5-sonnet-20240620 | 59.7 | 63.6 | 20.6 |
| claude-3-opus | 33.8 | 60.1 | 24.8 |
| gpt-4o-mini-2024-07-18 | 41.5 | 37.3 | 23.6 |
| gemini-1.5-pro | 46.3 | 49.4 | 13.9 |
| gemini-1.0-pro | 12.7 | 15.3 | 0.0 |
| qwen-vl-max | 1.3 | 9.0 | 3.0 |
| InternVL-2 | 41.6 | 35.5 | 32.7 |
| Qwen2-VL | 24.1 | 29.8 | 34.5 |
| GLM-4V | 32.2 | 29.4 | 29.7 |
| LLaVA-NeXT | 17.3 | 30.5 | 23.6 |
| CogVLM2 | 26.9 | 32.6 | 20.6 |
| CogAgent | 33.0 | 32.2 | 17.0 |
| CogVLM | 17.1 | 32.3 | 11.5 |
| LLaVA-1.5 | 14.9 | 33.5 | 9.1 |
| Qwen-VL | 18.8 | 26.5 | 4.8 |

## A.8 Analysis on Error Recovery Behaviors

we analyze the average steps needed for agents to recover from error to the correct directions (as below in Table 10, based on proprietary `gpt-4o` and opened `glm-4v`, only for those finally successful tasks). We find that GUI tasks usually require more steps to recover, as the action spaces for them are very large (e.g., any clickable elements on the web pages). And fine-tuned `glm-4v` has shorter mean error recovery steps compared to `gpt-4o`, probably because it can only recover from simpler errors.

Table 10: Statistics on average steps to recover from error.

| Models | VAB-OmniGibson | VAB-Minecraft | Webarena-Lite | VAB-AndroidLab | VAB-CSS |
|---|---|---|---|---|---|
| `gpt-4o-2024-05-13` | 2.5 | 3.3 | 6.0 | 8.3 | 2.6 |
| `glm-4v` | 2.3 | 2.3 | 4.0 | N/A | 2.2 |

## B  VAB-OmniGibson

In this section, we provide additional details about VAB-OmniGibson that are not covered in the main paper due to space limitations.

### B.1 Detailed Description

Current household datasets or benchmarks are not originally designed for LMMs, making them less suitable for evaluating today's LMMs. Behavior-1K (Li et al., 2023) offers an action space focused on low-level physical control over the robot (e.g., joint angles), while Alfred (Shridhar et al., 2020a) requires actions to output masks on images, which may not be practical for most LMMs. The ThreeDWorld Transport Challenge (Gan et al., 2022) provides high-level action APIs, but the simulator environment is less realistic and the tasks may not fully challenge LMMs. The recent work Octopus (Yang et al., 2023a) sets up household tasks for LMMs in the OmniGibson simulator. However, in this setting, vision input is less critical as the observed objects are also listed in text input for LMMs.

In order to set up a realistic and challenging benchmark for testing LMMs' embodied planning ability, we select the recent household simulator OmniGibson (Li et al., 2023) as the interactive environment, and build a pipeline for LMM to serve as a high-level planner on everyday household activities. An example of the task is shown in Fig. 7: The ego-centric image with annotated bounding boxes, high-level activity instruction and environment feedback are fed into the LMM, and it is tasked with reasoning over the current progress to decide on the next low-level action. It must interact with objects using the corresponding tags attached to the bounding boxes.

**Test Set.** We select 45 activity instances from Behavior-1K, and manually adapt some of them to ensure these activities are solvable within our provided action space and suitable for evaluating current LMMs' embodied planning ability. We instantiate each activity in several scenes, resulting in a total of 181 test task instances. All the activity instructions are manually annotated by us.

**Training Set.** We provide a set of successful trajectories using both program-based solving and LMM bootstrapping. We newly design 47 activities, each instantiated in several different scenes with various initializations of object positions, resulting in a total of 901 task instances. To solve these tasks, we develop a program-based solver that decomposes the long-horizon activities into subtasks and solves them sequentially. Running the program-based solver on the 901 training task instances yields 785 successful trajectories. Then we manually add a type of error recovery process (agent fails to place an object into a closed container, and then opens the container) into these trajectories, aiming to enhance LMMs' capability to rectify errors. Additionally, we select 464 training instances and utilize `gpt-4-vision-preview` to bootstrap 87 successful trajectories, resulting in a total of 872 training trajectories.

**Metrics.** We adopt task success rate as the metric of VAB-OmniGibson. In Behavior-1K, each activity is defined in the form of BEHAVIOR Domain Definition Language (BDDL) (Srivastava et al., 2022), which describes the concrete initial and goal conditions of a specific activity. Only when all the goal conditions are met within the limit of 100 turns, the task is judged as successfully completed.

## B.2 ACTIONS

In VAB-OmniGibson, we provide the LMM agent with 20 low-level actions to interact with objects and navigate the household environment. The actions marked with an asterisk (*) are adapted from OmniGibson, while the others are newly defined and implemented by us. With these provided actions, the LMM agent is possible to solve all the testing instances.

- **grasp**: Grasp a specific object into the robot's hand.
- **move**: Move towards a specific object.
- **move_to_room**: Move to a specific room in the house.
- **turn_left**: Turn the robot left 90 degrees.
- **turn_right**: Turn the robot right 90 degrees.
- **raise_camera**: Raise the camera of the robot to see higher objects.
- **lower_camera**: Lower the camera of the robot to see lower objects.
- **put_inside**: Place the object from the robot's hand inside another object.
- **put_on_top**: Place the object from the robot's hand on top of another object.
- **put_under**: Place the object from the robot's hand under another object.
- **put_next_to**: Place the object from the robot's hand next to another object.
- **get_fridge_view**: Obtain the view inside a nearby fridge.
- **cook***: Cook a specific object.
- **burn***: Burn a specific object.
- **freeze***: Freeze a specific object.
- **heat***: Heat a specific object.
- **open***: Open a specific object.
- **close***: Close a specific object.
- **toggle_on***: Turn on a specific object.
- **toggle_off***: Turn off a specific object.

## B.3 PROGRAM-BASED SOLVER FOR TRAINING TRAJECTORY COLLECTION

**BDDL task goals.** Among activities of VAB-OmniGibson, each of the BDDL task goal can be decomposed into a sequence of subgoals (e.g., a specific door should be open, or a specific bottle should be on a specific countertop). All subgoals can be categorized into 2 types: identifying the state of a specific object, or the positional relationship between two objects.

**Method of program-based solver.** To achieve the BDDL task goal of a VAB-OmniGibson activity, the program-based solver need to sequentially fulfill all the subgoals. For the first type of subgoal, the program-based solver can navigate (`move_to_room`, `move`, `turn_left`, `turn_right`, `raise_camera`, `lower_camera`, `get_fridge_view`) to find the specific object; and then move towards it (`move`) and change its state (`cook`, `burn`, `freeze`, `heat`, `open`, `close`, `toggle_on`, `toggle_off`). For positional relationships, the solver should find and approach an object, grasp it (`grasp`), move to the other object, and finally complete the subgoal with `put_inside`, `put_on_top`, `put_under` or `put_next_to`.

## B.4 PROMPT EXAMPLE

The system message that describes the detailed task information to the agent is shown as follows:

```
# Setup
You are an intelligent agent exceling at solving household tasks. You are
    in a household environment given a task to finish.
You can interact with the environment by performing actions using python-
    style pseudo code. For each turn, please call exactly one predefined
    action.

# Valid Actions
## Predefined Action List:
```
```

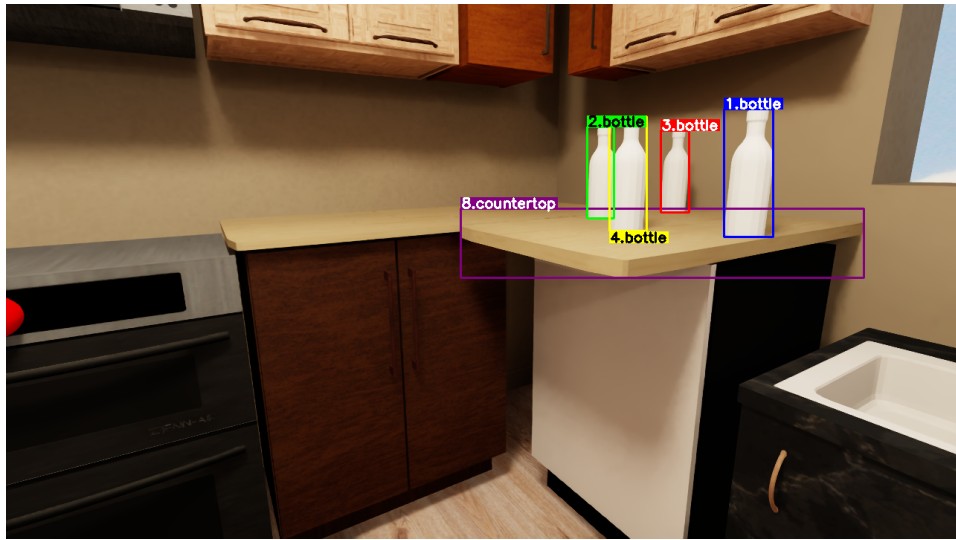

Figure 7: This is an example task of VAB-OmniGibson. The task asks the agent to bring all 4 bottles from the countertop into the fridge. The agent should grasp one bottle, navigate to find the fridge, open the fridge, put the grasped bottle into the fridge, and then repeat this process for the remaining bottles.

```
def grasp(obj):
    '''Grasp the object in your hand.
    Args:
        :param obj: the digital identifier of the object to grasp.
    Returns:
        A string message of the environment feedback.
    '''
def move(obj):
    '''Move yourself towards the object.
    Args:
        :param obj: the digital identifier of the object to move towards.
    Returns:
        A string message of the environment feedback.
    '''
def move_to_room(room):
    '''Move yourself to a random position in the room.
    Args:
        :param room: the name of the room to move to.
    Returns:
        A string message of the environment feedback.
    '''
def turn_left():
    '''Turn the robot left 90 degrees.
    Returns:
        A string message of the environment feedback.
    '''
def turn_right():
    '''Turn the robot right 90 degrees.
    Returns:
        A string message of the environment feedback.
    '''
def raise_camera():
    '''Raise the camera to see objects that are higher.
    Returns:
        A string message of the environment feedback.
    '''
def lower_camera():
```

```
        '''Lower the camera to see objects that are lower.
        Returns:
            A string message of the environment feedback.
        '''
def put_inside(obj1, obj2):
        '''Put obj1 within your hand inside obj2. If obj2 is openable, make
            sure it is open before putting obj1 inside.
        Args:
            :param obj1: the digital identifier of the object to put inside.
            :param obj2: the digital identifier of the object to put inside
                of.
        Returns:
            A string message of the environment feedback.
        '''
def put_on_top(obj1, obj2):
        '''Put obj1 within your hand to the top of obj2.
        Args:
            :param obj1: the digital identifier of the object to put on top.
            :param obj2: the digital identifier of the object to put on top
                of.
        Returns:
            A string message of the environment feedback.
        '''
def put_under(obj1, obj2):
        '''Put obj1 within your hand to the bottom of obj2.
        Args:
            :param obj1: the digital identifier of the object in your hand.
            :param obj2: the digital identifier of the object to put obj1
                under.
        Returns:
            A string message of the environment feedback.
        '''
def put_next_to(obj1, obj2):
        '''Put obj1 within your hand next to obj2.
        Args:
            :param obj1: the digital identifier of the object in your hand.
            :param obj2: the digital identifier of the object to put obj1
                next to.
        Returns:
            A string message of the environment feedback.
        '''
def get_fridge_view():
        '''Get the image captured by a camera in the fridge. This function is
            only valid when you are near a fridge and the fridge is open.
        Returns:
            A string message of the environment feedback.
        '''
def cook(obj):
        '''Cook the object.
        Args:
            :param obj: the digital identifier of the object to cook.
        Returns:
            A string message of the environment feedback.
        '''
def burn(obj):
        '''Burn the object.
        Args:
            :param obj: the digital identifier of the object to burn.
        Returns:
            A string message of the environment feedback.
        '''
def freeze(obj):
        '''Freeze the object.
        Args:
            :param obj: the digital identifier of the object to freeze.
```

```
        Returns:
            A string message of the environment feedback.
        '''
def heat(obj):
    '''Heat the object.
    Args:
        :param obj: the digital identifier of the object to heat.
    Returns:
        A string message of the environment feedback.
    '''
def open(obj):
    '''Open the object.
    Args:
        :param obj: the digital identifier of the object to open.
    Returns:
        A string message of the environment feedback.
    '''
def close(obj):
    '''Close the object.
    Args:
        :param obj: the digital identifier of the object to close.
    Returns:
        A string message of the environment feedback.
    '''
def toggle_on(obj):
    '''Toggle on the object.
    Args:
        :param obj: the digital identifier of the object to toggle on.
    Returns:
        A string message of the environment feedback.
    '''
def toggle_off(obj):
    '''Toggle off the object.
    Args:
        :param obj: the digital identifier of the object to toggle off.
    Returns:
        A string message of the environment feedback.
    '''
def done():
    '''Call this function if you think the task is completed. Note that
        you have no chance to take any actions after calling this
        function.
    Returns:
        None. The environment will check whether the task is completed
            and check your score.
    '''
```
## Reminder
1. You can only hold one object at a time.
2. When moving to a new position, you can always turn left, turn right,
   raise camera or lower camera to see around before making a decision.
3. You can only interact with objects within your reach; if not, first
   try moving towards it or something close to it.
4. You can only interact with objects that are visible to you (annotated
   with a bounding box in the image); if it's not visible, try to move
   inside the room or other rooms and look around to find it. You can
   open refrigerators or other enclosures to see inside them.
5. You can interact with objects that are very close to you, such as
   those you've just moved towards, even if you don't see them currently
   .
6. When you are out of the room and see nothing useful, try moving to a
   room.
7. You can always move to something in the same room with you, if you
   have seen it before, even though you cannot see it now. So when you
   are in a new room, try to move around and see around to record more

```
        objects in your observation so that you can move to them flexibly
        afterwards.
8. Don't repeat the failed action in the next round. Try to understand
    what went wrong and make a different decision.
9. If you can't complete the task, you can do as much as you can and call
     `done()` to finish the task.

# Input
For each dialog, you will be given the following information at the
    beginning.
1. Task Goal: The task is finished only when these conditions are met.
2. Reachable Rooms: Rooms you can move to. Please refer to them with
    their names provided here.
For each turn, you will be given the following information.
1. Action Feedback: Environment feedback of the last action.
2. At Hand Object: The object you are currently holding.
3. Current Room: The room you are currently in.
4. Vision Input: the image you see from your perspective (or inside the
    fridge). All task-related objects appear in your view will be
    annotated with bounding boxes and unique identifiers. Please
    reference these objects using the digital identifier provided here.
    Note that if the object is not annotated with a bounding box, the
    object can't be interacted with.

# Output
Now, given these information, you need to think and call the action
    needed to proceed with the task. Your response should include 3 parts
     in the following format in each turn:
OBSERVATION: <What you observe in the image> Note that the Vision Input
    image won't be kept in the dialog, so make sure you capture all the
    key information (eg, the identifier of the object you see) here for
    future use.
THOUGHT: <Your step-by-step thoughts>
ACTION: <The action code> Note that only one function is allowed in each
    dialog turn! Only one line of code is allowed in each dialog turn! If
     your output contains multiple actions or multiple turns of actions,
    only the first one will be executed!
```

Here is a concrete example of the task input shown in Fig. 7, where the image is enclosed within "{{}}":

```
Your task is: There are 4 beer bottles on a countertop in the kitchen.
    Please put all of them into the fridge.
The reachable rooms during the task are: corridor_0, dining_room_0,
    kitchen_0, living_room_0, pantry_room_0, storage_room_0.
Action Feedback: None actions before.
At Hand Object: None.
Current Room: kitchen_0.
Vision Input: {{Image}}
```

## C  VAB-MINECRAFT

In this section, we provide additional details about VAB-Minecraft that are not covered in the main paper due to space limitations.

The game Minecraft has become a popular open-world environment for developing generalist embodied agents (Fan et al., 2022; Lifshitz et al., 2024) due to its diverse tasks (e.g., survival, harvest, craft, combat, and creative tasks), varied environments, and interactive mobs, necessitating generalized agent abilities. Recent pioneering works (Zhu et al., 2023; Wang et al., 2023a;c) have integrated LLMs into embodied agents to tackle Minecraft tasks. However, these efforts did not focus on a standardized pipeline for evaluating LMMs' planning abilities. So we adapt the JARVIS-1 (Wang et al., 2023c) pipeline to assess LMMs' high-level action planning abilities in item-obtaining tasks.

## C.1    DETAILED DESCRIPTION

In VAB-Minecraft, we adapt the action space of JARVIS-1 (Wang et al., 2023c) to develop a pipeline for LMM, enabling it to function as a high-level embodied planner. Additionally, we also use item-obtaining tasks to benchmark LMMs' high-level embodied planning abilities. These tasks are comprehensive, requiring task analysis and decomposition, as well as ingredient collection. Each aspect respectively challenges an LMM agent's planning skills and its ability to interact with the environment.

**Test Set.** We manually annotate 116 different tasks, each with a specific target item and a corresponding initial configuration to ensure the task is solvable. For example, Fig. 8 illustrates the VAB-Minecraft task of obtaining a cake, where we have set up the initial configuration of necessary surrounding resources and inventory items. These 116 test tasks span the Minecraft tech tree, covering items across 6 material levels (wood, stone, iron, gold, diamond and netherite) and involving a diverse range of raw ingredients from various resources: 11 types of plants, 4 types of animals, and 6 types of hostile mobs. This diversity greatly challenges the agent's ability to interact with the environment.

**Training Set.** Training trajectories are collected using bootstrapping from two sources: pure `gpt-4-turbo` bootstrapping on newly designed tasks, and `gpt-4o` bootstrapping with JARVIS-1 memory on tasks from JARVIS-1. For the first type, we design 40 new tasks instantiated in different world seeds or spawn points, resulting in 512 task instances, and `gpt-4-turbo` bootstraps 176 successful trajectories. For the second type, we use 132 tasks from JARVIS-1, set up in 596 task instances, and run with memory using `gpt-4o`, resulting in 206 successful trajectories. In total, we gain 382 successful trajectories.

**Metrics.** We adopt success rate as the evaluation metric in VAB-Minecraft. For a specific item-obtaining task, if the agent can obtain the specific item within the limitation of 100 rounds, the task is regarded as successfully completed.

## C.2    ACTIONS

In VAB-Minecraft, we provide 6 types of actions for the LMM agent. 4 actions, marked with an asterisk (*), are adapted from the JARVIS-1 pipeline (Wang et al., 2023c), while the remaining 2 are newly implemented by us to enhance the LMM agent's capability to solve a wider range of tasks.

- **craft***: Utilize the inventory or crafting table to craft a specific item.
- **smelt***: Utilize a furnace to smelt a specific item.
- **equip***: Equip a specific item in the agent's hand.
- **teleport_to_spawn**: Teleport the agent back to the spawn point. As we will prepare necessary ingredients around the agent's spawn point, this action enables the agent to conveniently collect these ingredients. This function is also helpful if the agent stuck somewhere (e.g., underground).
- **look_up**: Look up the crafting/smelting information about a specific item. This reference guides the agent to make a plan on how to accomplish the task.
- **execute***: Use natural language prompt to instruct a low-level minecraft planner, Steve-1 (Lifshitz et al., 2024). With proper prompting, it can solve most basic tasks, like mining common blocks, collecting plants, interacting with animals and hostile mobs, and navigating between different biomes.

## C.3    PROMPT EXAMPLE

The system message that describes the detailed task information to the agent is shown as follows:

```
# Setup
You are a skilled Minecraft player. You are born in the survival mode and
    asked to obtain a specific item.
You can interact with the game environment by outputing actions using
    python-style pseudo code. For each turn, please call exactly one
    predefined function.

# Valid Actions
## Predefined Function List:
```

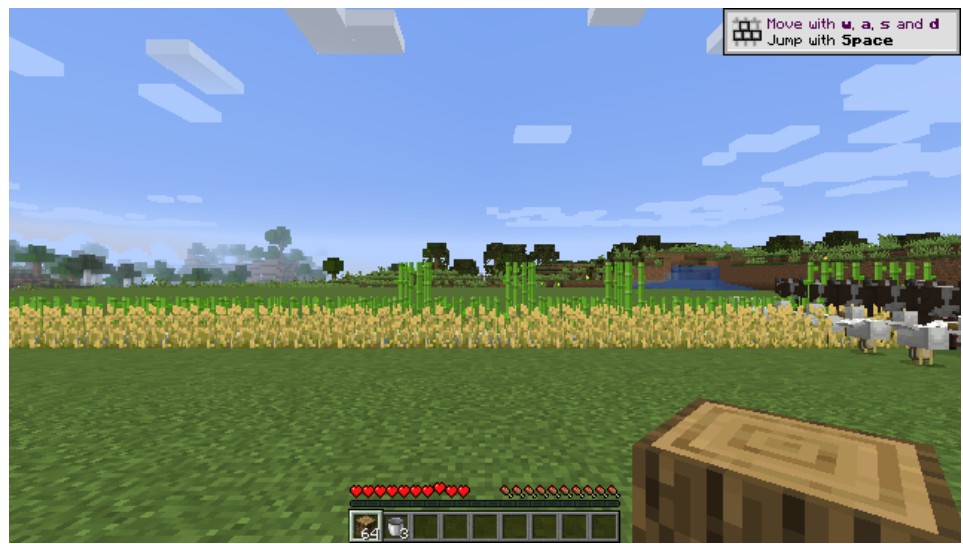

Figure 8: This is an example of VAB-Minecraft task. This task asks the agent to obtain a cake in the inventory. Initially, we provide 3 buckets and 64 logs in the inventory. Additionally, we grow mature wheat and sugar cane in front of the agent and spawn a few chickens and cows around it, ensuring that the agent can conveniently find the necessary ingredients.

```
def craft(item: str, num: int = 1):
    '''Craft specified number of items. Please ensure that you get enough
        ingredients and a craft_table in your inventory.
    Args:
        obj: the name of the item to craft.
        num: the number of items to craft. Default is 1.
    Returns:
        A string message about whether the crafting is successful.
    Examples:
        >>> craft("wooden_pickaxe")
        Successfully crafted 1 wooden_pickaxe.
        >>> craft("bookshelf", 2)
        Not enough materials for 2 bookshelf.   # You don't have 12
            planks and 6 books in your inventory.
    '''

def smelt(item: str, num: int = 1):
    '''Smelt specified number of items. Please ensure that you get enough
        fuels, ingredients, a furnace and a **wooden_pickaxe** in your
        inventory.
    Args:
        obj: the name of the item to smelt.
        num: the number of items to smelt. Default is 1.
    Returns:
        A string message about whether the smelting is successful.
    Examples:
        >>> smelt("iron_ingot", 2)
        Successfully smelted 2 iron_ingot.
        >>> smelt("glass")
        Not enough fuels.  # You don't have enough coals, logs or planks
            as fuel.
    '''

def equip(item: str):
    '''Select an item from your inventory to your hand. Note that if you
        want to use some item, you must equip it first!
```

```
    Args:
        item: the name of the item to equip.
    Returns:
        A string message about whether the equipping is successful.
    Examples:
        >>> equip("diamond_sword")
        Successfully equipped diamond_sword.
        >>> equip("diamond_axe")
        Can not find diamond_axe in inventory.  # You must have the item
            in your inventory before equipping it.
    '''

def teleport_to_spawn():
    '''teleport yourself to the spawn position.
    Args:
        None.
    Returns:
        A string message about whether the teleportation is successful.
    Examples:
        >>> teleport_to_spawn()
        Successfully teleported.

def look_up(item: str):
    '''Look up the information about crafting the item.
    Args:
        item: the name of the item/tag to look up.
    Returns:
        A string message about the information of the item. Note that if
            the argument is a tag, information about all possible items
            will be returned.
    Examples:
        >>> look_up("iron_pickaxe")
        iron_pickaxe: Crafting iron_pickaxe needs 2 stick, 3 iron_ingot.
            Every time you craft iron_pickaxe with the ingredients above,
             you will get 1 iron_pickaxe.
        >>> look_up("stone_tool_materials")
        stone_tool_materials is a tag. Following items belong to this tag
            : cobblestone, blackstone.
        cobblestone: It is a raw item you can mine from the environment.
        blackstone: It is a raw item you can mine from the environment.
    '''

def execute(prompt: str, goal_item: Optional[str] = None, num: Optional[
    int] = None)
    '''Instruct a lower-level executor model to perform some simple tasks
        , like mine something, collect something, move to some place.
    Args:
        prompt: the prompt to instruct the lower-level executor model. It
             should be a simple **verb-object phrase**.
        goal_item (optional): the name of the item to obtain during the
            execution. If the item is obtained, the executor model will
            stop.
        num (optional): the number of items to obtain.
    Returns:
        A string message about the execution.
    Negative Examples: # examples that may cause failure
        Your Inventory: Now your inventory has 1 stone_pickaxe, 2 stick.
        Equipped Item: Now you hold the stone_pickaxe in your hand.
        >>> execute("break iron_ore blocks", "iron_ore", 64)
        The executor has reached the maximum number of steps for this
            turn without completing your subgoal. # Each turn is limited
            in time, 64 iron_ore is too much for one turn.

        Your Inventory: Now your inventory has 1 wooden_axe, 12 logs, 4
            stick, 1 seed, 1 iron_pickaxe.
```

```
        Equipped Item: Now you hold the wooden_axe in your hand.
        >>> execute("find and mine diamond", "diamond_ore", 1)
        The executor has reached the maximum number of steps for this
            turn without completing your subgoal. # You are not holding
            the right tool for mining diamonds. You should equip the
            iron_pickaxe first.

        Your Inventory: Now your inventory has 64 dirt.
        Equipped Item: Now you hold nothing in your hand.
        >>> execute("climb on a tree")
        The executor has attempted to execute the action according to
            your prompt. You should check whether your intention has been
             fulfilled. # The executor can't plan for complex tasks; you
            have to break down complex tasks into simple ones. For
            example, break down the task of `climb on a tree` into `find
            a tree`, `use dirt blocks to elevate`, and `jump on the tree
            `.

        Your Inventory: Now your inventory has nothing.
        Equipped Item: Now you hold nothing in your hand.
        >>> execute("dig a hole and jump in")
        Error: No complex sentences allowed. Keep the prompt a simple **
            verb-object phrases**. # Your prompt contains multiple tasks
            that may be confusing to the executor.

        Your Inventory: Now your inventory has 4 logs.
        Equipped Item: Now you hold nothing in your hand.
        >>> execute("craft a wooden_axe", "wooden_axe", 1)
        Error: You cannot use `execute` to craft items. Use `craft`
            instead. # The executor cannot craft or smelt items, call `
            craft` for `smelt` function instead.

        Your Inventory: Now your inventory has 4 logs, 1 crafting_table.
        Equipped Item: Now you hold nothing in your hand.
        >>> execute("place crafting_table")
        Error: You cannot use `execute` to craft items or place the
            crafting_table. Directly use `craft` instead. No need to
            place the crafting_table. # The `craft` function will
            automatically place the crafting_table during crafting.

        Your Inventory: Now your inventory has nothing.
        Equipped Item: Now you hold nothing in your hand.
        >>> execute("hold down left button to punch the tree to collect
            wood", "logs", 1)
        The executor has reached the maximum number of steps for this
            turn without completing your subgoal. # The description of
            the task is too complex, it should be a **verb-object phrase
            **.

    Positive Examples: # good examples for reference
        Your Inventory: Now your inventory has stone_pickaxe, stick.
        Equipped Item: Now you hold the stone_pickaxe in your hand.
        >>> execute("break iron_ore blocks", "iron_ore", 2)
        Your subgoal has been successfully completed by the executor. #
            You have seen the iron_ore and you are using the correct tool
            . Note that if you haven't seen the iron_ore, you'd better
            use `break stone, obtain iron ore` as your prompt.

        Your Inventory: Now your inventory has nothing.
        Equipped Item: Now you hold nothing in your hand.
        >>> execute("collect wood", "logs", 1)
        Your subgoal has been successfully completed by the executor. #
            The executor can only understand the instructions of simple
            **verb-object phrases**.
```

```
        Your Inventory: Now your inventory has nothing.
        Equipped Item: Now you hold nothing in your hand.
        >>> execute("dig a hole", "dirt", 4)
        Your subgoal has been successfully completed by the executor. #
            Your instructions are simple and easy to understand.

        Your Inventory: Now your inventory has 1 wooden_axe, 2 stick.
        Equipped Item: Now you hold the wooden_axe in your hand.
        >>> execute("find a river")
        The executor has attempted to execute the action according to
            your prompt. You should check whether your intention has been
             fulfilled. # The executor has the ability to find the
            environment you are looking for, despite the possibility of
            failure.

    Prompt Examples: # some simple prompts for reference
    "chop down the tree", "break leaves", "collect seeds", "break a
        flower", "dig down", "break stone, obtain iron ore", "break
        gold_ore blocks", "mine diamond ore", "kill sheep", "milk cow", "
        combat spider", "find a river", "break stones", "break sand
        blocks", "move out of the cave".
    '''
```
## Reminder
1. You can only call one function in each turn.
2. If you have no idea on how to solve the task or are unfamiliar with
   some items, please call the `look_up` function to check the item.
3. For some items that you can not mine or obtain with your bare hand,
   try to equip a pickaxe (wooden_pickaxe, stone_pickaxe, ...) before
   mining it.
4. Some necessary resources (e.g., mobs, plants) might be prepared for
   you near the spawn point. If you're struggling to find certain
   ingredients or find yourself stuck somewhere, you can use the `
   teleport_to_spawn` function to return there.
5. When calling the executor, keep the positive examples and negative
   examples in mind! If the executor cannot complete your subgoal, check
    whether you have the right item in your hand, and try to break your
   prompt into smaller steps and adjust your subgoal, modify the prompt,
    or carefully repeat the prompt.
6. Do not repeat the failed action in the next round. Try to understand
   what went wrong and make a different decision.

# Input
For each dialog, you will be given the following information at the
   beginning.
- Task Goal: The item you should obtain in your inventory.
For each turn, you will be given the following information.
1. Feedback on the Action: The feedback on the action you output in the
   last turn.
2. Your Inventory: The items in your inventory.
3. Equipped Item: The item you are currently holding in your hand.
4. Location and Orientation: including X, Y, Z, Pitch and Yaw. X and Z
   are horizontal coordinates; Y is the height. Pitch measures the tilt
   of the player's view: 0, positive values and negative values mean the
    player is looking horizontally, downward, and upward, respectively.
   Yaw measures the rotation around the player's vertical axis: 0 or 360
    degrees north, 90 degrees east, 180 degrees south, and 270 degrees
   west.
5. Vision Input: What you see from your perspective.

# Output
Now, given these information, you need to think and call the action
   needed to proceed with the task. Your response should include 3 parts
    in the following format in each turn:

```
OBSERVATION: <What you observe in the image> Note that the Vision Input
    image won't be kept in the dialog, so make sure you capture all the
    key information (eg, the biome or items you see) here for future use.
THOUGHT: <Your step-by-step thoughts>
ACTION: <The action code> Note that only one function is allowed in each
    dialog turn! Only one line of code is allowed in each dialog turn! If
     your output contains multiple functions or multiple turns of
    functions, only the first one will be executed!
```

Here is a concrete example of the task input shown in Fig. 8, where the image is enclosed within
"{{}}":

```
Your task is to get a cake in your inventory.
Feedback on the Action: No action before.
Your Inventory: Now your inventory has 64 oak_log, 3 bucket.
Equipped Item: Now you hold the oak_log in your hand.
Location and Orientation: Now you locate in X: 431.50, Y: 65.00, Z:
    -158.50, Pitch: 0.00, Yaw: 0.00.
Vision Input: {{Image}}
```

# D  VAB-ANDROIDLAB

In this section, we provide additional details regarding VAB-AndroidLab that are not covered in the
main text due to space limitations.

## D.1  DETAILED DESCRIPTION

To introduce the Android Eval benchmark, we developed a framework including an operational
environment and a benchmark tailored for agents interacting with Android.

Android Eval benchmark comprises 119 tasks across 8 different apps, offering evaluation suites
considering the device's and screen's state. It implements evaluation frameworks for both the
ReAct (Yao et al., 2023) and SeeAct (Zheng et al., 2024) methods. For reproducibility, the Android
virtual device provides standard evaluation virtual machines preloaded with various apps' operation
histories and offline data, ensuring that network or temporal factors do not affect evaluations. To
simulate real-world tasks, we offer Android virtual machine images with randomized operations,
ensuring evaluations do not have to start from an initial usage state and enabling more complex task
completion recognition based on the machine and current page state.

## D.2  ACTIONS

In VAB-AndroidLab, agents are required to accomplish diverse user tasks through predefined actions.

- **tap**: Tap element with specific id.
- **type**: Type the message into the input box and press enter if needed.
- **long press**: Tap element with specific id for a long duration.
- **swipe**: Swipe with distance and direction.
- **finish**: Finish the task with optional message.
- **press back**: Press back button.
- **press home**: Press home button.

## D.3  METRICS

The metric we designed is directly oriented towards task completion. We can directly assess the
task's success rate by checking whether the operation sequence includes necessary screens or device
states that indicate task completion. For example, in setting an alarm time, we sequentially check if
the task sequence includes the correctly set alarm time and if the alarm is turned on. Specifically, the
metrics we designed are as follows:

- **Success Rate**: We measure the success rate by device state and screen state for the operation task.We measure the success rate for the query task by comparing the model answer with the ground truth.

## D.4 PROMPT EXAMPLE

Here is the system prompt we use.

```
You are an agent that is trained to complete certain tasks on a
    smartphone. You will be
given a screenshot of a smartphone app. The interactive UI elements on
    the screenshot are labeled with numeric tags
starting from 1.

You can call the following functions to interact with those labeled
    elements to control the smartphone:

1.tap(index: int)

Taps the UI element labeled with the given number.
Example: tap(5)

2.text(input_str: str)

Inserts the given text into an input field.
Example: text("Hello, world!")
Since we use ADB keyboard, if ADB keyboard ON is displayed on the bottom
    of the screen, you can use this function.
If you think that the keyboard is displayed after your previous operation
    , you can try to use this function to input text.

3.long_press(index: int)

Long presses the UI element labeled with the given number.
Example: long_press(5)

4. swipe(index: int, direction: str, dist: str)

Swipes the UI element in the specified direction and distance. "direction
    " is a string that
represents one of the four directions: up, down, left, right. "dist"
    determines the distance of the swipe and can be one
of the three options: short, medium, long.
Example: swipe(21, "up", "medium")

5. back()

Simulates a back button press on the smartphone.

6. home()

Simulates a home button press on the smartphone.

7. wait(interval: int)

Pauses the execution for the given number of seconds. Default is 5 second
    .

8. finish(message: str)

Ends the task and provides the final output. You can return the final
    output of the task as a string.
Example: finish("Task completed")
```

```
Now, given the following labeled screenshot, you need to think and call
    the function needed to proceed with the task.
Your output should include only action part in the given format:

Action: <The function call with the correct parameters to proceed with
    the task. If you believe the task is completed or
there is nothing to be done, you should use finish function. You cannot
    output anything else except a function call
in this field.>

Whenever you think the task is finished, you should use finish function
    to avoid extra operations.

If you found yourself in a loop or the task is not proceeding as expected
    , you might consider changing your operation and try other methods.
If you operate same action 5 times, the program will automatically stop.
If tap operation is not working, you can try long press operation.

You can only take one action at a time, so please directly call the
    function.
```

# E    WEBARENA-LITE

In this section, we provide additional details regarding WebArena-Lite that are not covered in the main text due to space limitations.

## E.1    DETAILED DESCRIPTION

WebArena (Zhou et al., 2023) is designed to evaluate the ability of agents to perform complex user tasks described in high-level natural language in a realistic, interactive web environment. To achieve this goal, WebArena presented a highly simulated and interactive web environment, which consists of five common websites, including Gitlab, map, forum, online shopping, and content management platform. It is also equipped with external tools such as sketch pad and calculator, which enhance the ability of the agents to perform user tasks. In contrast to other benchmarks where the agents are constrained to act as website users, WebArena proposed innovative ways to simulate different user roles. For instance, they constructed a content management platform (CMS) and granted the agent full administrative privileges. This assesses the agent's capacity to assume various roles in complex scenarios.

- **Task Description:** As web GUI agents, LMMs are asked to accomplish user instructions on certain websites. For example, on `OneStopShop` website, an instruction would be "`What do customers say about brush from sephora`", and LMM agents should search for the product, enter the review section, and summarize the customer reviews (or turn out finding no review). To enable the action of LMM agents with visual input, we implement HTML SoM (Koh et al., 2024) to annotate operable HTML elements with ids on the screenshot, we also provide a list of textual information for all clickable elements. LMM agents generate actions and the id of elements being operated by `playwright`.
- **Test Set:** We build WebArena-Lite, a subset of 165 representative tasks by selection, refinement, and adaptation to multimodality evaluation (i.e., screenshot). Our refinement focuses on resolving implausible judge conditions, where 30 tasks are being manually fixed (Cf. Appendix E.4). The implausibility may involve wrong answers, too-strict criteria (e.g., `exact_match`), impossible tasks due to environment bugs, etc. Additionally, we remove cross-website tasks for simplicity of testing.
- **Training Set:** Creating environment-dependent task instructions and trajectories for training on web could be challenging. In VAB, for each website we first summarize the basic functions and valid items for synthetic queries to condition on. Based on summarized functions, we come up with an array of task prototypes (with item placeholders) and manually write `playwright` scripts as rule-based solvers for each task prototype. We fill task prototypes with both valid and invalid items to yield detailed instructions (later being rephrased by LLMs for expression

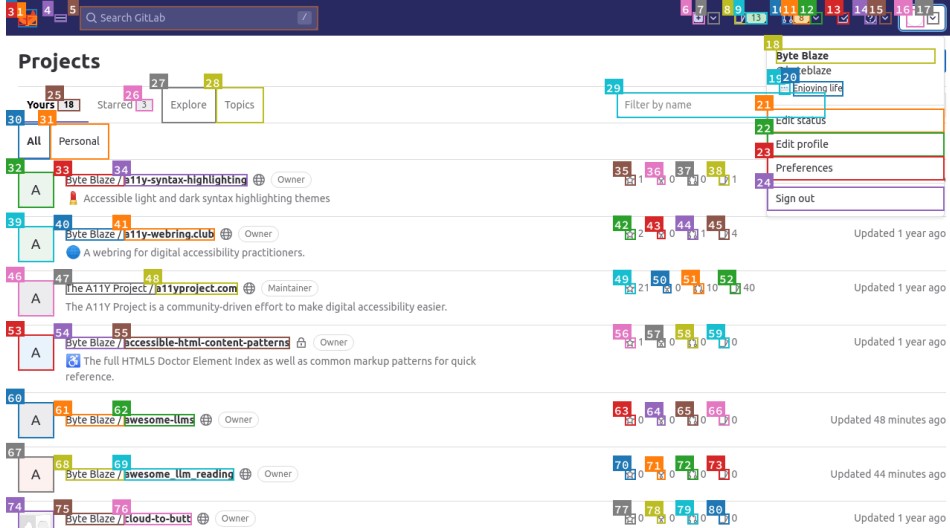

Figure 9: This is an example of WebArena-Lite task where we use the SoM approach to highlight actionable elements. This task requires the agent to modify the user's status information. To accomplish this, the agent initially clicks on the user's avatar, which directs them to the status shown in the figure. At this point, the agent should select the option labeled "(21) Edit Status" in order to access the modification page and complete the task.

diversity), and run corresponding solvers on the website to collect groundtruth trajectories with screenshots and operations. 5 authors create around 40 task prototypes with corresponding solvers, and generating 1,186 valid training samples (i.e., instruction, trajectory, and reward function) for WebArena-Lite.

### E.2 ACTIONS

In WebArena-Lite, agents are required to accomplish diverse user tasks through a series of predefined actions. However, real-world webpages are often complex, and thus, we provide these actions in order to ensure simplicity and practicality.

- **click**: Click element with specific id.
- **hover**: Hover element with specific id.
- **type**: Type the message into the input box with a specific id and press enter if needed.
- **press**: Emulates a keyboard key combination.
- **scroll**: Scrolls the page up or down.
- **new_tab**: Opens a new tab in the current browser.
- **tab_focus**: Switches to the tab with specific index.
- **close_tab**: Closes the current tab.
- **goto**: Go to specific URL.
- **go_back**: Go back to the previous page.
- **go_forward**: Go to the next page if it exists.
- **stop**: Terminates the operation, returns the response, and exits.

### E.3 METRICS

In real-world web browsing scenarios, there can be multiple ways for an agent to accomplish a task. Consequently, WebArena-Lite only considers whether the task has been completed or not, without considering the execution trajectory of the agent, therefore, the metric used in WebArena-Lite is **Success Rate (SR)**. We maintain the evaluation method described by WebArena (Zhou et al., 2023), which can be categorized into three categories based on task type.

- **Question Answering**: Agent needs to give an answer and the score depends on the string-matching result.
- **Webpage Navigation**: Agent must navigate to a specific web page. The completion of the task is dependent on the URL of the page on which the agent terminated.
- **Content modification**: Agent needs to interact with the environment to modify the configuration of the webpage, and the evaluation will extract the content of the page and match it to check whether the content meets the expectations.

In light of the aforementioned considerations, string-matching patterns can be classified into three distinct categories:

- **exact_match**: The response of the agent is scored when it exactly matches the token sequence corresponding to the answer.
- **must_include**: Answers that contain a specific token sequence are considered a match.
- **fuzzy_match**: Utilizes LLMs such as GPT-4 to assist in determining whether an answer is correct.

The selection of appropriate evaluation metrics for distinct types of tasks enables the construction of a comprehensive and relatively accurate test set.

### E.4    TASK AMENDMENT

Some tasks in WebArena have typos, incorrect answers, and inaccurate scoring criteria. Therefore, we selected 165 tasks from WebArena with different templates and then corrected 39 of them, as shown in Table E.4. Considering that the model uses natural language to answer the questions, we change the tasks that require exact match to must include or fuzzy match, and also correct the answers.

Table 11: Task instructions fixed in WebArena-Lite

| ID | Website | Task | Before | After |
|---|---|---|---|---|
| 7 | Map | Tell me the full address of all international airports that are within a driving distance of 50 km to Carnegie Mellon University. | exact_match | fuzzy_match |
| 33 | Map | I will arrive Pittsburgh Airport soon. Provide the name of a Hilton hotel in the vicinity, if available. Then, tell me the the shortest walking distance to a supermarket from the hotel. | must_include | fuzzy_match |
| 37 | Map | Check if the police station in pittsburgh can be reached in one hour by car from gates building at CMU. | must_include | fuzzy_match |
| 43 | CMS | List the top 3 search terms in my store. | hollister, Joust Bag, Antonia Racer Tank | hollister, Joust Bag, nike |
| 65 | CMS | Which customer has completed the fifth most number of orders in the entire history? | Jane Doe | Matt Baker |
| 71 | Map | What is the zip code of Chatham University? | exact_match | must_include |
| 82 | Map | What is the duration required to first walk from Massachusetts Institute of Technology to Harvard University, and then drive to Boston Logan International Airport? | 63 min | 64 min |
| 97 | Map | Tell me the distance to drive from Carnegie Mellon University to the top computer science school in massachusetts. | must_include | fuzzy_match |
| 98 | Map | Where is the nearest tea cafe to University of Pittsburgh, and what is the walking distance to it? | must_include | fuzzy_match |
| 103 | Gitlab | Display the list of issues in the kkroening/ffmpeg-python repository that have labels related to questions. | | URL: sort by created_date, state is opened |

Table 11: Task instructions fixed in WebArena-Lite

| ID | Website | Task | Before | After |
|---|---|---|---|---|
| 109 | CMS | Presents the monthly count of successful orders {{period}} in MM:COUNT format. | January: 11 orders, February: 16 orders | 01:11, 02:16 |
| 127 | CMS | What brands appear most frequently among the top search terms? | Hollister, Joust, Antonia | Hollister |
| 135 | Gitlab | How many commits did Eric and Kilian make to a11yproject on 1/3/2023? | 1 | 0 |
| 167 | OSS | What are the main criticisms of this product? Please extract the relevant sentences. | must_include | fuzzy_match |
| 215 | CMS | What are the key aspects that the customers don't like about Circe ice fleece. | fuzzy_match ("Material quality, . . .") | exact_match ("N/A") |
| 225 | OSS | What do customers say about brush from sephora. | N/A | No reviews available |
| 235 | OSS | Get the order number of my most recent under delivery order. | fuzzy_match | must_include |
| 236 | Map | Where is the nearest pharmacy from Carnegie Mellon I can walk within 20mins. | must_include | fuzzy_match |
| 240 | OSS | I am doing a market survey for one stop market, show me the most expensive product from competative swimwear category. | | URL |
| 254 | Map | What is the phone number of Western Pennsylvania Hospital. | exact_match | fuzzy_match |
| 259 | Gitlab | Get me my RSS feed token. | exact_match | must_include |
| 287 | Map | How much time does it take from Pittsburgh to Philadelphia by car? | 5h 47m | 8h 33m |
| 288 | CMS | Tell me the name of the customer who has the most cancellations in the history. | exact_match | must_include |
| 296 | Gitlab | Show me the command to clone the best GAN python implementation with SSH. | metis.lti. cs.cmu.edu | localhost |
| 311 | Gitlab | Tell me who has made the most contributions, in terms of number of commits, to the Pytorch GAN project. | exact_match | must_include |
| 348 | CMS | How many reviews our shop received in May 2023? | must_include | fuzzy_match |
| 349 | Gitlab | Who else have access to my repo gimmiethat.space, show me their usernames. | exact_match | must_include |
| 367 | Map | Measure distance between Carnegie Mellon University and CVS (closet one) by walking. | exact_match | fuzzy_match |
| 376 | OSS | Summarize customer reviews for Amazon Echo Dot 3rd generation. | N/A | fuzzy_match ("Serena: it. . .") |
| 392 | Gitlab | Post \"Good idea\" for the merge request related to color ulitity in a11yproject.com project". | | URL |
| 528 | OSS | Draft a refund message via their \"contact us\" form for the phone screen protector I bought March 2023. It broke after three days of use. The shop requires the order id, the reason and the amount to refund in the message. Don't submit yet. | must_include | fuzzy_match |
| 538 | CMS | Modify the address of order #299 to 456 Oak Avenue, New York, NY, 10001. | . . . 456 Oak Avenue, Apartment 5B, New York. . . | . . . 456 Oak Avenue, New York. . . |
| 641 | Reddit | Post a notice on a virtual meetup for Harry Potter enthusiasts on July 8th in the books subreddit. | must_include | fuzzy_match |
| 645 | Reddit | Post in books subreddit about what could machine learning help the correpong field. | must_include | fuzzy_match |

Table 11: Task instructions fixed in WebArena-Lite

| ID | Website | Task | Before | After |
|----|---------|------|--------|-------|
| 657 | OSS | Fill the \"contact us\" form in the site for a refund on the speaker I bought, stating that it broke after just three days of use. Also, ensure to include the order number #148 and the product SKU. Don't submit yet, I will check. | must_include | fuzzy_match |
| 668 | Gitlab | Submit a merge request for a11yproject.com/redesign branch to be merged into master branch, assign Roshan Jossy as the reviewer. | Justin Armstrong | Roshan Jossy |
| 693 | OSS | Draft an email to the shop owner via their contact us function for a coupon as my refund is suppoed to be replaced by a coupon. | program_match | url_match |
| 798 | OSS | Change the delivery address for my most recent order to 77 Massachusetts Ave, Cambridge, MA. | fuzzy_match | must_include |

### E.5 PROMPT EXAMPLE

Here is the system prompt we use, you can find more prompt examples in VisualWebArena (Koh et al., 2024).

```
You are an autonomous intelligent agent tasked with navigating a web
    browser. You will be given web-based tasks. These tasks will be
    accomplished through the use of specific actions you can issue.

Here's the information you'll have:
The user's objective: This is the task you're trying to complete.
The current web page's accessibility tree: This is a simplified
    representation of the webpage, providing key information.
The current web page's URL: This is the page you're currently navigating.
The open tabs: These are the tabs you have open.
The previous action: This is the action you just performed. It may be
    helpful to track your progress.

The actions you can perform fall into several categories:

Page Operation Actions:
```click [id]```: This action clicks on an element with a specific id on
    the webpage.
```type [id] [content]```: Use this to type the content into the field
    with id. By default, the "Enter" key is pressed after typing unless
    press_enter_after is set to 0, i.e., ```type [id] [content] [0]```.
```hover [id]```: Hover over an element with id.
```press [key_comb]```:  Simulates the pressing of a key combination on
    the keyboard (e.g., Ctrl+v).
```scroll [down]``` or ```scroll [up]```: Scroll the page up or down.

Tab Management Actions:
```new_tab```: Open a new, empty browser tab.
```tab_focus [tab_index]```: Switch the browser's focus to a specific tab
     using its index.
```close_tab```: Close the currently active tab.

URL Navigation Actions:
```goto [url]```: Navigate to a specific URL.
```go_back```: Navigate to the previously viewed page.
```go_forward```: Navigate to the next page (if a previous 'go_back'
    action was performed).

Completion Action:
```

```
```stop [answer]```: Issue this action when you believe the task is
    complete. If the objective is to find a text-based answer, provide
    the answer in the bracket.

Homepage:
If you want to visit other websites, check out the homepage at http://
    homepage.com. It has a list of websites you can visit.
http://homepage.com/password.html lists all the account name and password
    for the websites. You can use them to log in to the websites.

To be successful, it is very important to follow the following rules:
1. You should only issue an action that is valid given the current
    observation
2. You should only issue one action at a time.
3. You should follow the examples to reason step by step and then issue
    the next action.
4. Generate the action in the correct format. Start with a "In summary,
    the next action I will perform is" phrase, followed by action inside
    ``````. For example, "In summary, the next action I will perform is
    ```click [1234]```".
5. Issue stop action when you think you have achieved the objective. Don'
    t generate anything after stop.
```

## F  VAB-CSS

In this section, we provide additional details regarding VAB-CSS that are not covered in the main text due to space limitations.

### F.1  DETAILED DESCRIPTION

Existing datasets for frontend design have two major shortcomings: 1) They focus mainly on single-round interactions, and 2) They do not provide definitive success metrics for individual tasks. Instead, these benchmarks assess using continuous metrics like CLIP score (Si et al., 2024) or qualitative analysis only (Laurençon et al., 2024). The reason is that they expect the model to output an entire HTML file replicating the target web design, which is too challenging and unrealistic for current LMMs. Therefore, employing a definitive success rate as the metric is meaningless for them. Consequently, they may fail to adequately assess LMMs' potential in serving as adaptive agents that can make new decisions based on varying observations. Also, a binary success rate is often more decisive and crucial to determine whether agents can faithfully execute human instructions, which is essential for practical use. To address these issues, we introduce a VAB-CSS, which is better tailored for evaluating multimodal agents. In VAB-CSS, an agent is expected to strictly take a sequence of actions using our provided toolkit to accomplish a task (Section. F.3). Specifically, it needs to iteratively refine the CSS definition based on the rendering outcomes it receives. The more constrained action space based on our toolkit, compared to outputting an entire HTML file, along with a more practical goal for current LMMs (i.e., CSS bug-fixing), makes it possible to evaluate a definitive success rate for a given task. Additionally, VAB-CSS makes minimal assumptions in terms of simplifying the task environment, such as embedding all CSS definitions within a single HTML page or replacing images with placeholders in existing datasets. Instead, the agent directly operates over the entire web frontend project to fix the CSS style. See a comprehensive checklist in Table 12.

### F.2  DATA COLLECTION

**Random CSS Corruption.** To ensure the task is manageable for LMMs, each task instance involves corrupting a single categorical property of a random CSS rule by either altering its value or removing it entirely. Note that, even fixing a single corruption is already highly challenging for current LMMs, and a tiny corruption can often lead to a drastic change in visual effect (see Figure 10). We can increase the task's complexity in the future by involving multiple corruptions once the single-corruption task has been mastered.

Table 12: A fine-grained comparison of VAB-CSS with existing datasets on web frontend development. VAB-CSS provides both training and test data. Additionally, its multi-round nature, definitive success rate metric, and multi-file environment make it well-suited as a practical multimodal agent task.

| | Train | Test | Multi Round | Definitive Eva. | Multi-File Env. |
|---|---|---|---|---|---|
| WebSight (Laurençon et al., 2024) | ✓ | ✗ | ✗ | ✗ | ✗ |
| Design2Code (Si et al., 2024) | ✗ | ✓ | ✗ | ✗ | ✗ |
| VAB-CSS | ✓ | ✓ | ✓ | ✓ | ✓ |

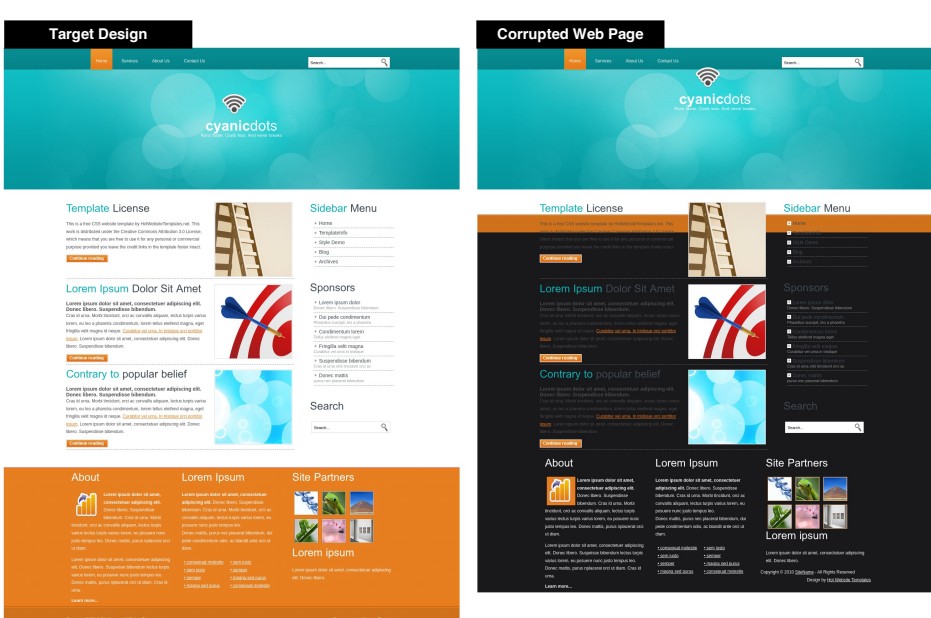

Figure 10: This is an example of our annotation task. Authors are shown the target design and a corrupted web page side by side to prompt them to describe necessary adjustments in natural language. In this example, the instruction is: "*Correct the background color of the footer and main section, and adjust the positioning of elements, including centering the website logo in the header by moving it downward.*" The two screenshots, along with the HTML code and annotated instruction, will collectively serve as the initial task input for the agent.

**Human Annotations.** Existing LMMs struggle to identify the difference between the current rendering and the target design, so we manually annotate each instance with a natural language description of the difference between the two images. Such natural language descriptions could serve as additional clues for the agent to perceive the visual difference (see a concrete example of annotation in Figure 10).

**Training Trajectories.** To collect training trajectories, we primarily sample from the predictions of `gpt-4o` on our training instances, retaining the successful trajectories for training. Given the success rate of `gpt-4o` is around 35%, we also sample its trajectories in a more lenient setting where the target CSS rule to edit is provided as input. For task instances where `gpt-4o` succeeds in the lenient setting, we combine its successful trajectory with its failure trajectory in the standard setting to create a more realistic trial-and-error trajectory.

### F.3 ACTIONS

In VAB-CSS, the agent is expected to interact with a practical frontend project, potentially with numerous CSS files, to fix its style issues. Inputting the entire project directly into the agent is impractical and inefficient. Instead, the agent has access only to screenshots and the current HTML

code. To facilitate effective navigation and editing within the project, we provide the agent with a toolkit. This toolkit allows the agent to locate and edit incorrect CSS definitions seamlessly, without needing to know the specific file containing the CSS rule.

- **get_selectors_by_html_element**: This function allows the agent to locate a list of CSS selectors, potentially from various files, associated with an HTML element whose style appears to be incorrect.
- **select_rule**: This function allows the agent to check the definition of a CSS rule by specifying a CSS selector.
- **edit_rule**: This function enables the agent to update the property value of a CSS rule for a specified CSS selector.
- **revert_last_edit**: During the trial and error, the agent can revert an edit it later determines to be incorrect.

## F.4 METRICS

As discussed earlier, a critical feature of VAB-CSS, compared with existing benchmarks, is its definitive success rate evaluation. The most straightforward way to determine whether a task is successfully handled is to check whether the SSIM similarity between the target design and the final rendering is $1.0$. However, we have observed that this can be too strict. Typically, an SSIM greater than $0.9$ indicates minimal differences that are hard for humans to perceive.[4] Therefore, we define a task as successful if the final similarity is greater than $0.9$. Finally, we adopt two metrics on our entire test set.

- **Success Rate (SR)**: This is the primary metric indicating the ratio of tasks in the test set that have been successfully fixed based on our definition.
- **Improve Rate (IR)**: This metric evaluates the ratio of tasks where the final rendering is more similar to the target design than the initial rendering. It serves as a complementary soft metric to the success rate.

## F.5 PROMPT EXAMPLE

The system message that describes the detailed task information to the agent is shown as follows:

```
You are a CSS agent. You will be given a target screenshot and an html
    file. Your job is to correct perceive the layout difference between
    the current rendering and the target screenshot, then accordingly fix
     the css rules used in the html file to match the target screenshot.
To facilitate the process, you can use the following tools provided by
    the system:
1. get_selectors_by_html_elements
Sometimes, the exact selector of the rule you want to edit is not clear.
    This tool takes the html element specification that could be directly
     passed to soup.find_all as input and returns the matched selectors.
    For example, get_selectors_by_html_elements("’a’, {’data-custom’: ’
    custom-value’}, string=’haha’, class_=’xxx’"). The argument should be
     the string representation of valid arguments of the find_all method
    in BeautifulSoup, which means we can directly do eval(f"soup.find_all
    ({argument})"). Please strictly stick to the usage of BeautifulSoup.
    Make sure the arguments are valid (e.g., the tag name must be wrapped
     with quotes, attributes should be a dictionary). You can use this
    tool to first find the selector of the rule of a specific html
    element whose style you want to fix.
2. select_rule
This takes the css rule’s selectorText as input and returns the rule. You
     can use this tool to view the properties of a rule, which may help
    you to decide which rule to edit. Usually, it’s recommended to first
    use this tool to view the rule before deciding which rule to edit.
3. edit_rule
```

---

[4]This threshold of $0.9$ is an empirical choice based on our observations.

```
This takes the css rule's selectorText, the property name, and the value
    of the property as input. You can use this tool to change the value
    of a property of a rule or insert a new property to the rule, if you
    believe this change would make the rule closer to the target
    screenshot. Note that, most of the layout issues are related to the
    categorical properties, such as border, float, display, overflow,
    position, etc.
4. revert_last_edit
This tool reverts the last single edit you made. You can use this tool to
     undo the last edit, if you believe it was a mistake. This action
    takes no arguments.

Make sure the selectorText is valid based on the html file, i.e., it's
    from the class or id of the html elements. In addition, please focus
    on the major layout issue! Ignore the font size, font family, and
    color of the text, even if you believe they are not perfect.

You can only take ONE action at a time!! For each step, you may first
    state your thought, then take an action following the format of
    Thought: ...
 Action: ... (do not add any linebreak after the colon).
For example, you may output
"Thought: I think I should adjust the alignment property of the rule,
    because the target screenshot shows the text should be centered.
 Action: edit_rule('.templatemo_menu li', 'text-align', 'center')".

After editing a rule or inserting a rule, you will see the updated
    screenshot of the html file. You should decide your next action (e.g
    ., to revert the last edit or keep adjusting the css) based on the
    updated screenshot. If you think you have already fixed the css style
    , please say exactly "I have fixed the css style".

Please strictly follow the format specified above, and please don't
    repeat the same action in multiple rounds. Also note that, you don't
    need to worry about how these tools are executed, your job is just to
     correctly predict the tool invocation.
```

Here is a concrete example of the task input shown in Fig. 10, where variables are enclosed within "{{}}":

```
Here is a screenshot of the target design:
{{Image 1}}
Here is the screenshot of the current web page:
{{Image 2}}
Here is the HTML code of the current web page:
{{HTML file}}

Correct the background color of the footer and main section, and adjust
    the positioning of elements, including centering the website logo in
    the header by moving it downward.
```

## G CASE STUDIES

### G.1 VAB-OMNIGIBSON CASES

In this section, we show successful and failed examples of gpt-4o and InternVL-2 in Figure 11, Figure 12, Figure 13 and Figure 14.

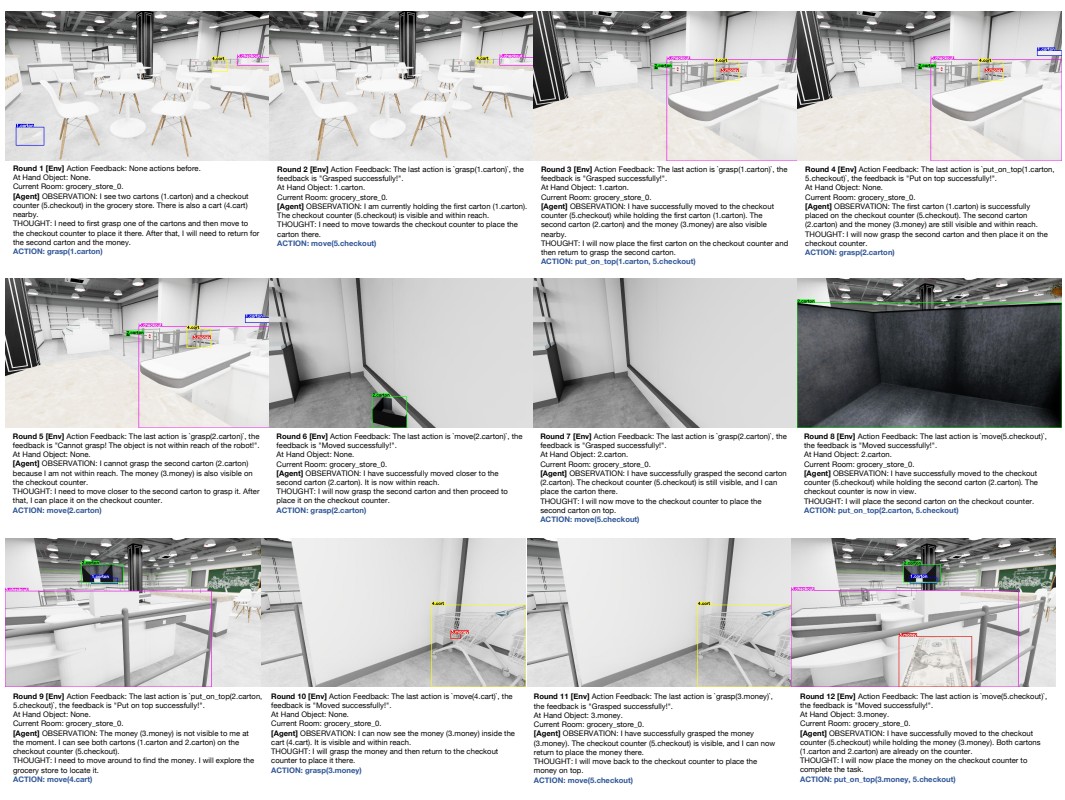

Figure 11: This is an example of `gpt-4o` successfully completing a VAB-OmniGibson task. Task instruction: "Place the two cartons and the money on the checkout counter in the grocery store."

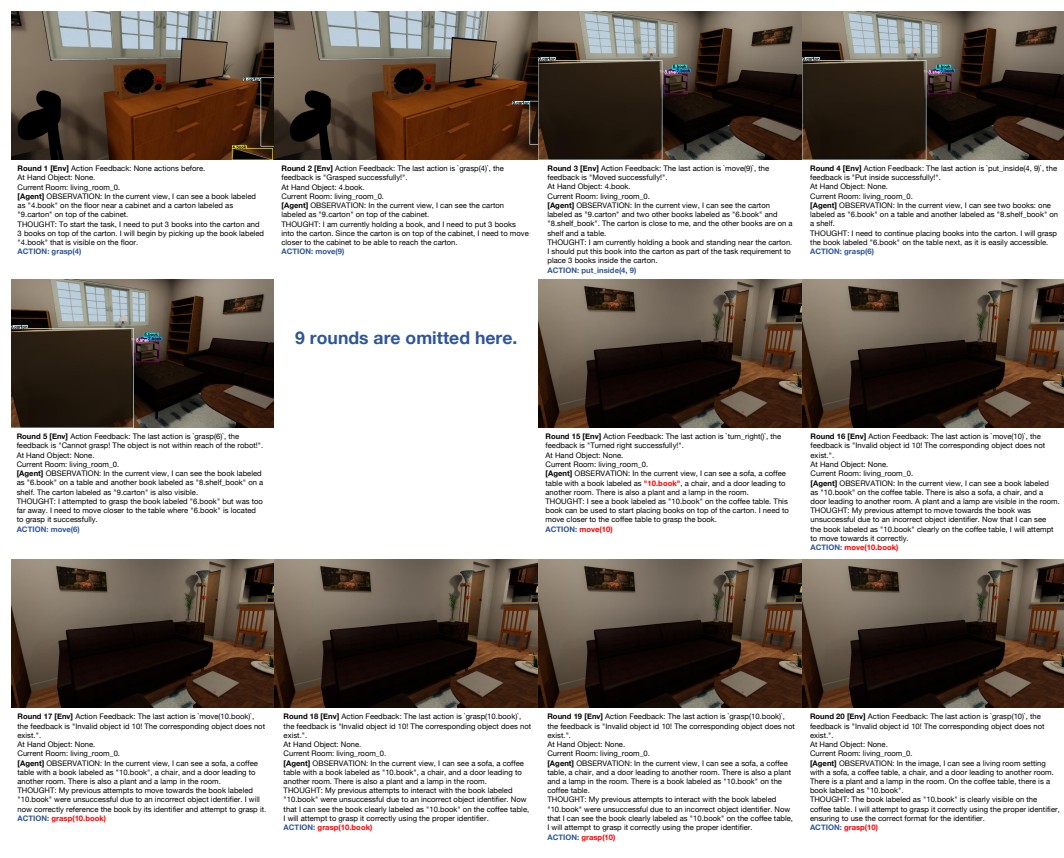

Figure 12: This is an example of `gpt-4o` failing to complete a VAB-OmniGibson task. Task instruction: "There are 7 books in the living room. Please put 3 books into the carton, and 3 books on top of the carton." `gpt-4o` incorrectly identified the presence of "10.book" in the image, and the erroneous actions are highlighted in red.

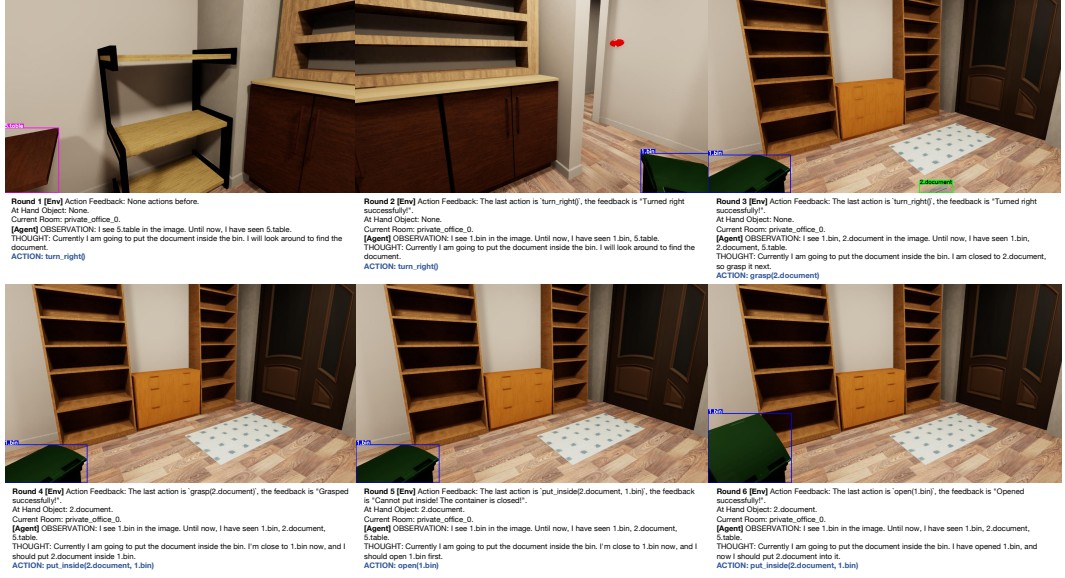

Figure 13: This is an example of `InternVL-2` successfully completing a VAB-OmniGibson task. Task instruction: "Pick up a document from the breakfast table or the floor, and then throw it into the recycling bin in the private office."

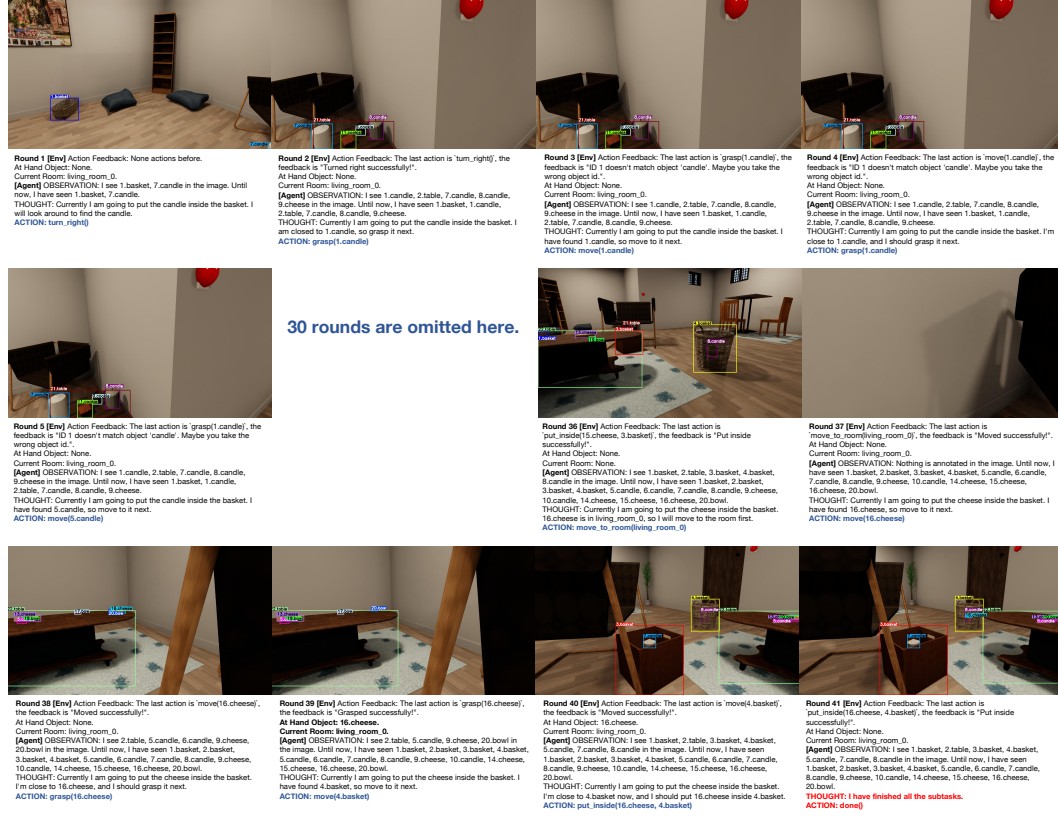

Figure 14: This is an example of `InternVL-2` failing to complete a VAB-OmniGibson task. Task instruction: "There are 4 candles, 4 cheeses and 4 baskets in the living room. Please place 5.candle and 13.cheese into 1.basket, 6.candle and 14.cheese into 2.basket, 7.candle and 15.cheese into 3.basket, 8.candle and 16.cheese into 4.basket." `InternVL-2` didn't put "5.candle" into the basket, but terminated the task with action "done()".

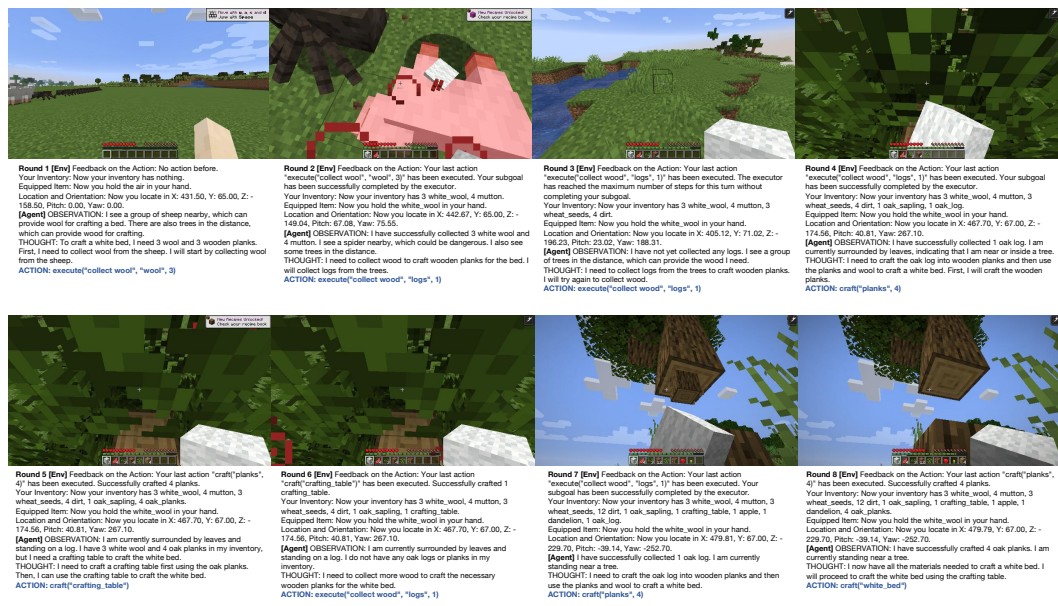

Figure 15: This is an example of `gpt-4o` successfully completing a VAB-Minecraft task. Task instruction: "Your task is to get a white_bed in your inventory."

## G.2 VAB-MINECRAFT CASES

In this section, we show successful and failed examples of `gpt-4o` and `InternVL-2` in Figure 15, Figure 16, Figure 17 and Figure 18.

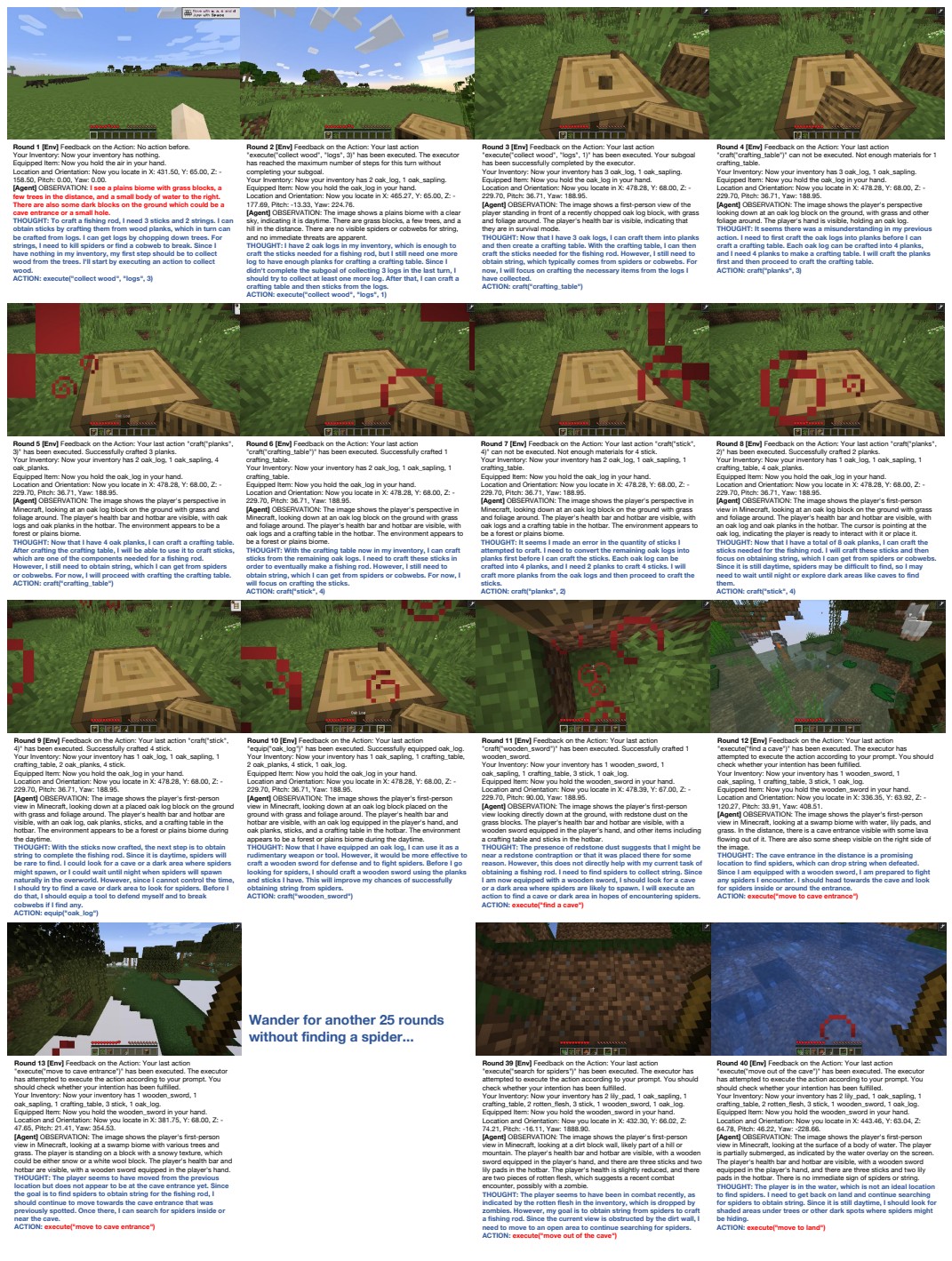

Figure 16: This is an example of `gpt-4o` failing to complete a VAB-Minecraft task. Task instruction: "Your task is to get a fishing_rod in your inventory." `gpt-4o` didn't notice the spiders near the spawn point and couldn't find any other spiders elsewhere. Instead of wandering around (actions marked as red), it should "teleport_to_spawn" to find spiders.

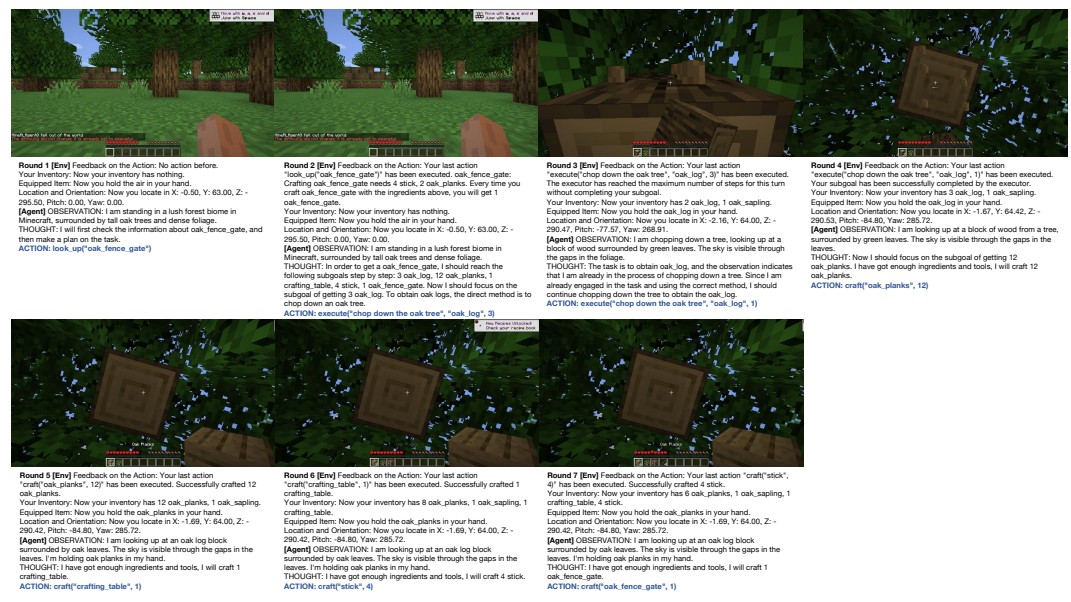

Figure 17: This is an example of `InternVL-2` successfully completing a VAB-Minecraft task. Task instruction: "Your task is to get a oak_fence_gate in your inventory."

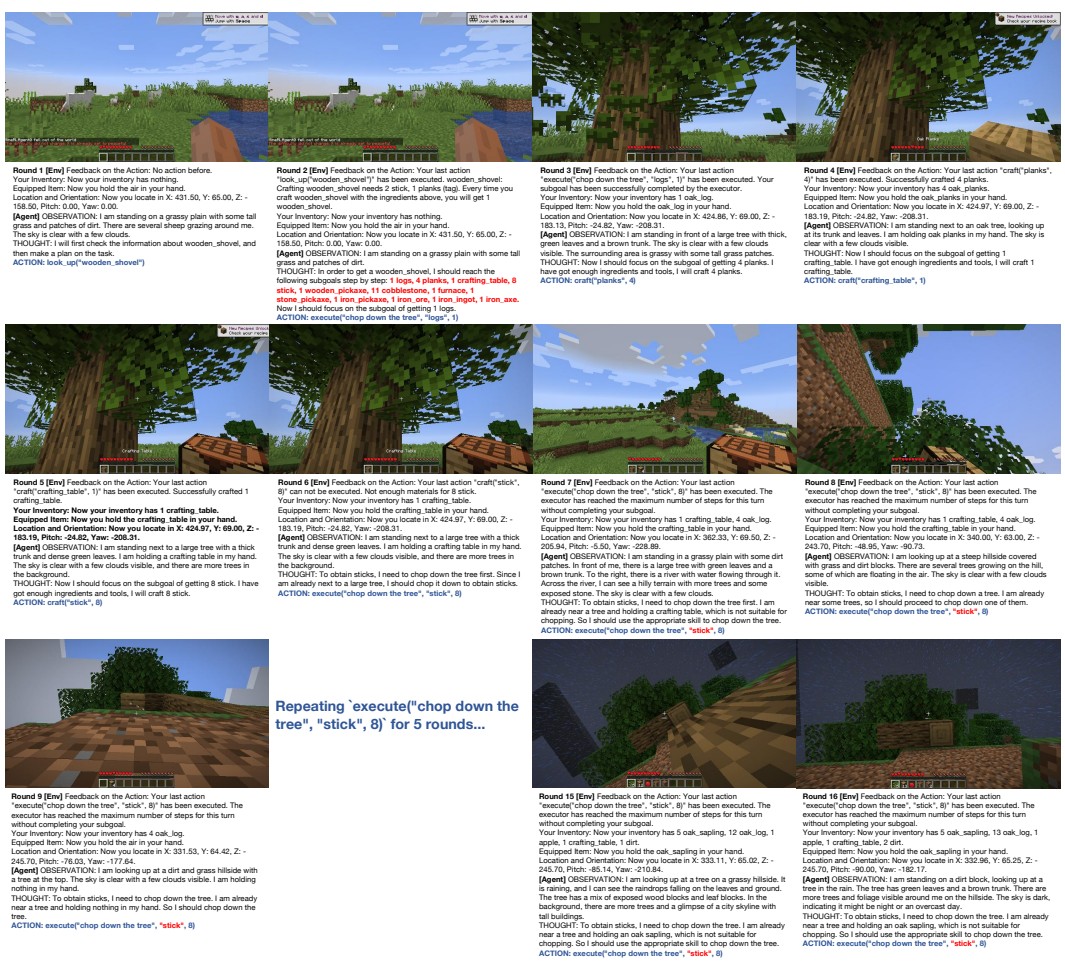

Figure 18: This is an example of `InternVL-2` failing to complete a VAB-Minecraft task. Task instruction: "Your task is to get a wooden_shovel in your inventory." `InternVL-2` made a wrong plan for it (marked as red in Round 2), and persistently tried to obtain sticks by chopping down trees, which is infeasible (actions marked as red from Round 7 to 16).

### G.3 VAB-ANDROIDLAB CASES

In this section, we show successful and failed examples of `gpt-4o` in Figure 19 and Figure 20.

Figure 19: This is an example of `gpt-4o` successfully completing a VAB-AndroidLab task. Task instruction: "Check my default browser and change it to firefox"

Check the walking distance and time between Bus Stop of Stanford
Campus Oval and Bus Stop of Oxford Street & University Avenue

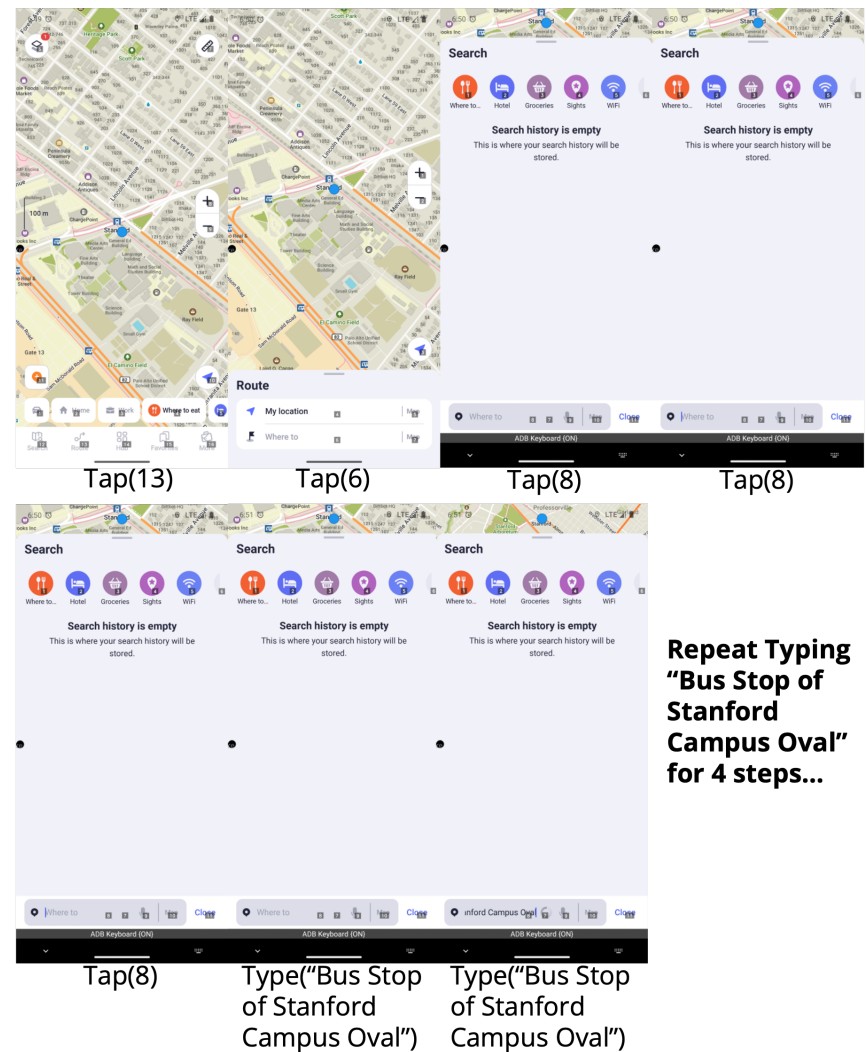

Figure 20: This is an example of `gpt-4o` failing to complete a VAB-AndroidLab task. Task instruction: "Check the walking distance and time between Bus Stop of Stanford Campus Oval and Bus Stop of Oxford Street & University Avenue" `gpt-4o` repeated doing typing without noticing the success of previous typing.

## G.4 WEBARENA-LITE CASES

In this section, we show successful and failed examples of `gpt-4o` in Figure 21 and Figure 22.

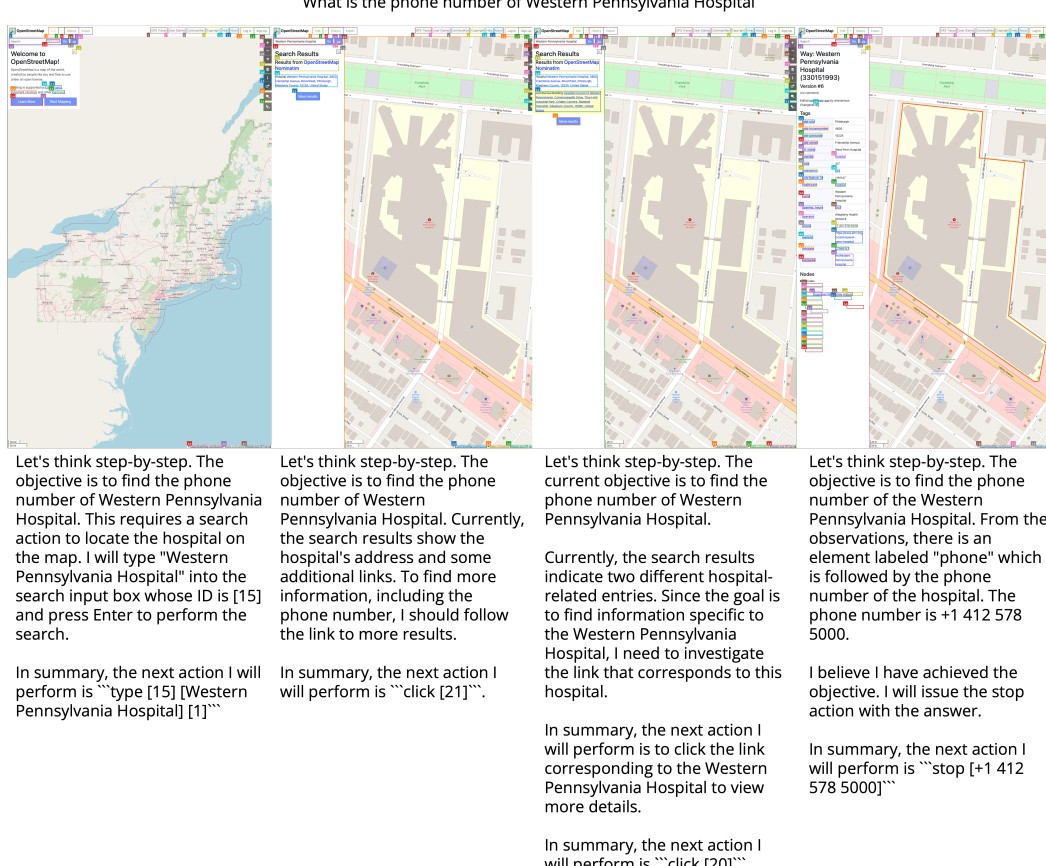

Let's think step-by-step. The objective is to find the phone number of Western Pennsylvania Hospital. This requires a search action to locate the hospital on the map. I will type "Western Pennsylvania Hospital" into the search input box whose ID is [15] and press Enter to perform the search.

In summary, the next action I will perform is ```type [15] [Western Pennsylvania Hospital] [1]```

Let's think step-by-step. The objective is to find the phone number of Western Pennsylvania Hospital. Currently, the search results show the hospital's address and some additional links. To find more information, including the phone number, I should follow the link to more results.

In summary, the next action I will perform is ```click [21]```.

Let's think step-by-step. The current objective is to find the phone number of Western Pennsylvania Hospital.

Currently, the search results indicate two different hospital-related entries. Since the goal is to find information specific to the Western Pennsylvania Hospital, I need to investigate the link that corresponds to this hospital.

In summary, the next action I will perform is to click the link corresponding to the Western Pennsylvania Hospital to view more details.

In summary, the next action I will perform is ```click [20]```.

Let's think step-by-step. The objective is to find the phone number of the Western Pennsylvania Hospital. From the observations, there is an element labeled "phone" which is followed by the phone number of the hospital. The phone number is +1 412 578 5000.

I believe I have achieved the objective. I will issue the stop action with the answer.

In summary, the next action I will perform is ```stop [+1 412 578 5000]```

Figure 21: This is an example of `gpt-4o` successfully completing a WebArena-Lite task. Task instruction: "What is the phone number of Western Pennsylvania Hospital"

Promote auth0/angular-storage to subreddit technology with the description from the repo itself.

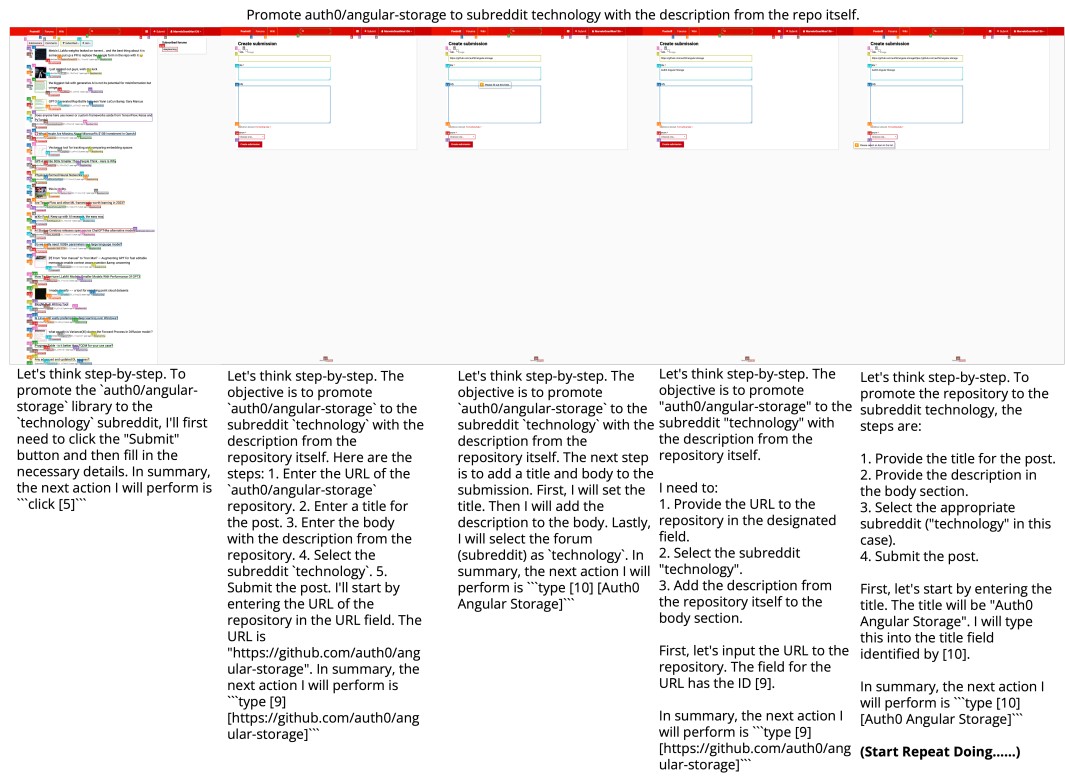

Let's think step-by-step. To promote the `auth0/angular-storage` library to the `technology` subreddit, I'll first need to click the "Submit" button and then fill in the necessary details. In summary, the next action I will perform is ```click [5]```

Let's think step-by-step. The objective is to promote `auth0/angular-storage` to the subreddit `technology` with the description from the repository itself. Here are the steps: 1. Enter the URL of the `auth0/angular-storage` repository. 2. Enter a title for the post. 3. Enter the body with the description from the repository. 4. Select the subreddit `technology`. 5. Submit the post. I'll start by entering the URL of the repository in the URL field. The URL is "https://github.com/auth0/angular-storage". In summary, the next action I will perform is ```type [9] [https://github.com/auth0/angular-storage]```

Let's think step-by-step. The objective is to promote `auth0/angular-storage` to the subreddit `technology` with the description from the repository itself. The next step is to add a title and body to the submission. First, I will set the title. Then I will add the description to the body. Lastly, I will select the forum (subreddit) as `technology`. In summary, the next action I will perform is ```type [10] [Auth0 Angular Storage]```

Let's think step-by-step. The objective is to promote "auth0/angular-storage" to the subreddit "technology" with the description from the repository itself.

I need to:
1. Provide the URL to the repository in the designated field.
2. Select the subreddit "technology".
3. Add the description from the repository itself to the body section.

First, let's input the URL to the repository. The field for the URL has the ID [9].

In summary, the next action I will perform is ```type [9] [https://github.com/auth0/angular-storage]```

Let's think step-by-step. To promote the repository to the subreddit technology, the steps are:

1. Provide the title for the post.
2. Provide the description in the body section.
3. Select the appropriate subreddit ("technology" in this case).
4. Submit the post.

First, let's start by entering the title. The title will be "Auth0 Angular Storage". I will type this into the title field identified by [10].

In summary, the next action I will perform is ```type [10] [Auth0 Angular Storage]```

**(Start Repeat Doing......)**

Figure 22: This is an example of `gpt-4o` failing to complete a WebArena-Lite task. Task instruction: "Promote auth0/angular-storage to subreddit technology with the description from the repo itself." `gpt-4o` repeated doing typing without noticing the success of previous typing.

## G.5 VAB-CSS CASES

We show successful and failed cases of both `gpt-4o` and `InternVL-2` in Figure 23, Figure 24, Figure 25 and Figure 26.

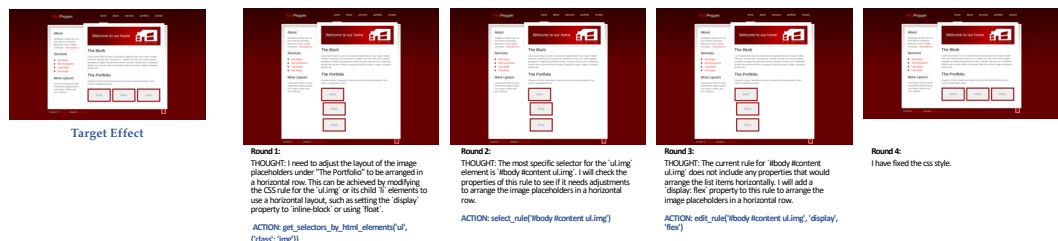

Figure 23: This is an example of `gpt-4o` successfully completing a VAB-CSS task.

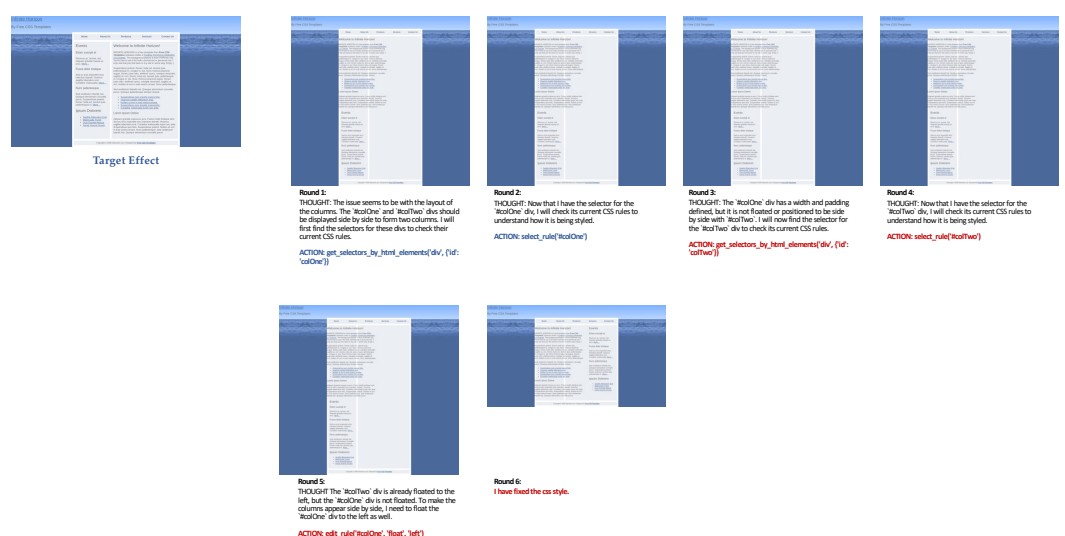

Figure 24: This is an example of `gpt-4o` failing to complete a VAB-CSS task.

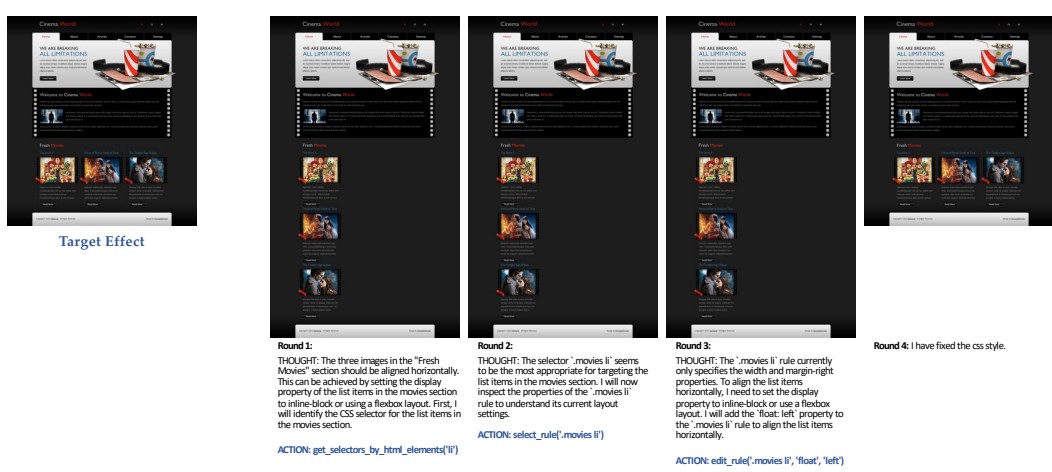

Figure 25: This is an example of `InternVL-2` successfully completing a VAB-CSS task.

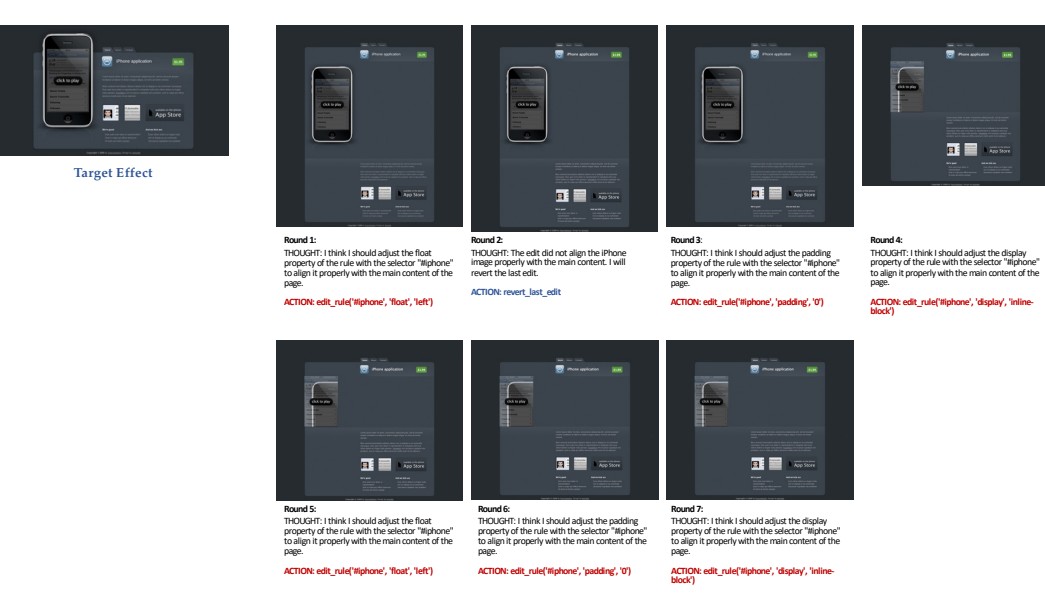

Figure 26: This is an example of `InternVL-2` failing to complete a VAB-CSS task.

## G.6 ERROR RECOVERY CASES

In this section, we show examples of error recovery in each environment, as illustrated in Figure 27, Figure 28, Figure 29, Figure 30, Figure 31.

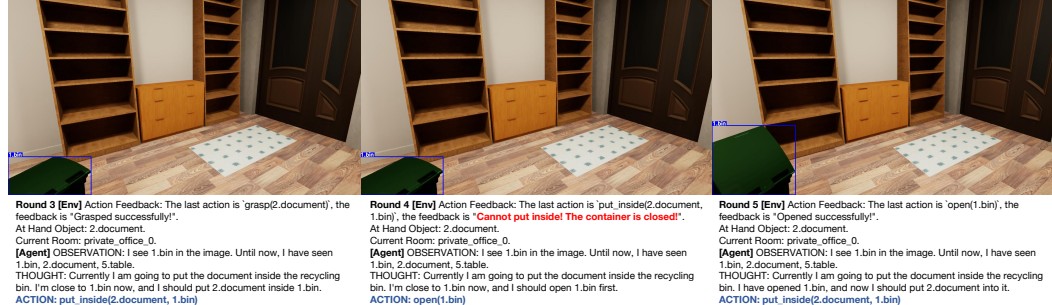

Figure 27: In VAB-OmniGibson, the agent encounters a failure when attempting to place the document into a closed bin. Upon receiving feedback from the environment, the agent first opens the bin and successfully deposits "2.document" into it.

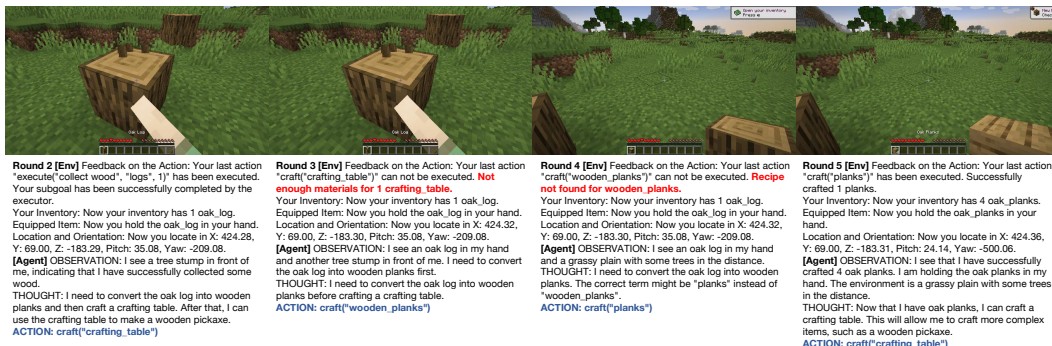

Figure 28: In VAB-Minecraft, the agent initially attempts to craft a crafting_table without the required ingredients. Upon receiving feedback, the agent first attempts to craft "wooden_planks", which is an incorrect ingredient name. Subsequently, it corrects its approach by crafting "planks" and is then able to successfully craft the crafting_table.

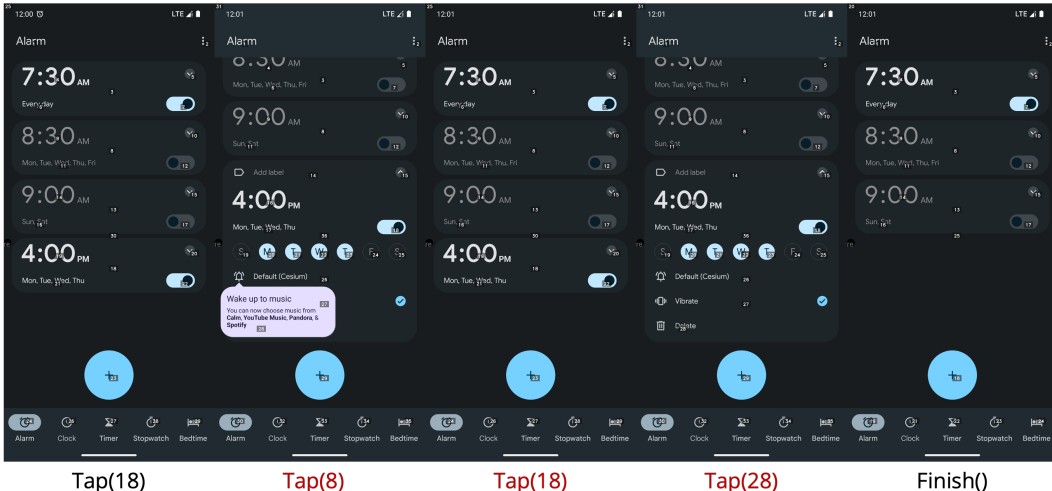

Figure 29: In VAB-AndroidLab, the agent initially clicks on the correct alarm to be deleted. However, a popup unexpected appears and trigger a wrong clicking of the agent. After the wrong action, it realizes the error and again perform the correct action.

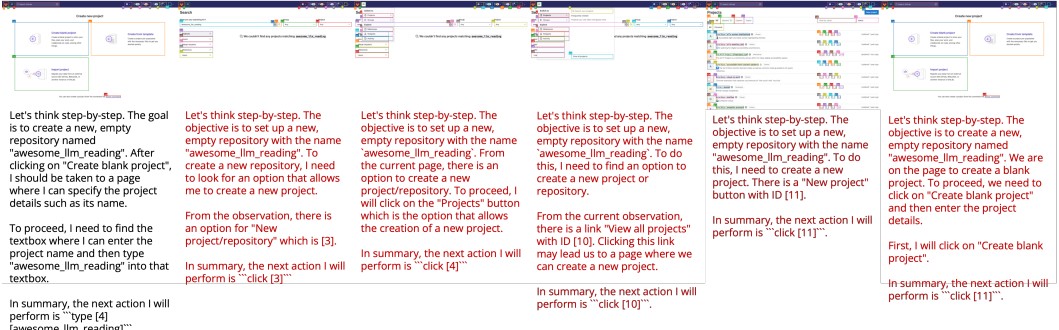

Figure 30: In WebArena-Lite, the agent initially searches for the target repository that need it to create. After realizing the problem, it struggles to return to the repository creating page to finally perform the correct action.

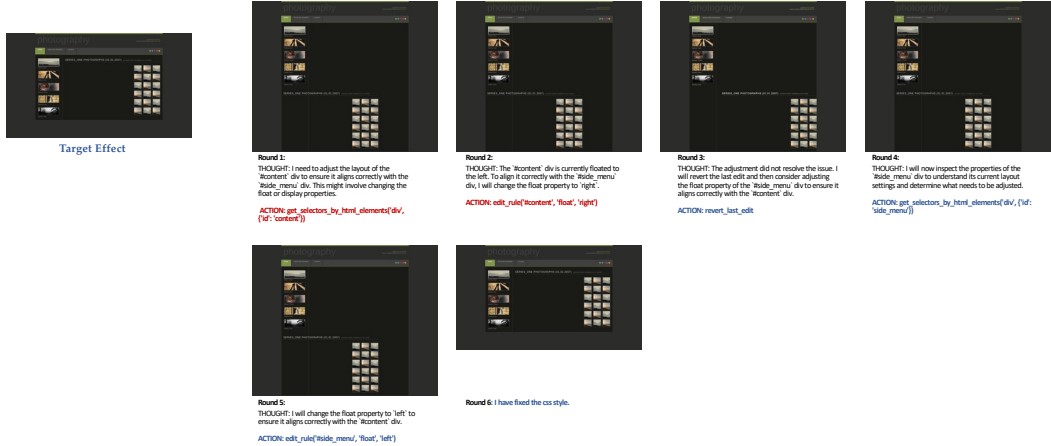

Figure 31: In VAB-CSS, the agent initially edits a wrong CSS rule. After rendering the new webpage, the agent realizes that this is not the correct change to make so it decides to revert the last edit and makes a new edit that finally fixes the layout.

