# OpenReview forum: "VisualAgentBench: Towards Large Multimodal Models as Visual Foundation Agents"
_ICLR.cc/2025/Conference — ICLR 2025 Poster_

### Official Review · Reviewer_CJyi · 2024-10-21

**Soundness:** 3
**Presentation:** 3
**Contribution:** 3
**Rating:** 6
**Confidence:** 4

**Summary:**

The authors propose a benchmark for evaluating LMMs across a wide range of task: including proposing (1) standardized interfaces, prompting, and data formats, (2) a strategy for creating valid test sets, (3) multitask multi-environment trajectory train sets, (4) benchmarking of 18 open-sourced and closed-sourced LMMs, (4) analyses of LMMs' abilities for grounding and planning.

**Strengths:**

I appreciate the diversity of tasks explored, from embodied agents to visual design. I believe that general-purpose LMMs that can perform well on a wide range of tasks are essential, and such benchmarks are necessary. Interaction through text/vision-based environment feedback is an important aspect of the paper. I also appreciate the scale of evaluation in the paper, and the finetuning of open LMMs.

**Weaknesses:**

1. Can you elaborate on how the task description, action spaces, few-shot demonstrations, and notices for each environment are formatted as prompts? How are visual outputs of the environment passed in as text in the interaction rounds for embodied and GUI agents? Could you elaborate on how error recovering works in planning? Interested in seeing experiments on error recovery behavior across all models, or in general, some evaluation on the interaction rounds instead of final success rate only (e.g., average number of steps to recover from an error).
2. I’m also interested in seeing more detailed analyses on the tasks that these models fail on. For example, which “prototypes” lead to failure? Does each model fail in a way that is consistent with one another (e.g., is there correlation between the accuracy on each prototype/subtask?) Does finetuning an open LMM help on the grounding aspect more, or the planning aspect more?
3. On a high-level, it would be great to have a quantitative metric established in this benchmark that systematically measures performance on grounding vs reasoning, instead of as an analysis only. Also, related to the first point, quantitative evaluation on interaction behavior (perhaps some proxy metric on progress to the goal).

**Questions:**

See above.

---

> ### Author Response · Authors · 2024-11-22
>
> Thanks for your thoughtful advice and review on VisualAgentBench! We really learn a lot from your comments. Here are our responses:
>
> 1. Details on Evaluation
>
> Thanks for your question. Our detailed prompts and few-shot examples (if used) are presented in Appendix B to F in the original submission. In the interaction rounds for embodied and GUI agents, output of environments as images are sent to models in the next round as the observations follow prior practices.
>
> 2. Analysis on Error Recovery Behaviors
>
> Thanks for your suggestion! According to your advice, we have updated detailed error recovery examples in the Appendix G.6. Basically, error recovery works in planning as an agent realizes the outcome of certain action has led to unexpected results. It usually involves two steps: return to the original state, and make another trial. We believe it is fundamental to agents due to their imperfect decision making (and so it is for us humans).
>
> Additionally, as you suggest we analyze the average steps needed for agents to recover from error to the correct directions (as below, based on proprietary gpt-4o and opened glm-4v, only for those finally successful tasks). We find that GUI tasks usually require more steps to recover, as the action spaces for them are very large (e.g., any clickable elements on the web pages). And fine-tuned glm-4v has shorter mean error recovery steps compared to gpt-4o, probably because it can only recover from simpler errors. Due to the limited time period of rebuttal, we will provide more analysis and statistics regarding error recovery in the final version.
>
> |        | VAB-OmniGibson | VAB-Minecraft | Webarena-Lite | VAB-AndroidLab | VAB-CSS |
> |--------|:----------:|:---------:|:--------:|:-------:|:---:|
> | gpt-4o |     2.5    |    3.3    |    6.0   |   8.3   | 2.6 |
> | glm-4v |     2.3    |    2.3    |    4.0   |   N/A   | 2.2 |
>
> 3. Detailed Analysis on Agent Error Modes
>
> Thanks for your comment. We first update example successful and failed trajectory case study in the Appendix G for readers’ intuitive understanding. In fact, we find agents to fail due to a variety of reasons, which can be a bit hard to comprehensively attribute in such a short period of response time. However, we endeavor to provide some statistics about major types of errors we observe, by sampling around 20 error traces in each environment for gpt-4o and internvl-2. More results and analysis will be presented in the final version of the paper.
>
> * Visual Grounding Error: Wrong detection or recognition of objects/elements in the visual observation.
> * Invalid Action: Outputting wrong formats of actions.
> * Loop: Agent repeats generating the same actions without quitting.
> * Task Limit Exceed: Agent does not accomplish the goal within reasonable maximum steps.
> * Hallucinated Task Completion: Agent makes wrong judgment on whether it has accomplished the task.
>
> |  gpt-4o  | visual grounding error | invalid action | loop | task limit exceed | hallucinated task completion |
> |:--------:|:----------------------:|:--------------:|:----:|:-----------------:|:----------------------------:|
> |    VAB-OmniGibson    |          0.30          |      0.04      | 0.17 |        0.17       |             0.30             |
> |    VAB-Minecraft    |          N/A         |      0.00      | 0.24 |        0.76       |             0.00             |
> | WebArena-Lite |          0.15          |      0.10      | 0.40 |        0.05       |             0.30             |
> |  VAB-AndroidLab |          0.10          |      0.00      | 0.65 |        0.15       |             0.10             |
> |    VAB-CSS   |           N/A          |        0.00       | 0.05 |        0.55       |             0.40             |
>
> | internvl-2 | visual grounding error | invalid action | loop | task limit exceed | hallucinated task completion |
> |:----------:|:----------------------:|:--------------:|:----:|:-----------------:|:----------------------------:|
> |     VAB-OmniGibson     |          0.00          |      0.00      | 0.25 |        0.50       |             0.25             |
> |     VAB-Minecraft     |          N/A          |      0.00      | 0.76 |        0.24       |             0.00             |
> |  WebArena-Lite  |          0.05          |      0.00      | 0.40 |        0.10       |             0.45             |
> |   VAB-AndroidLab  |          0.05          |      0.05      | 0.60 |        0.05       |             0.25             |
> |     VAB-CSS    |           N/A          |        0.00       | 0.45 |        0.30       |             0.25             |

---

> > ### Author Response · Authors · 2024-11-22
> >
> > 4. Questions About Grounding vs Reasoning
> >
> > Thanks for your question. Currently, it is actually a bit difficult to very clearly distinguish agents’ grounding and reasoning abilities in interactive evaluating benchmarks like VAB. For example, the evaluation of grounding usually requires a fixed plan and CoT thoughts. However, different models output different CoT thoughts that suit themselves best, but the difference in thoughts consequently makes the grounding comparison unfair. However, if we can set up a standard data format for grounding in the future, it might be possible to reasonably separate the evaluation for two abilities.
> >
> > 5. On Proxy Metric for Agent Progress to the Goal
> > Sure, for embodied and visual design problems in VAB (including 3 environments: VAB-OmniGibson, VAB-Minecraft, and VAB-CSS), we are setting up proxy metrics for evaluating progress of task completion.
> >
> > * VAB-OmniGibson: To complete a task, the LMM agent must achieve multiple subgoals (e.g., opening a specific door). Upon task termination, we compute the percentage of successfully achieved subgoals to provide an intermediate score.
> > * VAB-Minecraft: To acquire the goal item, the LMM agent must gather a series of items as ingredients. Consequently, we allocate intermediate scores to the agent as they collect these ingredients.
> > * VAB-CSS: To fix the CSS style to match the target screenshot, we can use screenshot similarity as a proxy metric for measuring progress of completion.
> >
> > Due to the short response period, we haven’t produced all model’s proxy metrics for progress. We will update them in the final version of the paper. For GUI tasks, since task goals are usually compound and can be solved in diverse routes, we find rule-based methods unsuitable for easy measuring of agent progress results. We will try to find new solutions to that.

---

> > > ### Comment · Reviewer_CJyi · 2024-11-25
> > > **Response**
> > >
> > > Thanks to the authors for the detailed analyses and clarifications. I maintain my positive score, and hope that if accepted the proxy metrics for progress will be added to the final version of the paper.

---

> > > > ### Author Response · Authors · 2024-12-01
> > > >
> > > > Dear Reviewer Cjyi,
> > > >
> > > > Thanks for your support of this work. Due to the extended response period, we managed to evaluate all proxy metrics of progress for the VAB-OmniGibson, VAB-Minecraft, and VAB-CSS as mentioned above. The results are as follow:
> > > >
> > > > | Model | OmniGibson | Minecraft | CSS |
> > > > |-------|------------|-----------|-----|
> > > > | gpt-4o-2024-05-13 | 62.6 | 61.9 | 46.7 |
> > > > | gpt-4-vision-preview | 58.4 | 55.9 | 38.2 |
> > > > | gpt-4-turbo-0409 | 50.6 | 59.7 | 37.6 |
> > > > | claude-3.5-sonnet-20240620 | 59.7 | 63.6 | 20.6 |
> > > > | claude-3-opus | 33.8 | 60.1 | 24.8 |
> > > > | gpt-4o-mini-2024-07-18 | 41.5 | 37.3 | 23.6 |
> > > > | gemini-1.5-pro | 46.3 | 49.4 | 13.9 |
> > > > | gemini-1.0-pro | 12.7 | 15.3 | 0.0 |
> > > > | qwen-vl-max | 1.3 | 9.0 | 3.0 |
> > > > | InternVL-2 | 41.6 | 35.5 | 32.7 |
> > > > | Qwen2-VL | 24.1 | 29.8 | 34.5 |
> > > > | GLM-4V | 32.2 | 29.4 | 29.7 |
> > > > | LLaVA-NeXT | 17.3 | 30.5 | 23.6 |
> > > > | CogVLM2 | 26.9 | 32.6 | 20.6 |
> > > > | CogAgent | 33.0 | 32.2 | 17.0 |
> > > > | CogVLM | 17.1 | 32.3 | 11.5 |
> > > > | LLaVA-1.5 | 14.9 | 33.5 | 9.1 |
> > > > | Qwen-VL | 18.8 | 26.5 | 4.8 |
> > > >
> > > > Hope these results will assure you of VAB’s contribution. We cannot thank you enough if you could raise your score to support us.

---

### Official Review · Reviewer_oNsV · 2024-11-01

**Soundness:** 3
**Presentation:** 3
**Contribution:** 3
**Rating:** 8
**Confidence:** 5

**Summary:**

This paper introduces a multimodal benchmark VisualAgentBench to evaluate LMMs as agents performing interactive tasks. The benchmark includes three scenarios based on five datasets/environments: Embodied (VAB-OmniGibson, VAB-Minecraft), Graphical User Interface (VAB-AndroidLab, VAB-WebArena-lite), and Visual Design (VAB-CSS). Besides benchmark data, the authors also provide additional task trajectories for training. All the task instances in the training and testing data are constructed by prototyping and instantiation. This paper applies a mix of three strategies to collect training trajectories according to the characteristics of different datasets. Experiment results show that the proposed benchmark is challenging for current LMMs and further SFT could improve the performance. The paper also conducts an analysis of visual grounding and planning to provide insights for the future development of LMM agents.

**Strengths:**

1. The proposed benchmark is a good complement to current multimodal and agent benchmarks to evaluate LMMs in challenging interactive scenarios.
2. The proposed benchmark has standardized environments with good consistency and reproducibility.
3. The paper also provides training trajectories for SFT.
4. The experiments are comprehensive, revealing problems of current models, and the analysis provides good insights.

**Weaknesses:**

1. The number of training data is still very limited. The paper does not show whether it is possible to scale up training data in the proposed environments in an efficient way.
2. There is no analysis and experiments to verify whether the proposed environments could be effectively used for RL training.
3. It would be helpful to train some non-LLM specialist models in each environment using RL/IL and report their performance as a reference.
4. After fine-tuning LMMs with the collected training data, authors should also evaluate their general multimodal abilities on other multimodal benchmarks. Also, the authors should explore whether it is possible to maintain the original abilities while improving the performance on the proposed benchmark after SFT.
5. The authors should provide some trajectories of models tested for better illustration.

**Questions:**

1. When training open LMMs, the paper says that they only use the vision input of the latest turn. Does this mean each turn in the training trajectory is treated as an independent sample (with previous turns provided in context) instead of training as a multi-turn conversation sample?
2. For evaluating LMM APIs, what do you mean by "Vision Input image won’t be kept in the dialog"? Wouldn't the images that appear in the previous turns in one conversation automatically be kept?

---

> ### Author Response · Authors · 2024-11-22
>
> Thanks for your thoughtful advice and encouragement on VisualAgentBench! We really learn a lot from your comments, and spare no efforts during this response stage to make improvements according to your questions. Here are our responses:
>
> 1. On Training Data and Subsequent Scaling Method
>
> Thanks for your advice. The initial purpose of providing training trajectory in VAB is to provide baseline data to include open LMMs into benchmarking, which performs too badly at instruction following without training. From experimental results, the goal is well accomplished as ability gaps between different LMMs’ on agents are distinguished after training.
>
> And indeed, we can scale up training data according to the synthetic methods we have described. The most probable one is LMM Agent Bootstrapping, from which we can bootstrap and filter high-quality new trajectories based on existing LMM agents. The challenge for the method lies in building a strong automatic filter, which we would like to leave for future work due to its complexity.
>
> 2. Verify and Conduct RL training
>
> Thanks for your great suggestion. Despite its feasibility to support RL, we do not intend to claim the support of it in the existing implementation of VAB. VAB is actually a very initial work in this field which aims to first set up the basic framework, tasks, and environments for future algorithmic study, such as RL.
>
> However, it is possible to use RL given our implementation, since VAB has provided interactive environments for online sampling. For example, recent works have shown the feasibility to run RL training for GUI agents in both Android Virtual Device and WebArena-Lite environments [1,2] similarly adopted in VAB. Therefore, VAB could offer an ideal testbed to foster RL algorithms for effective LMM agent training.
>
> 3. RL Specialist Training
>
> We agree that it is interesting to see if RL specialists could work in these environments. However, considering the implementation costs for customized training and architectural designs, we believe it is better to leave the idea for future study.
>
> 4. General Multimodal Abilities
>
> Thanks for your question. In VAB, for the purpose of benchmarking, we fine-tune LMMs merely on agent trajectories for simplicity. However, the maintenance of general multimodal abilities can ben addressed by mixing agent-domain with general-domain data, as indicated in literature [3,4].
>
> 5. Providing Example Trajectories
>
> We have uploaded 20 example trajectories (including both successes/failures and proprietary/open LMMs). Thanks for your advice.
>
> 6. Use of Vision Input in Evaluation
>
> Thanks for your question. We adopt the strategy because of two reasons: 1) most LMMs when the work was done only support single-image input. 2) Consider the long-horizon multi-turn interactions in agent tasks, it would introduce overlength context tokens if we keep all historical visual observation (i.e., images).
>
> However, the strategy is still multiturn interaction & conversation, since we only remove the historical visual content but leave other input and model-generated text contents in the context.
>
>
> References:
>
> [1] DigiRL: Training In-The-Wild Device-Control Agents with Autonomous Reinforcement Learning. NeurIPS 2024.
>
> [2] WebRL: Training LLM Web Agents via Self-Evolving Online Curriculum Reinforcement Learning. arXiv:2411.02337.
>
> [3] Agenttuning: Enabling generalized agent abilities for llms. Findings of ACL 2024.
>
> [4] Fireact: Toward language agent fine-tuning. arXiv:2310.05915.

---

> > ### Comment · Reviewer_oNsV · 2024-11-26
> > **Response to rebuttal**
> >
> > Thanks for the authors' responses. I will keep my original score.

---

### Official Review · Reviewer_cYwD · 2024-11-04

**Soundness:** 3
**Presentation:** 3
**Contribution:** 2
**Rating:** 3
**Confidence:** 5

**Summary:**

The authors propose to apply large multimodal models to visual agent tasks, which is mored grounded than existing VQA tasks.
The authors collected 5 datasets including 1 for robotics in simulation, 1 for game playing, 2 for GUI manipulation and 1 for web page design.
The authors design a function calling-based action space for each task.
The agent tasks are created by first generating a bunch of templates with placeholders and then instantiating these templates.
Action trajectories for each task are collected by 1) human-written scripts(for web GUI tasks) 2) prompting existing LMMs like GPT-4o 3) human demonstration.
The authors collected 746 test cases and 4482 training trajectories across 5 tasks and benchmarked 18 proprietary and open-source LMMs.

**Strengths:**

1. The proposed benchmark provides a unified “function-call” action space as well as diverse task domains for benchmarking the agent ability of LMMs
2. Experiments and ablation are solid. A wide range of both commercial and open-source LMMs are evaluated on the proposed benchmark. Ablations of input image prompting(labels, SoM), reflection(injecting error) and planner(ReAct w/ & w/o Thought) are conducted.

**Weaknesses:**

1. No clear advantage over existing benchmarks. There are plenty of existing benchmarks for both high-level planning for robot manipulation in simulation like RoboVQA as well as for GUI agent tasks like Webshop-V and VisualWebArena. The proposed visual agent benchmark is surely more grounded than VQA benchmarks like MMMU, but I don’t see what’s the real contribution is if compared with domain-specific benchmarks.
2. Low quality training trajectories. Take the GUI agent for instance, the proposed VAB-WebArene-Lite uses code script-based trajectory collection, which is well-known for its limited diversity compared with real-world human web browsing action trajectories.
3. The function calling action space biases toward LMMs with a strong coding ability so that some LMMs unfairly got low scores(like Qwen-VL-Max and Gemini-1.0-Pro, which both do a good job for pure VQA benchmarks.

**Questions:**

1. Does the physical dimension of robot arm given in Gibson? As shown in Figure 1 Round 3, I don't think the grasp banana is feasible given its lying on the far end of the countertop

---

> ### Author Response · Authors · 2024-11-22
>
> Thanks for your thoughtful advice and review on VisualAgentBench! We really learn a lot from your comments. Here are our responses:
>
> 1. Advantages and Comparison to Domain-specific Benchmarks
>
> Thanks for your comment. The interactive environment for all tasks in VAB is a crucial advantage, which clearly distinguishes it from many existing benchmarks. For example, the RoboVQA is a video-based static planning dataset against prepared trajectories. Its static nature prevents it from reflecting agents’ real performances in interactive real-world environments, where agents could present typical behaviors of exploring and error-recovering that are vital but only seen in interactive evaluation as VAB-OmniGibson does. In addition, VAB-AndroidLab and VAB-CSS environments are also among the first to provide interactive evaluation in their specific domains. VAB-Minecraft is the first to provide a publicly available evaluation set for agent evaluation (while previous Minecraft works fail to open-source their evaluation data). For web tasks, it is certain that the VisualWebArena and WebArena are quite good, so we only do some refinements to them in VAB (e.g., fix problematic judging functions).
>
> More importantly, the emphasis of VAB is the comprehensive benchmarking across multiple environments for interactive LMM agent evaluation. In the context of LLMs and LMMs, researchers wish to evaluate them across multi-domain datasets to verify their generalizability, which is a key to these models. However, existing domain-specific benchmarks fall short to satisfying the need due to their design features. Thus one of the VAB’s unique values lies in its LMM-oriented design and comprehensive evaluation sets.
>
> 2. On Quality of Training Trajectories
>
> Thanks for your comment. In fact, except for WebArena and OmniGibson, training trajectories for all other 3 environments are either human-annotated or LMM-bootstrapped to ensure diversity. But in fact, the adoption of program-based trajectory collection, while sacrificing some task diversity (alleviated by task instruction rewriting), actually improves the overall quality in boost of data accuracy as we have discussed in Section 3.2. Take the web data you mentioned as an example. Human-collected web browsing trajectories often exhibit some problematic features that could harm trained LMM performances. For instance, human trajectories usually present repeated scrolling ups and downs for information seeking. LMMs trained on such data would present patterns to repeat scrolling actions without knowing when to stop. Program-based collection avoids the problem by eliminating such up and down loops.
>
> Additionally, the training trajectories provided in VAB serve as a foundation for benchmarking open LMMs, which otherwise struggle significantly with instruction following without training. Experimental results demonstrate that this goal is successfully achieved, as training clearly distinguishes the capabilities of different LMMs when applied to agents. Overall, we position VAB not as a finalized solution, but as a robust starting point for the community to build upon and synthesize more effective data for agent training. We hope VAB will pave the way for further advancements.
>
>
>
> 3. Function-calling for Agent Tasks
>
> Function-calling capability is a core requirement for language agents and a necessary condition for enabling them to take actions across different environments. To clarify, our ultimate goal is to advance LMMs into generalist agents capable of operating in diverse domains, each characterized by a unique set of actions (i.e., functions). For instance, in our five environments, each has its own distinct action space. To effectively instruct an LMM to complete tasks in a specific domain, it must be able to invoke the appropriate functions (actions) based on the environment’s specifications.
> Therefore, VAB is not “biased” toward models with strong function-calling capabilities. Instead, achieving better performance through robust function-calling is an inherently desirable property for generalist agents
>
>
>
>
>
> 4. Physical Dimension of Robot Arms
>
> In VAB-OmniGibson, we allow the LMM agent to grasp objects within a 1.2-meter radius. In Figure 1 Round 3, the banana is within this range, so the LMM agent can successfully grasp it.

---

> ### Comment · Reviewer_cYwD · 2024-11-29
>
> 1. I still don't get the rationale behind why these tasks are chosen. The robotics, Minecraft, UI and webpage design are good domains but are not the only domains that matter. Why not include driving, drone and flat design, etc as well? Since VAB is not an exhaustive benchmark, how to choose each task is crucial. Also I don't know what does a comprehensive benchmark mean by the authors in the rebuttal. Does that mean if a VLM could do well in the VAB benchmark then it can do well in most cases?
> 2. Neither human-annotated nor LMM-bootstrapped ensures a high diversity. Only the careful task space design could achieve that goal. Maybe the authors could elaborate more about how the task lists are carefully crafted to maximize the diversity for each domain?
> 3. I don't understand why a model without function calling ability could not be a good VLM agentic model across different environments. I think Qwen-VL 1 is quite solid a VLM and could do well in many driving, robotics and UI tasks when fine-tuned as a VLA model without any function calling ability.
> 4. This is simply wrong, most robot arms like UR5 or franka have a workspace of 700mm-1000mm, if you allow the VLM to grasp objects within 1.2m there will be tons of failures. This makes me wonder if the VAB-OmniGibson really executed the action of VLM?
>
> I think the authors fail to address most of my concerns and decide to downgrade my rating.

---

> > ### Author Response · Authors · 2024-11-30
> >
> > We sincerely apologize that you felt our previous response did not adequately address your concerns. Your thoughtful feedback has helped us identify important areas that need clarification and improvement. We strive to provide a more comprehensive response to each of your points.
> >
> > To provide a general context, VAB primarily focuses on agents’ high-level planning and reasoning (as stated in Section 2 and Appendix A.1), which is the predominant focus of the LLM agent community [1-3]. The planning of LLM and LMM agents refers to their ability to understand compound instructions, make plans, and adjust plans when they meet obstacles. It differs from the research goal of the VLA community, which emphasizes low-level controlling of robots (we also recognize its great importance, but it is not the scope of VAB focusing on).
> >
> > **1. Rationale behind VAB benchmarks**
> >
> > We are very grateful for your insightful questions, which have helped us realize we need to provide better motivation and background in our paper. Please allow us to address your questions in detail:
> >
> > * Why not include driving, drone and flat design: VAB is our initial attempt to establish a multitask benchmark for evaluating visual agents based on LMMs. The tasks we selected, such as robotics [4], Minecraft [5-6], UI [7] and webpage design [8], have appeared in related LLM and LMM literature but lacked standardized evaluation frameworks. Your suggestion about including driving, drone and flat design highlights an important opportunity for improvement. We must acknowledge that our expertise is primarily in LLMs and LMMs. We would be very grateful for your guidance on how to thoughtfully incorporate these environments in future versions of VAB.
> >
> > * What does a comprehensive benchmark mean: Thank you for the opportunity to clarify our terminology. The concept of "comprehensive" stems from the foundation model community's perspective that these models should demonstrate broad task generalization. For example, MMLU [9] comprehensively evaluates LLMs across 57 disciplines to assess their knowledge breadth. It has been proved as an excellent proxy for LLM’s general capabilities and being adopted by OpenAI, Anthropic, Google, and so on. Similarly, VAB aims to serve as a comprehensive benchmark for LMMs' capabilities as visual agents. Regarding your important question about whether success in VAB indicates broader success - yes, that is our answer.

---

> > > ### Author Response · Authors · 2024-11-30
> > >
> > > **2. To Ensure High Diversity of Training Data**
> > >
> > > Thank you for this valuable suggestion. The question can be discussed from two perspectives, for which our original response focuses on the dimension of "solution-path diversity". It refers to the diversity of ways to achieve a goal. Human annotation and LMM-bootstrapping, compared to program-based methods, naturally introduce the randomness and some back-and-forth explore patterns, and can therefore enhance the diversity of solution paths.
> > >
> > > The second dimension regarding the design of task space is indeed another crucial point you've raised. Based on the "prototyping-instantiation" paradigm we adopt, we have undertaken extensive efforts to ensure the task space diversity by enriching scenes, task prototypes, and diversity in expressions:
> > >
> > > * VAB-OmniGibson: Within the household environment, we incorporate as many diverse scenes as possible. The test set features 19 distinct scenes, challenging the agent to navigate across multiple rooms and adapt to various situations in different surroundings. We also design a wide array of activities to evaluate the agent, including 181 tasks. These tasks require the agent to utilize multiple action types within the household scenario, including navigation, transporting objects, and altering object states.
> > >
> > > * VAB-Minecraft: Minecraft is considered a classic scenario for embodied agents in the gaming environment, and we include a diverse range of 116 tasks. These tasks span 6 material levels within the game and require diverse kinds of raw ingredients (11 plants, 4 animals, 6 hostile mobs). Also, the agent must interact within a variety of terrains, such as forests, plains, deserts, and caves. This level of diversity is a significant test of the agent's ability to adapt and engage with the game's environment.
> > >
> > > * VAB-WebArena-Lite: To facilitate the task diversity, we manually created up to 98 different task prototypes (CMS: 23, Reddit: 17, Map: 13, OSS: 24, GitLab: 21) and corresponding automatic programs to solve them. For each task prototype, we would fill in grounded entities we collected from the websites (e.g., order numbers, repo / user / product names, forums) to ensure the task is executable in the environment. The filled task prototypes are then rewritten by LLMs to change expressions and wordings to ensure diversity.
> > >
> > > * VAB-AndroidLab: Similar to practices in WebArena-Lite, we create 70 task prototypes from 18 common apps for training (stated in Appendix D). Task prototypes are rewritten by LLMs to enhance diversity in expressions. Since data is annotated by humans, we allow some degree of freedom for annotators to customize these instructions in their practical labeling to ensure task completeness.
> > >
> > > * VAB-CSS: For CSS, there are two major aspects for the task space, i.e., the diversity of the target websites and the diversity of the type of bugs (i.e., corruptions) for the agent to fix. For the diversity of websites, we cover 994 different website templates from different domains. For types of corruption, we also introduce diverse operations, such as adding, removing, or modifying a property, or removing an entire rule. This ensures comprehensive coverage of diverse scenarios within the domain of CSS bug fixing.

---

> > > > ### Author Response · Authors · 2024-11-30
> > > >
> > > > **3. VLMs' Function Call Ability**
> > > >
> > > > We appreciate your valuable perspective regarding Qwen-VL's capabilities. Our analysis of fine-tuned open LMMs (examining about 20 failed tasks per environment) shows that fine-tuning effectively eliminates invalid action errors. This suggests our evaluation of open LMMs remains valid:
> > > >
> > > > | internvl-2 | visual grounding error | invalid action | loop | task limit exceed | hallucinated task completion |
> > > > |:----------:|:----------------------:|:--------------:|:----:|:-----------------:|:----------------------------:|
> > > > |     VAB-OmniGibson     |          0.00          |      0.00      | 0.25 |        0.50       |             0.25             |
> > > > |     VAB-Minecraft     |          N/A          |      0.00      | 0.76 |        0.24       |             0.00             |
> > > > |  WebArena-Lite  |          0.05          |      0.00      | 0.40 |        0.10       |             0.45             |
> > > > |   VAB-AndroidLab  |          0.05          |      0.05      | 0.60 |        0.05       |             0.25             |
> > > > |     VAB-CSS   |           N/A          |        0.00       | 0.45 |        0.30       |             0.25             |
> > > >
> > > > Tablenotes:
> > > > * Visual Grounding Error: Wrong detection or recognition of objects/elements in the visual observation.
> > > > * Invalid Action: Outputting wrong formats of actions.
> > > > * Loop: Agent repeats generating the same actions without quitting.
> > > > * Task Limit Exceed: Agent does not accomplish the goal within reasonable maximum steps.
> > > > * Hallucinated Task Completion: Agent makes wrong judgment on whether it has accomplished the task.
> > > >
> > > > However, we emphasize VLMs' function calling ability (particularly for proprietary LMMs) because environments have distinctly different action spaces (here, what we mean by "functions", is those valid "actions" for agents in environments, but represented in the form of code to follow practices in prior LLM and LMM study). Fine-tuning would also reduce the model's generalizability to work well in other environments. Compared to existing open LMMs, very strong LMMs like gpt-4o do not need to be fine-tuned but can still make few mistakes in calling correct functions in different environments by merely giving these functions' descriptions in "system prompt" (Cf. Appendix B to F, where we document all of our system prompts used for calling proprietary LMMs).
> > > >
> > > > Such generalizable ability is the key that we are pursuing, which refers to the ability of LMMs to work in any environment by giving only those system prompts, just as we would prompt ChatGPT to write for us in our daily life. But open LMMs like intern2-vl and qwen-vl currently fail to follow those "system prompt" without fine-tuning (in our preliminary experiment, basically they all fail to follow any function call formats in system prompts and thus fail to have a success rate greater than 0). As a result, that is why we would suggest the community to improve LMMs' function calling so as to facilitate the development of generalized LMM agents.

---

> ### Author Response · Authors · 2024-11-30
>
> **4. Regarding the workspace of robot arms in OmniGibson**
>
> We sincerely thank you for this important technical complement about robot arm workspaces. Given our previously mentioned goal and context of VAB, in OmniGibson we make a simplification: as long as the agent is within the radius of the target object, we would recognize the grasping as successful. The 1.2m we originally chose, is a loose median value after our surveying (we must acknowledge our limited expertise in robotics). We did observe some robot arms with working radius up to 1.3-1.7m [10-11].
>
> Following your recommendation, we tested gpt-4o with the 1000mm radius. The results show:
>
> | gpt-4o | VAB-OmniGibson |
> |--------|----------------|
> | 1200mm | 41.4           |
> | 1000mm | 38.1           |
>
> While this adjustment slightly affects performance, it's unlikely to significantly alter VAB's main conclusions. We are committed to conducting a complete re-evaluation with this more accurate parameter.
>
> We truly value your thorough review and expert insights, which have helped us identify important areas for improvement. Please don't hesitate to raise any additional concerns - your guidance is essential for strengthening this research.
>
> Reference:
>
> [1] Yao, Shunyu, Jeffrey Zhao, Dian Yu, Nan Du, Izhak Shafran, Karthik R. Narasimhan, and Yuan Cao. "ReAct: Synergizing Reasoning and Acting in Language Models." ICLR 2022.
>
> [2] Park, Joon Sung, Joseph O'Brien, Carrie Jun Cai, Meredith Ringel Morris, Percy Liang, and Michael S. Bernstein. "Generative agents: Interactive simulacra of human behavior." UIST 2023.
>
> [3] "AgentBench: Evaluating LLMs as Agents." ICLR 2023.
>
> [4] Yang, Jingkang, Yuhao Dong, Shuai Liu, Bo Li, Ziyue Wang, Haoran Tan, Chencheng Jiang et al. "Octopus: Embodied vision-language programmer from environmental feedback." ECCV 2025
>
> [5] Wang, Guanzhi, Yuqi Xie, Yunfan Jiang, Ajay Mandlekar, Chaowei Xiao, Yuke Zhu, Linxi Fan, and Anima Anandkumar. "Voyager: An open-ended embodied agent with large language models." TMLR 2024
>
> [6] Zhu, Xizhou, Yuntao Chen, Hao Tian, Chenxin Tao, Weijie Su, Chenyu Yang, Gao Huang et al. "Ghost in the minecraft: Generally capable agents for open-world environments via large language models with text-based knowledge and memory." arXiv preprint arXiv:2305.17144 (2023).
>
> [7] Hong, Wenyi, Weihan Wang, Qingsong Lv, Jiazheng Xu, Wenmeng Yu, Junhui Ji, Yan Wang et al. "Cogagent: A visual language model for gui agents." In Proceedings of the IEEE/CVF Conference on Computer Vision and Pattern Recognition, pp. 14281-14290. 2024.
>
> [8] Si, Chenglei, Yanzhe Zhang, Zhengyuan Yang, Ruibo Liu, and Diyi Yang. "Design2Code: How Far Are We From Automating Front-End Engineering?." arXiv preprint arXiv:2403.03163 (2024).
>
> [9] Hendrycks, Dan, Collin Burns, Steven Basart, Andy Zou, Mantas Mazeika, Dawn Song, and Jacob Steinhardt. "Measuring Massive Multitask Language Understanding." ICLR 2021
>
> [10] UR10: https://www.rarukautomation.com/collaborative-robots/ur-6-axis-collaborative-robots/ur10-robot/
>
> [11] CR20A: https://www.dobot-robots.com/products/cra-series/cr20a.html

---

> > ### Comment · Reviewer_cYwD · 2024-12-01
> >
> > I am totally confused by the practice of "as long as the agent is within the radius of the target object, we would recognize the grasping as successful".
> >
> > As in the initial response, the authors highlighted that "For example, the RoboVQA is a video-based static planning dataset against prepared trajectories. Its static nature prevents it from reflecting agents’ real performances in interactive real-world environments, where agents could present typical behaviors of exploring and error-recovering that are vital but only seen in interactive evaluation as VAB-OmniGibson does."
> >
> > If you just don't execute any real action of the robot, what's the difference of having a simulation environment instead of static dataset since all your execution simply succeed.
> >
> > Also as far as I know, none of the stock robots(fetch, tiago, stretch and R1) provided in OmniGibson has a workspace of 1000mm. I assume you reuse most scene and robot assets in OmniGibson, so the construction and implementation of VAB-OmniGibson is technically flawed.

---

> > > ### Author Response · Authors · 2024-12-01
> > >
> > > Thank you for your prompt and detailed review. We greatly appreciate your feedback and would like to address your questions.
> > >
> > > **1. The value of simulation environments compared to static datasets**
> > >
> > > We would like to elaborate on two key advantages that interactive environments offer over static datasets when evaluating agents' high-level planning capabilities:
> > >
> > > * Solution path flexibility: Consider a simple example where an agent needs to collect two apples, X and Y. Both sequences—collecting X then Y, or Y then X—represent valid solutions. In practice, tasks often have multiple valid approaches. Static trajectory datasets typically capture only a limited subset of possible solutions, which may lead to incorrectly penalizing other valid approaches. Interactive environments, on the other hand, can evaluate success based on final outcomes (such as whether both apples are ultimately collected), providing a more comprehensive assessment of planning capabilities.
> > >
> > > * Error recovery assessment: The ability to recover from mistakes is fundamental to effective planning, both for agents and humans. For instance, when traveling from point A to C, an agent might initially move to point B, recognize the error, return to A, and then correctly proceed to C. Static datasets have inherent limitations in evaluating such recovery behaviors. As demonstrated in Figure 6 of our paper, these recovery patterns appear frequently in VAB benchmarking results, highlighting the importance of interactive environments for comprehensive planning evaluation.
> > >
> > > **2. The implementation of grasp action in OmniGibson**
> > >
> > > Thanks for your question. It has actually been a common practice to simplify the process implementation of grasp actions and keep its final effect when benchmarking household agents’ high-level planning abilities, as is reported in Alfred [1],  Octopus [2], and Behavior-1K [3]. Specifically, the Behavior-1K [3] paper (also the one to propose OmniGibson environment) also adopts the high-level action primitive to simplify the evaluation for agent’s planning. We quote their explanation here:
> > >
> > > > Grasping is a challenging research topic on its own…… to accelerate training, the action primitives check only the feasibility (e.g., reachability, collisions) of the final configuration, e.g. the grasping pose for pick or the desired location for navigate. If kinematically feasible, the action primitives will directly set the robot state to the final configuration, and continue to simulate from then on.
> > >
> > > As a result, we follow these prior practices to only check the feasibility of grasping when it is performed. To be specific, similar to Octopus [2], we check the feasibility of grasping using following rules:
> > >
> > > * Whether the agent has nothing on “hand”: if not, the grasping would fail
> > > * Whether the object is within the operating radius: if not, the grasping would fail
> > > * Whether the object is movable: if not, the grasping would fail
> > > * Otherwise, the grasping would be successful and the object would be grasped by the agent’s “hand”
> > >
> > > Simplified though it is, such configuration is sufficient for benchmarking the high-level planning abilities of agents and could accelerate the whole evaluation process (or the simulation would be slow).
> > >
> > > Regarding our robot configuration, we are using the default “fetch”. However, as we mentioned above, since the benchmarking of agent’s high-level planning does not engage the exact physical grasping, our evaluation does not rely on a real robot configuration. Additionally, in preliminary study, we find that adopting a small radius standardly provided would simply make most of common tasks infeasible due to too poor object reachability in the OmniGibson environment. Given the focus of VAB-OmniGibson is to evaluate agent’s high-level planning behaviors, we decided to relax the constraint to 1.2m. Similar practices are also observed in literature. For example, Octopus [2] adopts a radius of 2.0m for grasping in the OmniGibson environment.
> > >
> > > Generally, our position is that the high-level planning problem is a unique challenge that deserves benchmarking. We do agree on the importance of low-level controlling, but it is surely a challenging research problem on its own. And to facilitate the study of high-level planning, we suppose it might be better to first relax the constraint of low-level controlling to allow relatively disentangled studying on the high-level planning problem.
> > >
> > > Except for these questions, we hope our previous response has mitigated other concerns of yours. We really learn a lot from your review and suggestions, which would definitely make VAB a better work. Certainly, the study to join foundation models with agent tasks is really challenging and interdisciplinary. Many concepts and thoughts still need to be aligned across communities, but we believe VAB would be an imperfect but acceptable starting point. We really need your support of VAB to together advance research in this field.

---

> > > > ### Author Response · Authors · 2024-12-01
> > > >
> > > > References:
> > > >
> > > > [1] Shridhar, Mohit, Jesse Thomason, Daniel Gordon, Yonatan Bisk, Winson Han, Roozbeh Mottaghi, Luke Zettlemoyer, and Dieter Fox. "Alfred: A benchmark for interpreting grounded instructions for everyday tasks." CVPR 2020.
> > > >
> > > > [2] Yang, Jingkang, Yuhao Dong, Shuai Liu, Bo Li, Ziyue Wang, Haoran Tan, Chencheng Jiang et al. "Octopus: Embodied vision-language programmer from environmental feedback." ECCV 2024.
> > > >
> > > > [3] Li, Chengshu, Ruohan Zhang, Josiah Wong, Cem Gokmen, Sanjana Srivastava, Roberto Martín-Martín, Chen Wang et al. "Behavior-1k: A benchmark for embodied ai with 1,000 everyday activities and realistic simulation." CoRL 2022.

---

> > > > > ### Comment · Reviewer_cYwD · 2024-12-03
> > > > >
> > > > > Generally, I think a good agent benchmark must allow the VLM to interact with an environment (real or simulated) to test its ability to recover from errors in a closed-loop manner.
> > > > > The closed-loop evaluation is the most important part of the benchmark compared to a static environment where you can simply use an LLM to determine if a high-level planning makes sense in an open-loop manner.
> > > > > However, VAB-OmniGibson violates this golden design rule by simplifying real grasping trials into a rule-based text environment instead(if the robot has hand and is within 1.2m then grasp is success).
> > > > > Following the simpler implementation approach like previous works is tempting, but I think the authors should rethink the rationale behind constructing this benchmark and what they want to contribute to the community.

---

### Official Review · Reviewer_4jCV · 2024-11-08

**Soundness:** 3
**Presentation:** 3
**Contribution:** 3
**Rating:** 6
**Confidence:** 4

**Summary:**

The paper introduces VisualAgentBench (VAB), a benchmark designed to evaluate and train LMMs as visual agents in diverse, realistic scenarios, including embodied, GUI, and visual design tasks. VAB provides a unified, standardized framework for assessing LMMs across multiple domains, synthesizes high-quality multimodal data using a mix of programmatic solvers, LMM bootstrapping, and human demonstrations, and benchmarks 18 LMMs, uncovering both strengths and limitations in real-world task performance. Key insights include challenges in visual grounding, planning, and error recovery, offering a valuable testbed to push LMMs toward more adaptable and practical visual agents.

**Strengths:**

- Comprehensiveness: The main strength of this paper is its comprehensiveness in benchmarking Large Multimodal Models (LMMs) as visual agents. The authors introduce VisualAgentBench (VAB), a benchmark that covers a wide range of real-world application scenarios by including five major task categories: embodied agents, GUI-based agents, web agents, gaming agents, and visual design agents. This breadth makes VAB a thorough evaluation tool, enabling a more holistic assessment of LMMs' capabilities across different domains rather than focusing on a single application area.

- Extensive Experiments: The paper demonstrates substantial experimental rigor by benchmarking 18 different LMMs, encompassing both proprietary and open-source models. This extensive testing provides a solid foundation for the insights presented, which shed light on various LMM challenges, such as visual grounding and error recovery. These experiments allow for more reliable comparisons between models, offering valuable insights into how different LMMs perform in complex, interactive tasks. The conclusion on ReAct framework is also interesting.

- Insightful Analysis: Through the VAB benchmark, the authors provide some useful observations on the current state of LMMs as visual agents. They highlight specific limitations in visual grounding, action planning, and error handling across various environments, which helps to pinpoint areas for future improvement in LMM design. While these insights are not groundbreaking, they add value by identifying practical challenges that developers and researchers may encounter when deploying LMMs in real-world applications.

**Weaknesses:**

- Insufficient Explanation for VL Model Performance: Some vision-language models perform poorly without adequate explanation. For instance, the paper doesn’t explore why certain models achieved low scores, leaving questions about the benchmark’s application across models.
- Unclear Role of Visual Information in Certain Tasks: The paper lacks clarity on how specific tasks, such as those in Minecraft, leverage visual information effectively and whether VLM is genuinely necessary for all actions. For instance, Minecraft actions like "Teleport" don't inherently require visual information since they can execute without reference to the visual state, raising doubts about the added value of VL models in such contexts. Clarifying how the benchmark ensures each action necessitates visual input, as opposed to pure language model decision-making, would help demonstrate the benchmark’s relevance and justify the use of VL models over text-only approaches in specific environments.
- Ambiguities in Figure Interpretation and Process Flow: Figures like Figure 2 could benefit from clearer annotations or explanations. The figure includes multiple input and output connections but lacks a clear process flow or indication of sequential dependencies, making it challenging to follow the intended agent behavior across rounds.

**Questions:**

1. Could you clarify the role of bounding boxes and object tags in Figure 7? Does this mean that objects and tags must be visible in the input images so that the simulator can recognize and interact with these objects by their tag names? In Section 5.1, the authors discuss the use of object labels in embodied environments. How exactly does the agent operate when no object label or tag is provided?

2. To ensure ease of use, what practice does VAB provide? unified API access or modular code structure across different task environments? More details on engineering side for easy usage could be beneficial.

---

> ### Author Response · Authors · 2024-11-22
>
> Thanks for your thoughtful advice and review on VisualAgentBench! We really learn a lot from your comments, and spare no efforts during this response stage to make improvements according to your questions. Here are our responses:
>
> 1. More Explanation and Analysis on Poor-Performed LMMs
>
> Thanks for your comment. We first update example successful and failed trajectory case study in the Appendix G for readers’ intuitive understanding. In fact, we find agents to fail due to a variety of reasons, which can be a bit hard to comprehensively attribute in such a short period of response time. However, we endeavor to provide some statistics about major types of errors we observe, by sampling around 20 error traces in each environment for gpt-4o and internvl-2. More results and analysis will be presented in the final version of the paper.
>
> * Visual Grounding Error: Wrong detection or recognition of objects/elements in the visual observation.
> * Invalid Action: Outputting wrong formats of actions.
> * Loop: Agent repeats generating the same actions without quitting.
> * Task Limit Exceed: Agent does not accomplish the goal within reasonable maximum steps.
> * Hallucinated Task Completion: Agent makes wrong judgment on whether it has accomplished the task.
>
> |  gpt-4o  | visual grounding error | invalid action | loop | task limit exceed | hallucinated task completion |
> |:--------:|:----------------------:|:--------------:|:----:|:-----------------:|:----------------------------:|
> |    VAB-OmniGibson    |          0.30          |      0.04      | 0.17 |        0.17       |             0.30             |
> |    VAB-Minecraft    |          N/A         |      0.00      | 0.24 |        0.76       |             0.00             |
> | WebArena-Lite |          0.15          |      0.10      | 0.40 |        0.05       |             0.30             |
> |  VAB-AndroidLab |          0.10          |      0.00      | 0.65 |        0.15       |             0.10             |
> |    VAB-CSS   |           N/A          |        0.00       | 0.05 |        0.55       |             0.40             |
>
> | internvl-2 | visual grounding error | invalid action | loop | task limit exceed | hallucinated task completion |
> |:----------:|:----------------------:|:--------------:|:----:|:-----------------:|:----------------------------:|
> |     VAB-OmniGibson     |          0.00          |      0.00      | 0.25 |        0.50       |             0.25             |
> |     VAB-Minecraft     |          N/A          |      0.00      | 0.76 |        0.24       |             0.00             |
> |  WebArena-Lite  |          0.05          |      0.00      | 0.40 |        0.10       |             0.45             |
> |   VAB-AndroidLab  |          0.05          |      0.05      | 0.60 |        0.05       |             0.25             |
> |     VAB-CSS    |           N/A          |        0.00       | 0.45 |        0.30       |             0.25             |
>
>
> 2. Clarification on Role of Visual Information
>
>  In VAB, visual information is indispensable for all tasks. We provide a detailed explanation in Appendix A.2 to address your concern.
> In most cases, the agent must rely on visual input to determine the affordances for its actions, such as identifying objects to interact with in a room or buttons to click on a website. For your example of “teleport,” you are correct that this represents a rare case of a global action available at any time. However, deciding whether it is reasonable to use this action requires the agent to recognize that it is trapped in certain locations, which can only be inferred using visual information, specifically the last several frames of gameplay.
>
>
>
> 3. On Clearer Annotations for Figure 2
>
> Thanks for your comment. We have updated more annotations in the figure and the corresponding caption.

---

> > ### Author Response · Authors · 2024-11-22
> >
> > 4. Role of Bounding Boxes and Object Tags in Figure 7
> >
> > The core challenge here lies in enabling the LMM to unambiguously specify the target object to operate on based on the current scene (thus  vision-centric). Generally, there are two potential solutions. The first involves asking the LMM to output a coordinate in the input image to locate the target object. However, this can be difficult without fine-tuning the LMM on a substantial amount of coordinate output data. The second, more commonly adopted method, is known as the Set of Marks (SoM). In this approach, the input image is annotated with a set of bounding boxes that highlight the objects within it. The LMM only needs to output the ID of the appropriate bounding box to unambiguously identify the target object.
> >
> >
> > In VAB-OmniGibson, all objects visible in the input image are annotated with both a bounding box and a corresponding tag name (e.g., "1.soda", "7.bed"). This annotation design enables LMM agents to precisely reference objects using their tag names. Only visible objects with bounding boxes (task-relevant objects) are operable in the simulator.
> >
> > In Section 5.1, we evaluate a comparative setting where objects are annotated only with bounding boxes and numerical indices (e.g., "1", "7", as illustrated in Figure 4), without object tag names. In this setting, LMM agents must reference objects by their indices, requiring them to accurately recognize the objects within the bounding boxes through visual understanding.
> >
> > 5. Details on VAB Engineering Side Effort To Enable Easy Use
> >
> > Thanks for your suggestions. By basing on the AgentBench framework, VAB enables flexible customization of new tasks, models and improves efficiency via several important means. Some of them are as follow:
> >
> > * Docker-based Environments:  VAB packs up environments as dockers to simplify the deployment for parallel evaluation. It inherently supports simultaneous environments serving to accelerate the benchmarking, cutting down evaluating time compared to single-server by 80%.
> > * Server and Client Architecture: To enable parallel evaluation, the whole framework is implemented in a server-client architecture, where all interactions are implemented based on HTTP protocols.
> > * Network Flow Assignment: Since it takes varied time to evaluate LMMs in different tasks, Edmonds-Karp based Max-flow algorithm is implemented to dynamically assign agents to proprietary LMM APIs (or deployed open LMMs) with limited concurrency to maximize overall evaluation time.
> >
> > More details will be updated in the next version of the paper. Thanks again for your advice!

---

> > > ### Comment · Reviewer_4jCV · 2024-11-29
> > >
> > > Thank you for the response and I am happy with the response.
> > > I checked others' reviews.
> > > Although there could be some weaknesses as R-cYwD mentioned - the paper could indeed benchmark more applications and the significance could be weakened by loose precision, I still think the paper makes good contributions to the community - if the environment is really easy to use.

---

> > > > ### Author Response · Authors · 2024-11-30
> > > >
> > > > Thank you for your feedback and for recognizing the updates and technical discussions in our rebuttal. Your constructive comments have been invaluable in improving our work, and we cannot thank you enough if you could raise your score to support us.

---

### Author Response · Authors · 2024-12-04
**General Response to All Reviewers**

We would like to express our sincere gratitude to all the reviewers for their thoughtful and constructive feedback on our work. The insights and suggestions provided have been invaluable in helping us refine and strengthen our submission.

Multiple reviewers appreciated our benchmark for setting up diverse LMM agent application scenarios (Reviewers 4jCV, cYwD, CJyi), and extensively evaluating a wide range of 18 proprietary and open-source LMMs (4jCV, cYwD). Reviewers also highlighted the provided training trajectories for SFT open-source LMMs (oNsV, CJyi). The comprehensive and insightful analysis on planning and visual grounding are especially noted (4jCV, cYwD, oNsV).

In response to Reviewer 4jCV and CJyi's suggestion for a more detailed analysis of agent error modes, we have included statistics on the major types of errors for gpt-4o and internvl-2 in the rebuttal discussion, along with examples of agent trajectories in Appendix G. Additionally, we have provided an elaborated statistic on error recovery behaviors and proxy metrics of progress in VAB-OmniGibson, VAB-Minecraft, and VAB-CSS, as advised by Reviewer CJyi.

We are sincerely grateful that Reviewer 4jCV, oNsV and CJyi support our work and are generally satisfied with our response. Our primary point of disagreement with Reviewer cYwD pertains to the action space within VAB-OmniGibson. While Reviewer cYwD advocates for a physically low-level interaction with the household environment to ensure closed-loop evaluation, we emphasize that VisualAgentBench is focused on high-level planning and reasoning. Our designed action configuration is sufficient for this purpose, and VAB-OmniGibson supports closed-loop evaluation in a high-level manner, as the agent must understand environmental feedback and recover from errors (demonstrated in Figure 2).

Once again, we thank all the reviewers for their constructive reviews and suggestions, which have significantly contributed to the improvement of our work.

Best regards,

The authors of VisualAgentBench

---

### Public Comment · ~Jian_Yao4 · 2025-02-05
**Thank you for releasing the benchmark**

Thank you for your work on this project. I am interested in knowing when the training dataset will be available to the public.

---

### Meta-Review · Area_Chair_6rMP · 2024-12-24

**Metareview:**

This paper proposes a unified benchmark that supports investigation of LLMs/VLMs/LMMs for decision-making, including for Embodied AI, GUI control, and Visual Design. A number of baselines/models (both open and closed) and an analysis of synthetic data considerations is conducted. Reviewers appreciated the comprehensiveness of the LMM benchmark, extensive experimentation, insightful analysis, and importantly inclusion of SFT trajectories. Concerns included lack of rigor in justifying/selecting the particular benchmarks chosen, insufficient explanation of the model performance (especially failures/error modes), quality of the SFT trajectories, and lack of proxy metrics for progress to the goal that can aid understanding of interactive behavior. In response, the authors provided additional error-mode and error-recovery behavior analysis as well as proxy metrics for progress, among other things. After rebuttal, reviewers 4jCV, oNsV, and CJyi had positive scores, while reviewer cYwD still had concerns that the selection of datasets was not rigorous and the simplification of OmniGibson as a robotics benchmark.

  After considering all of the materials, I overall agree with the three positive reviewers. Specifically, unifying high-level decision-making problems, and not focusing on difficult low-level problems such as robot grasping and manipulation, is still a valuable contribution given that even such high-level policies do not fair that well even with state-of-art LMMs. The contribution to the community made through the effort to including SFT trajectories and an overall unified benchmark outweighs some of the remaining limitations mentioned. I highly encourage the authors to incorporate elements of the discussion and especially analysis (e.g. failure modes), and perhaps even strengthen the benchmark further thorugh a reasoned-out inclusion of additional benchmarks that can test new axes of LMM capabilities and provide more insightful analysis.

**Additional Comments On Reviewer Discussion:**

Reviewers raised a number of concerns about analysis, etc. that were addressed. Reviewer cYwD still maintained concerns about the simplification of the robot benchmarks, but overall I believe that the effort made in creating the benchmark, testing the large number of models, and analysis of the results, all outweigh this limitation.

---

### Decision · Program_Chairs · 2025-01-22

Accept (Poster)